# Pure Transformers are Powerful Graph Learners

Jinwoo Kim[1][*]   Tien Dat Nguyen[1]   Seonwoo Min[2]   Sungjun Cho[2]
Moontae Lee[2,3]   Honglak Lee[2][†]   Seunghoon Hong[1,2][†]
[1]KAIST  [2]LG AI Research  [3]University of Illinois Chicago

## Abstract

We show that standard Transformers without graph-specific modifications can lead to promising results in graph learning both in theory and practice. Given a graph, we simply treat all nodes and edges as independent tokens, augment them with token embeddings, and feed them to a Transformer. With an appropriate choice of token embeddings, we prove that this approach is theoretically at least as expressive as an invariant graph network (2-IGN) composed of equivariant linear layers, which is already more expressive than all message-passing Graph Neural Networks (GNN). When trained on a large-scale graph dataset (PCQM4Mv2), our method coined **Tokenized Graph Transformer (TokenGT)** achieves significantly better results compared to GNN baselines and competitive results compared to Transformer variants with sophisticated graph-specific inductive bias. Our implementation is available at `https://github.com/jw9730/tokengt`.

## 1 Introduction

In recent years, Transformer [68] has served as a versatile architecture in a broad class of machine learning problems, such as natural language processing [17, 7], computer vision [18], and reinforcement learning [9], to name a few. It is because the fully-attentional structure of Transformer is general and powerful enough to take, process, and relate inputs and outputs of arbitrary structures, eliminating a need for data- and task-specific inductive bias to be baked into the network architecture. Combined with large-scale training, it opens up a new chapter for building a versatile model that can solve a wide range of problems involving diverse data modalities and even a mixture of modalities [31, 30, 57].

In graph learning domain, inspired by the breakthroughs, multiple works tried combining self-attention into graph neural network (GNN) architecture where message passing was previously dominant [50]. As global self-attention across nodes cannot reflect the graph structure, however, these methods introduce *graph-specific* architectural modifications. This includes restricting self-attention to local neighborhoods [69, 51, 19], using global self-attention in conjunction with message-passing GNN [58, 43, 34], and injecting edge information into global self-attention via attention bias [72, 78, 29, 54]. Despite decent performance, such modifications can be a limiting constraint in terms of versatility, especially considering future integration to multi-task and multi-modal general-purpose attentional architectures [31]. In addition, deviating from pure self-attention, these methods may inherit the issues of message-passing such as oversmoothing [40, 8, 52], and become incompatible with useful engineering techniques *e.g.*, linear attention [65] developed for standard self-attention.

Instead, we explore the opposite direction of *applying a standard Transformer directly for graphs*. For this, we treat all nodes and edges as independent tokens, augment them with appropriate token-wise embeddings, and feed the tokens as input to the standard Transformer. The model operates identically to Transformers used in language and vision; each node or edge is treated as a token, identical to the words in a sentence or patches of an image [68, 18]. Perhaps surprisingly, we show that this simple approach yields a powerful graph learner both in theory and practice.

---

[*]Work done during an internship at LG AI Research.
[†]Equal corresponding authors.

36th Conference on Neural Information Processing Systems (NeurIPS 2022).

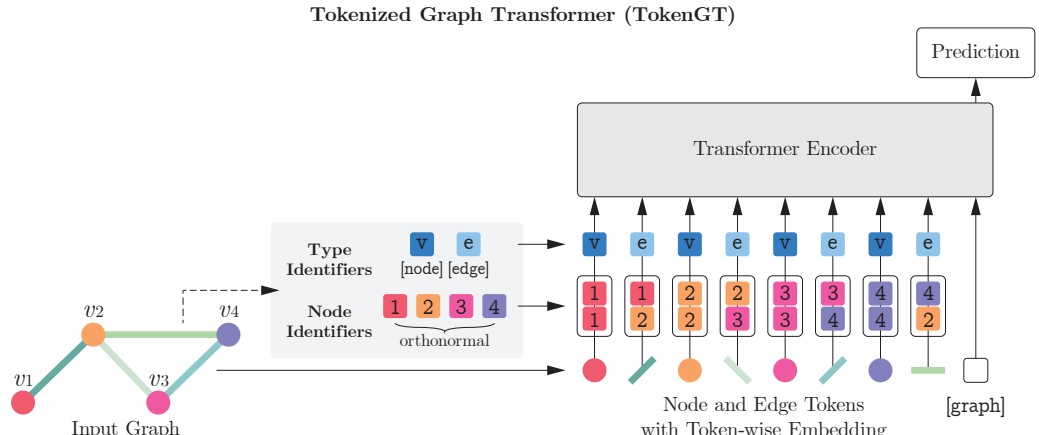

Figure 1: Overview of Tokenized Graph Transformer (TokenGT). We treat all nodes and edges of an input graph as independent tokens, augment them with orthonormal node identifiers and trainable type identifiers, and feed them to a standard Transformer encoder. For graph-level prediction, we follow the common practice [17, 18] of using an extra trainable [graph] token.

As a key theoretical result, we prove that with appropriate token-wise embeddings, self-attention over the node and edge tokens can approximate *any* permutation equivariant linear operator on a graph [47]. Remarkably, we show that a very simple choice of embedding composed of *node identifiers* and *type identifiers* is sufficient for accurate approximation. This provides a solid theoretical guarantee that, with the embeddings and enough attention heads, a Transformer is at least as expressive as a second-order invariant graph network (2-IGN) [47, 34], which is already more expressive than all message-passing GNNs [21]. This also immediately grants the model with the expressive power at least as good as the 2-dimensional Weisfeiler-Lehman (WL) graph isomorphism test [46], which is often sufficient for real-world graph data [83]. We further extend our theoretical result to *hypergraphs* with order-$k$ hyperedges, showing that a Transformer with order-$k$ generalized token embeddings is at least as expressive as $k$-IGN and, consequently $k$-WL test.

We test our model, named Tokenized Graph Transformer (TokenGT), mainly on the PCQM4Mv2 large-scale quantum chemical property prediction dataset containing 3.7M molecular graphs [27]. Even though TokenGT involves minimal graph-specific architectural modifications, it performs significantly better than all GNN baselines, showing that the advantages of Transformer architecture combined with large-scale training surpass the benefit of hard inductive bias of GNNs. Furthermore, TokenGT achieves competitive performance compared to Transformer variants with strong graph-specific modifications [78, 29, 54]. Finally, we demonstrate that TokenGT can naturally utilize efficient approximations in Transformers in contrast to these variants, using kernel attention [11] that enables linear computation cost without much degradation in performance.

## 2   Tokenized Graph Transformer (TokenGT)

In this section, we present the Tokenized Graph Transformer (TokenGT), a pure Transformer architecture for graphs with token-wise embeddings composed of *node identifiers* and *type identifiers* (Figure 1). Our goal in this section is to provide a practical overview – for theoretical analysis of the architecture, we guide the readers to Section 3.

Let $\mathcal{G} = (\mathcal{V}, \mathcal{E})$ an input graph with $n$ nodes $\mathcal{V} = \{v_1, ..., v_n\}$ and $m$ edges $\mathcal{E} = \{e_1, ..., e_m\} \subseteq \mathcal{V}^2$, associated with features $\mathbf{X}^{\mathcal{V}} \in \mathbb{R}^{n \times C}$ and $\mathbf{X}^{\mathcal{E}} \in \mathbb{R}^{m \times C}$, respectively. We treat each node and edge as an independent token (thus $(n + m)$ tokens in total) and construct their features by $\mathbf{X} = [\mathbf{X}^{\mathcal{V}}; \mathbf{X}^{\mathcal{E}}] \in \mathbb{R}^{(n+m) \times C}$. A naïve way to process a graph is to directly provide the tokens $\mathbf{X}$ as input to a Transformer, but it is inappropriate as graph connectivity is discarded. To thoroughly represent graph structure, we augment the tokens $\mathbf{X}$ with token-wise embeddings, more specifically orthonormal *node identifiers* used for representing the connectivity of the tokens and trainable *type identifiers* that encode whether a token is a node or an edge. Despite the simplicity, we show that a Transformer applied on these embeddings is a theoretically powerful graph learner.

**Node Identifiers**    The first component of token-wise embedding is the orthonormal node identifier that we use to represent the connectivity structure given in the input graph.

For a given input graph $\mathcal{G} = (\mathcal{V}, \mathcal{E})$, we first produce $n$ node-wise orthonormal vectors $\mathbf{P} \in \mathbb{R}^{n \times d_p}$ that we refer to as node identifiers. Then, we augment the tokens $\mathbf{X}$ with node identifiers as follows.

- For each node $v \in \mathcal{V}$, we augment the token $\mathbf{X}_v$ as $[\mathbf{X}_v, \mathbf{P}_v, \mathbf{P}_v]$.
- For each edge $(u, v) \in \mathcal{E}$, we augment the token $\mathbf{X}_{(u,v)}$ as $[\mathbf{X}_{(u,v)}, \mathbf{P}_u, \mathbf{P}_v]$.

Intuitively, a Transformer operating on the augmented tokens can fully recognize the connectivity structure of the graph since comparing the node identifiers between a pair of tokens reveals their *incidence* information. For instance, we can tell if an edge $e = (u, v)$ is connected with a node $k$ through dot-product (attention) since $[\mathbf{P}_u, \mathbf{P}_v][\mathbf{P}_k, \mathbf{P}_k]^\top = 1$ if and only if $k \in (u, v)$ and 0 otherwise. This allows the Transformer to identify and exploit the connectivity structure of a graph, for instance by putting more weights on incident pairs when the local operation is important.

Notably, as the node identifiers $\mathbf{P}$ are *only* required to be orthonormal, we have a large degree of freedom in implementation choices. We outline two practical methods below as examples. Their implementation details can be found in Appendix A.3.1.

- Orthogonal random features (ORFs), *e.g.*, rows of random orthogonal matrix $\mathbf{Q} \in \mathbb{R}^{n \times n}$ obtained with QR decomposition of random Gaussian matrix $\mathbf{G} \in \mathbb{R}^{n \times n}$ [79, 12].
- Laplacian eigenvectors obtained from eigendecomposition of graph Laplacian matrix, *i.e.*, rows of $\mathbf{U}$ from $\mathbf{\Delta} = \mathbf{I} - \mathbf{D}^{-1/2}\mathbf{A}\mathbf{D}^{-1/2} = \mathbf{U}^\top \mathbf{\Lambda} \mathbf{U}$, where $\mathbf{A} \in \mathbb{R}^{n \times n}$ is adjacency matrix, $\mathbf{D}$ is degree matrix, and $\mathbf{\Lambda}, \mathbf{U}$ correspond to eigenvalues and eigenvectors respectively [20].

Among the two methods, node identifiers generated as ORFs do not encode any information about the graph structure as they are entirely random. This means the Transformer that operates on the ORF-based node identifiers needs to compile and recognize graph structure only from the incidence information provided by the node identifiers. Although this is challenging, perhaps surprisingly, we empirically show in Section 5 that Transformers are strong enough to learn meaningful structural representations out of ORF-based node identifiers and outperform GNNs on large-scale task.

In contrast to ORFs, Laplacian eigenvectors provide a kind of graph positional embeddings (graph PEs) that describes the distance between nodes on a graph. Due to the positional information, it yields better performance compared to ORFs in our experiments in Section 5. One interesting aspect of Laplacian eigenvectors is that they can be viewed as a generalization of sinusoidal positional embeddings of NLP Transformers to graphs, as the eigenvectors of 1D chain graphs are sine and cosine functions [20]. Thus, by choosing Laplacian eigenvectors as node identifiers, our approach can be interpreted as a direct extension of the NLP Transformer for inputs involving relational structures.

**Type Identifiers**    The second component of token-wise embedding is the trainable type identifier that encodes whether a token is node or edge. For a given input graph $\mathcal{G} = (\mathcal{V}, \mathcal{E})$, we first prepare a trainable parameter matrix $\mathbf{E} = [\mathbf{E}^{\mathcal{V}}; \mathbf{E}^{\mathcal{E}}] \in \mathbb{R}^{2 \times d_e}$ that contains two type identifiers $\mathbf{E}^{\mathcal{V}}$ and $\mathbf{E}^{\mathcal{E}}$ for nodes and edges respectively. Then, we further augment the tokens with type identifiers as follows.

- For each node $v \in \mathcal{V}$, we augment the token $[\mathbf{X}_v, \mathbf{P}_v, \mathbf{P}_v]$ as $[\mathbf{X}_v, \mathbf{P}_v, \mathbf{P}_v, \mathbf{E}^{\mathcal{V}}]$.
- For each edge $(u, v) \in \mathcal{E}$, we augment the token $[\mathbf{X}_{(u,v)}, \mathbf{P}_u, \mathbf{P}_v]$ as $[\mathbf{X}_{(u,v)}, \mathbf{P}_u, \mathbf{P}_v, \mathbf{E}^{\mathcal{E}}]$.

These embeddings provide information on whether a given token is a node or an edge, which is critical, *e.g.*, when an attention head tries to attend specifically to node tokens and ignore edge tokens.

**Main Transformer**    With node identifiers and type identifiers, we obtain augmented token features $\mathbf{X}^{in} \in \mathbb{R}^{(n+m) \times (C + 2d_p + d_e)}$, which is further projected by a trainable matrix $w^{in} \in \mathbb{R}^{(C + 2d_p + d_e) \times d}$ to be an input to Transformer. For graph-level prediction, we prepend a special token [graph] with trainable embedding $\mathbf{X}_{[\texttt{graph}]} \in \mathbb{R}^d$ similar to BERT [17] and ViT [18]. We utilize the feature of [graph] token at the output of the encoder as the graph representation, on which a linear prediction head is applied to produce the final graph-level prediction. Overall, the tokens $\mathbf{Z}^{(0)} = [\mathbf{X}_{[\texttt{graph}]}; \mathbf{X}^{in} w^{in}] \in \mathbb{R}^{(1+n+m) \times d}$ are used as the input to the main encoder. As an encoder, we adopt the standard Transformer [68], which is an alternating stack of multihead self-attention layers (MSA) and feedforward MLP layers. We provide further details in Appendix A.1.1.

**Inductive Bias**  Similar to Transformers in language and vision [17, 18], Tokenized Graph Transformer treats input nodes and edges as independent tokens and applies self-attention to them. This approach leads to much less inductive bias than current GNNs, where the sparse graph structure, or more fundamentally, *permutation symmetry* of graphs is deliberately baked into each layer [21, 47, 46, 34]. For TokenGT, such information is provided entirely as a part of input using token-wise embeddings, and the model has to learn how to interpret and utilize the information from data. Although such weak inductive bias might raise questions on the expressiveness of the model, our theoretical analysis in Section 3 shows that TokenGT is a powerful graph learner thanks to the token-wise embeddings and expressive power of self-attention. For example, we show that TokenGT is more expressive than all message-passing GNNs under the framework of Gilmer et al. (2017) [21].

## 3  Theoretical Analysis

We now present our theory. Our key result is that TokenGT, a standard Transformer with node and type identifiers presented in Section 2, is provably *at least* as expressive as the second-order Invariant Graph Network (2-IGN [47]), which is built upon *all* possible permutation equivariant linear layers on a graph. This provides solid theoretical guarantees for TokenGT, such as being at least as powerful as the 2-WL graph isomorphism test and more expressive than all message-passing GNNs. Our theory is based on a general framework on hypergraphs represented as higher-order tensors, which leads to the formulation of order-$k$ TokenGT that is at least as expressive as order-$k$ IGN ($k$-IGN [47]).

### 3.1  Preliminary: Permutation Symmetry and Invariant Graph Networks

**Representing and Processing Sets and (Hyper)Graphs**  For a set of $n$ nodes, we often represent their features as $\mathbf{X} \in \mathbb{R}^{n \times d}$ where $\mathbf{X}_i \in \mathbb{R}^d$ is the feature of the $i$-th node. The set is unordered and, therefore, should be treated invariant to the renumbering of the nodes. Let $S_n$ the symmetric group or the group of permutations $\pi$ on $[n] = \{1, ..., n\}$. By $\pi \cdot \mathbf{X}$ we denote permuting rows of $\mathbf{X}$ with $\pi$, *i.e.*, $(\pi \cdot \mathbf{X})_i = \mathbf{X}_{\pi^{-1}(i)}$. Here, $\mathbf{X}$ and $\pi \cdot \mathbf{X}$ represent the identical set for all $\pi \in S_n$.

Generally, we consider (hyper)graphs represented as order-$k$ tensor $\mathbf{X} \in \mathbb{R}^{n^k \times d}$ with feature $\mathbf{X_i} = \mathbf{X}_{i_1,...,i_k} \in \mathbb{R}^d$ attached to (hyper)edge represented as multi-index $\mathbf{i} = (i_1, ..., i_k) \in [n]^k$. Similar to sets, the tensor should be treated invariant to node renumbering by any $\pi \in S_n$ that acts on $\mathbf{X}$ by $(\pi \cdot \mathbf{X})_{\mathbf{i}} = \mathbf{X}_{\pi^{-1}(\mathbf{i})}$ where $\pi^{-1}(\mathbf{i}) = (\pi^{-1}(i_1), ..., \pi^{-1}(i_k))$. That is, $\mathbf{X}$ and $\pi \cdot \mathbf{X}$ represent the identical (hyper)graph for all $\pi$. Due to such symmetry, to build a function $F(\mathbf{X}) \approx T$ for tensor $\mathbf{X}$ and target $T$, a suitable way is to make them *invariant* $F(\pi \cdot \mathbf{X}) = F(\mathbf{X})$ when the target is a vector or *equivariant* $F(\pi \cdot \mathbf{X}) = \pi \cdot F(\mathbf{X})$ when the target is also a tensor, for all $\mathbf{X} \in \mathbb{R}^{n^k \times d}$ and $\pi \in S_n$.

In our theoretical analysis, we work on order-$k$ dense tensor representation $\mathbf{X} \in \mathbb{R}^{n^k \times d}$ of a graph as they can represent node features ($k = 1$), edge features ($k = 2$), or hyperedge features ($k > 2$) in a unified manner. This is interchangeable but slightly different from the sparse representation of a graph with edge set $\mathcal{E}$ used in Section 2. Nevertheless, in Section 5 we empirically verify that our key theoretical findings work equally well for dense and sparse graphs.

**Invariant Graph Network**  We mainly develop our theoretical analysis upon *Invariant Graph Networks (IGNs)* [47, 46], a family of expressive graph networks derived from the permutation symmetry of tensor representation of graphs. Here we provide a summary. In general, we define:

**Definition 1.** *An order-$k$ Invariant Graph Network ($k$-IGN) is a function $F_k : \mathbb{R}^{n^k \times d_0} \to \mathbb{R}$ written as the following:*

$$F_k = \text{MLP} \circ L_{k \to 0} \circ L_{k \to k}^{(T)} \circ \sigma \circ ... \circ \sigma \circ L_{k \to k}^{(1)}, \tag{1}$$

*where each $L_{k \to k}^{(t)}$ is equivariant linear layer [47] from $\mathbb{R}^{n^k \times d_{t-1}}$ to $\mathbb{R}^{n^k \times d_t}$, $\sigma$ is activation function, and $L_{k \to 0}$ is a invariant linear layer from $\mathbb{R}^{n^k \times d_T}$ to $\mathbb{R}$.*

A body of previous work have shown appealing theoretical properties of $k$-IGN, including universal approximation [48] and alignment to $k$-Weisfeiler-Lehman ($k$-WL) graph isomorphism test [46, 10]. In particular, it is known that $k$-IGNs are theoretically at least as powerful as the $k$-WL test [46]. It is also known that 2-IGNs are already more expressive [47, 34] than all message-passing GNNs under the framework of Gilmer et al. (2017) [21].

The core building block of IGN is invariant and equivariant *linear* layers [47] with maximal expressiveness while respecting node permutation symmetry. The layers are defined as follows:

**Definition 2.** *An equivariant linear layer is a function $L_{k \to l} : \mathbb{R}^{n^k \times d} \to \mathbb{R}^{n^l \times d'}$ written as follows for order-$k$ input $\mathbf{X} \in \mathbb{R}^{n^k \times d}$:*

$$L_{k \to l}(\mathbf{X})_{\mathbf{i}} = \sum_{\mu} \sum_{\mathbf{j}} \mathbf{B}_{\mathbf{i},\mathbf{j}}^{\mu} \mathbf{X}_{\mathbf{j}} w_{\mu} + \sum_{\lambda} \mathbf{C}_{\mathbf{i}}^{\lambda} b_{\lambda}, \tag{2}$$

*where $\mathbf{i} \in [n]^l, \mathbf{j} \in [n]^k$ are multi-indices, $w_{\mu} \in \mathbb{R}^{d \times d'}$, $b_{\lambda} \in \mathbb{R}^{d'}$ are weight and bias parameters, and $\mathbf{B}^{\mu} \in \mathbb{R}^{n^{l+k}}$ and $\mathbf{C}^{\lambda} \in \mathbb{R}^{n^l}$ are binary basis tensors corresponding to order-$(l+k)$ and order-$l$ equivalence classes $\mu$ and $\lambda$, respectively. Invariant linear layer is a special case of $L_{k \to l}$ with $l = 0$.*

We provide the definition of the equivalence classes and basis tensors in Appendix A.1.1. For now, it is sufficient to know that the basis tensors are binary tensors that form the orthogonal basis of the full space of linear equivariant layers. In general, in Eq. (2) it is known that there exists bell$(k + l)$ number of basis tensors $\mathbf{B}^{\mu}$ for the weight and bell$(l)$ number of basis tensors $\mathbf{C}^{\lambda}$ for the bias.

## 3.2 Can Self-Attention Approximate Equivariant Basis?

Now, we present an intuition that connects Transformer (Section 2) and equivariant linear layer (Definition 2). For that, we write out the multihead self-attention layer as follows:

$$\text{MSA}(\mathbf{X})_i = \sum_{h=1}^{H} \sum_{j} \boldsymbol{\alpha}_{ij}^h \mathbf{X}_j w_h^V w_h^O \text{ where } \boldsymbol{\alpha}^h = \text{softmax} \left( \frac{\mathbf{X} w_h^Q (\mathbf{X} w_h^K)^{\top}}{\sqrt{d_H}} \right), \tag{3}$$

where $H$ is number of heads, $d_H$ is head size, and $w_h^Q, w_h^K \in \mathbb{R}^{d \times d_H}$, $w_h^V \in \mathbb{R}^{d \times d_v}$ $w_h^O \in \mathbb{R}^{d_v \times d}$.

Our intuition is that the weighted sum of values with self-attention matrix $\boldsymbol{\alpha}^h$ in Eq. (3) is analogous to the masked sum with basis tensor $\mathbf{B}^{\mu}$ in Eq. (2) up to normalization. This naturally leads to the following question: for a given equivariant layer $L_{k \to k} : \mathbb{R}^{n^k \times d} \to \mathbb{R}^{n^k \times d}$, can we use a Transformer layer with multihead self-attention MSA : $\mathbb{R}^{N \times d'} \to \mathbb{R}^{N \times d'}$ with $N = n^k$ to accurately approximate $L_{k \to k}$ by having $H = \text{bell}(2k)$ attention heads approximate each equivariant basis $\mathbf{B}^{\mu}$?

We show that this can be possible, but only if we provide appropriate auxiliary information to input. For example, let us consider first-order layer $L_{1 \to 1}$. The layer has bell$(2) = 2$ basis tensors $\mathbf{B}^{\mu_1} = \mathbf{I}$ and $\mathbf{B}^{\mu_2} = \mathbf{1}\mathbf{1}^{\top} - \mathbf{I}$ for the weight, and bell$(1) = 1$ basis tensor $\mathbf{C}^{\lambda_1} = \mathbf{1}$ for the bias. Given an input set $\mathbf{X} \in \mathbb{R}^{n \times d}$ it computes the following with $w_1, w_2 \in \mathbb{R}^{d \times d}$, $b \in \mathbb{R}^d$:

$$L_{1 \to 1}(\mathbf{X}) = \mathbf{I}\mathbf{X} w_1 + (\mathbf{1}\mathbf{1}^{\top} - \mathbf{I})\mathbf{X} w_2 + \mathbf{1}b^{\top}. \tag{4}$$

Now consider approximating basis tensor $\mathbf{B}^{\mu_1} = \mathbf{I}$ with an attention matrix $\boldsymbol{\alpha}^1$. The approximation is accurate when $i$-th query always only attends to $i$-th key and ignores the rest. To achieve the attention structure consistently, *i.e.*, agnostic to input $\mathbf{X}$, we need to provide auxiliary input that self-attention can "latch onto" to faithfully approximate $\boldsymbol{\alpha}^1 \approx \mathbf{I}$. Without this, attention must entirely rely on the inputs $\mathbf{X}$, which is unreliable and can lead to approximation failure, *e.g.*, when $\mathbf{X}$ has repeated rows.

For the auxiliary information, we prepare $n$ node-wise orthonormal vectors $\mathbf{P} \in \mathbb{R}^{n \times d_p}$ (note that this is identical to node identifiers in Section 2), and augment the input to $\mathbf{X}^{in} = [\mathbf{X}, \mathbf{P}] \in \mathbb{R}^{n \times (d+d_p)}$. Let us assume that the query and key projections in Eq. (3) ignore $\mathbf{X}$ and only leave $\mathbf{P}$ scaled by $\sqrt{a}$ with $a > 0$. Then attention matrix is computed as $\boldsymbol{\alpha}^1 = \text{softmax}(\mathbf{S})$ where $\mathbf{S}_{ij} = a\mathbf{P}_i^{\top} \mathbf{P}_j$. Here, due to the orthonormality of $\mathbf{P}$, we have $\mathbf{P}_i^{\top} \mathbf{P}_j = 1$ only if $i = j$ and otherwise 0, which leads to $\mathbf{S} = a\mathbf{I}$. With $a \to \infty$ by scaling up the query and key projection weights, the softmax becomes arbitrarily close to the hardmax operator, and we obtain the following:

$$\boldsymbol{\alpha}^1 = \text{softmax}(a\mathbf{I}) \to \mathbf{I} \text{ as } a \to \infty. \tag{5}$$

Thus, self-attention can utilize the auxiliary information $\mathbf{P}$ to achieve an input-agnostic approximation of $\boldsymbol{\alpha}^1$ to $\mathbf{I}$. Notably, we can achieve a similar approximation for $\mathbf{B}^{\mu_2} = \mathbf{1}\mathbf{1}^{\top} - \mathbf{I}$ using the *same* $\mathbf{P}$ by flipping the sign of keys, which gives $\boldsymbol{\alpha}^2 = \text{softmax}(-a\mathbf{I})$ due to orthonormality. By sending $a \to \infty$, now attention from the $i$-th query to the $i$-th key is suppressed, and we obtain the following:

$$\boldsymbol{\alpha}^2 = \text{softmax}(-a\mathbf{I}) \to \frac{1}{n-1}(\mathbf{1}\mathbf{1}^{\top} - \mathbf{I}) \text{ as } a \to \infty. \tag{6}$$

Note that this approximation is accurate only up to row normalization as rows of $\boldsymbol{\alpha}^2$ always sum to one due to softmax, while $\mathbf{B}^{\mu_2} = \mathbf{1}\mathbf{1}^\top - \mathbf{I}$ is binary. In our proofs of the theoretical results, we perform appropriate denormalization with MLP after MSA to achieve an accurate approximation.

Overall, we see that simple auxiliary input $\mathbf{P}$ suffices for two attention heads to approximate the equivariant basis of $L_{1 \to 1}$ accurately. We now question the following. Given appropriate auxiliary information as input, can a Transformer layer with $\mathrm{bell}(2k)$ attention heads accurately approximate $L_{k \to k}$ by having each head approximate each equivariant basis $\mathbf{B}^\mu$? What would be the sufficient auxiliary input? We answer the question by showing that, with (order-$k$ generalized) node and type identifiers presented in Section 2, Transformer layers can accurately approximate equivariant layers $L_{k \to k}$ via input-agnostic head-wise approximation of each equivariant basis.

### 3.3 Pure Transformers are Powerful Graph Learners

We now present our main theoretical results that extend the discussions in Section 3.2 to any order $k$. Note that $k = 2$ corresponds to TokenGT for graphs presented in Section 2. With $k > 2$, we naturally extend TokenGT to hypergraphs. All proofs can be found in Appendix A.1.

We first introduce generalized node and type identifiers (Section 2) for order-$k$ tensors $\mathbf{X} \in \mathbb{R}^{n^k \times d}$. We define the node identifier $\mathbf{P} \in \mathbb{R}^{n \times d_p}$ as an orthonormal matrix with $n$ rows, and the type identifier as a trainable matrix $\mathbf{E} \in \mathbb{R}^{\mathrm{bell}(k) \times d_e}$ that contains $\mathrm{bell}(k)$ rows $\mathbf{E}^{\gamma_1}, ..., \mathbf{E}^{\gamma_{\mathrm{bell}(k)}}$, each of which is designated for an order-$k$ equivalence class $\gamma$. Then, we augment each entry of input tensor as $[\mathbf{X}_{i_1,...,i_k}, \mathbf{P}_{i_1}, ..., \mathbf{P}_{i_k}, \mathbf{E}^\gamma]$ where $(i_1, ..., i_k) \in \gamma$.

Let us exemplify. For $k = 1$ (sets), each $i$-th entry is augmented as $[\mathbf{X}_i, \mathbf{P}_i, \mathbf{E}^{\gamma_1}]$, consistent with our discussion in Section 3.2. For $k = 2$ (graphs), each $(i, i)$-th entry is augmented as $[\mathbf{X}_{ii}, \mathbf{P}_i, \mathbf{P}_i, \mathbf{E}^{\gamma_1}]$ and each $(i, j)$-th entry $(i \neq j)$ is augmented as $[\mathbf{X}_{ij}, \mathbf{P}_i, \mathbf{P}_j, \mathbf{E}^{\gamma_2}]$. This is consistent with TokenGT in Section 2, which augments nodes with $\mathbf{E}^{\mathcal{V}} = \mathbf{E}^{\gamma_1}$ and edges with $\mathbf{E}^{\mathcal{E}} = \mathbf{E}^{\gamma_2}$.

With node and type identifiers, we obtain augmented order-$k$ tensor $\mathbf{X}^{in} \in \mathbb{R}^{n^k \times (d + k d_p + d_e)}$. We use a trainable projection $w^{in} \in \mathbb{R}^{(d + k d_p + d_e) \times d_{\mathcal{T}}}$ to map them to hidden dimension $d_{\mathcal{T}}$ of a Transformer. We now show that self-attention on $\mathbf{X}^{in} w^{in}$ can accurately approximate equivariant basis:

**Lemma 1.** *For all $\mathbf{X} \in \mathbb{R}^{n^k \times d}$ and their augmentation $\mathbf{X}^{in}$, self-attention coefficients $\boldsymbol{\alpha}^h$ (Eq. (3)) computed with $\mathbf{X}^{in} w^{in}$ can approximate any basis tensor $\mathbf{B}^\mu \in \mathbb{R}^{n^{2k}}$ of order-$k$ equivariant linear layer $L_{k \to k}$ (Definition 2) to arbitrary precision up to normalization.*

Consequently, with the node and type identifiers, a collection of $\mathrm{bell}(2k)$ attention heads can approximate the collection of all basis tensors of order-$k$ equivariant layer. This leads to the following:

**Theorem 1.** *For all $\mathbf{X} \in \mathbb{R}^{n^k \times d}$ and their augmentation $\mathbf{X}^{in}$, a Transformer layer with $\mathrm{bell}(2k)$ self-attention heads that operates on $\mathbf{X}^{in} w^{in}$ can approximate an order-$k$ equivariant linear layer $L_{k \to k}(\mathbf{X})$ (Definition 2) to arbitrary precision.*

While the approximation in Lemma 1 is only accurate up to normalization over inputs (keys) due to softmax normalization, for the approximation in Theorem 1 we perform appropriate denormalization using MLP after multihead self-attention and can obtain an accurate approximation.

By extending the result to multiple layers, we arrive at the following:

**Theorem 2.** *For all $\mathbf{X} \in \mathbb{R}^{n^k \times d}$ and their augmentation $\mathbf{X}^{in}$, a Transformer composed of $T$ layers that operates on $\mathbf{X}^{in} w^{in}$ followed by sum-pooling and MLP can approximate an $k$-IGN $F_k(\mathbf{X})$ (Definition 1) to arbitrary precision.*

This directly leads to the following corollary:

**Corollary 1.** *A Transformer on node and type identifiers in Theorem 2 is at least as expressive as $k$-IGN composed of order-$k$ equivariant linear layers.*

Corollary 1 allows us to draw previous theoretical results on the expressiveness of $k$-IGN [46, 47, 34] and use them to lower-bound the provable expressiveness of a standard Transformer:

**Corollary 2.** *A Transformer on node and type identifiers in Theorem 2 is at least as powerful as $k$-WL graph isomorphism test and is more expressive than all message-passing GNNs within the framework of Gilmer et al. (2017) [21].*

# 4 Related Work

We outline relevant work including equivariant neural networks, theory on expressive power of Transformers and their connection to modeling equivariance, and Transformers for graphs.

**Equivariant Neural Networks** A machine learning task is often invariant or equivariant to specific symmetry of input data, *e.g.*, image classification is invariant to the translation of an input image. A large body of literature advocated baking the invariance or equivariance into a neural network as a type of inductive bias (*e.g.*, translation equivariance of image convolution), showing that it reduces the number of parameters and improves generalization for a wide range of learning tasks involving various geometric structures [13, 14, 73, 66, 49, 53, 60, 6, 34, 39]. Ravanbakhsh et al. (2017) [56] showed that any equivariant layer for discrete group actions is equivalent to a specific parameter sharing structure. Zaheer et al. (2017) [82] and Maron et al. (2019) [47] derived the parameter sharing for node permutation-symmetric data (sets and (hyper)graphs), which gives the maximally expressive equivariant linear layers and $k$-IGN in Section 3.1. The work on equivariant neural networks underlie our theory of how a standard Transformer can be a powerful learner for sets and (hyper)graphs.

**Expressive Power of Transformers and Its Connection to Equivariance** Recent work involving Transformers often focus on minimizing the domain- and task-specific inductive bias and scaling the model and data so that any useful computation structure can be learned [18, 31, 30, 7, 17, 9, 39]. The success of this approach is, to some degree, attributed to the high expressive power of Transformers that allows learning diverse functions suited for the data at hand [81, 39, 3, 4, 41]. Recent theory has shown that Transformers are expressive enough to even model certain equivariant functions [1, 15, 39]. Andreoli et al. (2019) [1] cast self-attention and convolution into a unified framework using basis tensors similar to ones in Section 3.1. Cordonnier et al. (2020) [15] advanced the idea and showed that Transformers with relative positional encodings can approximate any image convolution layers. Lee et al. (2019) [39] and Kim et al. (2021) [34] showed that Transformers can model equivariant linear layers for sets [82], which can be viewed as the first-order case of our theory (see Section 3.2). To our knowledge, our work is the first to show that standard Transformers are expressive enough to provably model maximally expressive equivariant layers and $k$-IGN for (hyper)graphs with $k \geq 2$.

**Transformers for Graphs** Unlike in language and vision, developing Transformers for graphs is challenging due to **(1)** the presence of edge connectivity and **(2)** the absence of canonical node ordering that prevents adopting simple positional encodings [50]. To incorporate the connectivity of edges, early methods restricted self-attention to local neighborhoods (thus reducing to message-passing) [19, 51, 69] or used global self-attention with auxiliary message-passing modules [58, 43]. As message-passing suffers from limited expressive power [77] and oversmoothing [40, 8, 52], recent works often discard them and use global self-attention on nodes with heuristic modifications to process edges [78, 29, 54, 38, 42]. Ying et al. (2021) [78] proposed to inject edge encoding based on shortest paths through self-attention bias. Kreuzer et al. (2021) [38] proposed to incorporate edges into self-attention matrix via elementwise multiplication. On the contrary, we leave the self-attention unmodified and provide both nodes and edges with certain token-wise embeddings (Section 2) as its input. To incorporate graph structure into nodes, on the other hand, some approaches focus on developing graph positional encoding, *e.g.*, based on Laplacian eigenvectors [20, 42, 38]. While these can be directly incorporated into our work via auxiliary node identifiers for better performance, we leave this as future work. We further note that current graph Transformers that utilize Laplacian positional encoding rely heavily on heuristic edge encoding [29, 38] while ours does not. Another closely related approach is the Higher-order Transformer [34] which generalizes $k$-IGN with masked self-attention. While it is highly complex to implement due to hard-coded head-wise equivariant masks, our method can be implemented effortlessly using any available implementation of standard Transformer. Furthermore, our method is more flexible as the model can choose to use different attention heads to focus on a specific equivariant operator (*e.g.*, local propagation) if needed. We further discuss the difficulty in applying linear attention to graph Transformers in Appendix A.2.

# 5 Experiments

We first conduct a synthetic experiment that directly confirms our key claims in Lemma 1 (Section 3). Then, we empirically explore the capability of Tokenized Graph Transformer (TokenGT) (Section 2) using the PCQM4Mv2 large-scale quantum chemistry regression dataset [27]. We further present experiments on transductive node classification datasets involving large graphs in Appendix A.4.3.

Table 1: Second-order equivariant basis approximation. We report average and standard deviation of L2 error averaged over heads over 3 runs. For Random/ORF (first-order), we sample random embeddings independently for each token.

| node id. | type id. | dense input | | sparse input | |
|---|---|---|---|---|---|
| | | train L2 ↓ | test L2 ↓ | train L2 ↓ | test L2 ↓ |
| ✗ | ✗ | $47.95 \pm 0.600$ | $53.93 \pm 1.426$ | $29.88 \pm 0.450$ | $34.70 \pm 1.167$ |
| ✗ | ◯ | $32.38 \pm 0.448$ | $40.06 \pm 1.202$ | $15.92 \pm 0.275$ | $20.39 \pm 0.765$ |
| Random (first-order) | ◯ | $32.19 \pm 0.476$ | $32.49 \pm 3.687$ | $15.87 \pm 0.247$ | $16.56 \pm 0.904$ |
| ORF (first-order) | ◯ | $32.35 \pm 0.369$ | $39.87 \pm 1.263$ | $15.87 \pm 0.247$ | $16.56 \pm 0.908$ |
| Random | ✗ | $5.909 \pm 0.019$ | $5.548 \pm 0.090$ | $8.152 \pm 0.042$ | $8.270 \pm 0.285$ |
| ORF | ✗ | $5.472 \pm 0.035$ | $5.143 \pm 0.078$ | $7.167 \pm 0.025$ | $7.190 \pm 0.217$ |
| Laplacian eigenvector | ✗ | $1.899 \pm 3.050$ | $1.702 \pm 2.912$ | $0.288 \pm 0.019$ | $0.064 \pm 0.010$ |
| Random | ◯ | $0.375 \pm 0.009$ | $0.234 \pm 0.011$ | $0.990 \pm 0.108$ | $0.875 \pm 0.042$ |
| ORF | ◯ | $0.080 \pm 0.001$ | $0.009 \pm 5e\text{-}5$ | $0.129 \pm 0.002$ | $\mathbf{0.011 \pm 0.002}$ |
| Laplacian eigenvector | ◯ | $\mathbf{0.053 \pm 1.5e\text{-}5}$ | $\mathbf{0.005 \pm 1e\text{-}4}$ | $\mathbf{0.101 \pm 0.003}$ | $0.019 \pm 0.007$ |

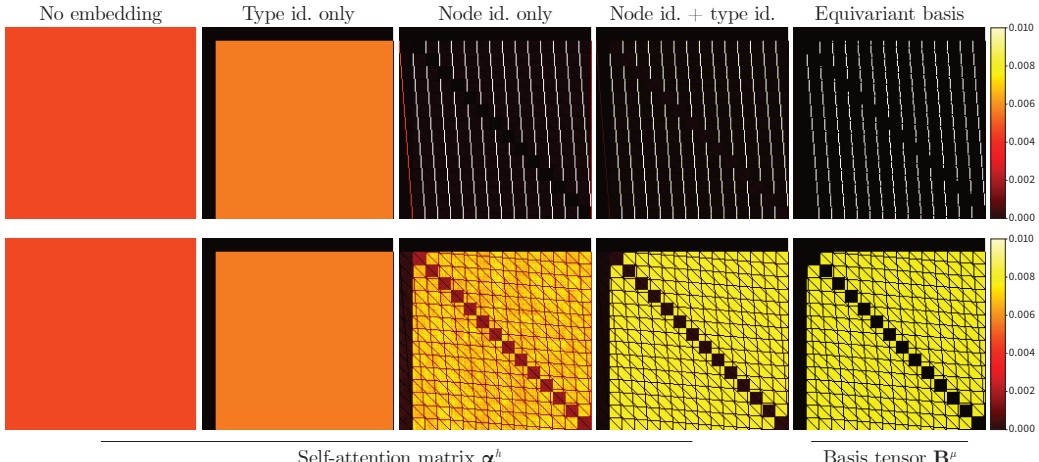

Figure 2: Self-attention maps learned under various node and type identifier configurations for two target equivariant basis tensors (out of 15). For better visualization, we clamp the entries by 0.01. Self-attention learns acute patterns coherent to equivariant basis when orthonormal node identifiers and type identifiers are both provided as input. More images can be found in Appendix A.4.1.

## 5.1 Approximating Second-Order Equivariant Basis

As in Theorem 1 and 2 (Section 3), our argument on the expressive power of TokenGT relies on its capability to approximate order-$k$ permutation equivariant linear layers $L_{k \to k}$ (Definition 2). Specifically, Lemma 1 states that such capability depends on the ability of each self-attention head $\boldsymbol{\alpha}^1, ..., \boldsymbol{\alpha}^H$ (Eq. (3)) to accurately approximate each equivariant basis $\mathbf{B}^{\mu_1}, ..., \mathbf{B}^{\mu_{\text{bell}(2k)}}$ (Definition 2) up to normalization.

We verify this claim for $k = 2$ (second-order; graphs) in a synthetic setup using Barabási-Albert random graphs. We use a multihead self-attention layer (Eq. (3)) with bell$(2 + 2) = 15$ heads and explicitly supervise head-wise attention scores $\boldsymbol{\alpha}^h$ to approximate each (normalized) equivariant basis tensor $\mathbf{B}^{\mu_h}$ by minimizing L2 loss. Having the layer hyperparameters fixed, we provide different combinations of node and type identifiers, and test if multihead self-attention can jointly approximate *all* 15 equivariant basis on unseen graphs. We experiment with both dense and sparse graph representations; for graphs with $n$ nodes and $m$ edges, the dense graph considers all $n^2$ pairwise edges as input as in Section 3, whereas the sparse graph considers only the present $m$ edges as in Section 2. Further details can be found in Appendix A.3.2.

We outline the results in Table 1. Consistent with Lemma 1, self-attention achieves accurate approximation of equivariant basis only when both the orthonormal node identifiers and type identifiers are given. Here, Laplacian eigenvectors (Lap, ◯) often yield slightly better results than orthogonal random features (ORF, ◯) presumably due to less stochasticity. Interestingly, we see that self-attention transfers the learned (pseudo-)equivariant self-attention structure to unseen graphs near perfectly.

Table 2: Results on PCQM4Mv2 large-scale graph regression benchmark. We report the Mean Absolute Error (MAE) on the validation set, and report MAE on the unavailable test set if possible.

| method | # parameters | valid MAE ↓ | test-dev MAE ↓ | asymptotics |
|---|---|---|---|---|
| *Message-passing GNNs* | | | | |
| GCN [27] | 2.0M | 0.1379 | 0.1398 | $\mathcal{O}(n+m)$ |
| GIN [27] | 3.8M | 0.1195 | 0.1218 | $\mathcal{O}(n+m)$ |
| GAT | 6.7M | 0.1302 | N/A | $\mathcal{O}(n+m)$ |
| GCN-VN [27] | 4.9M | 0.1153 | 0.1152 | $\mathcal{O}(n+m)$ |
| GIN-VN [27] | 6.7M | 0.1083 | 0.1084 | $\mathcal{O}(n+m)$ |
| GAT-VN | 6.7M | 0.1192 | N/A | $\mathcal{O}(n+m)$ |
| GAT-VN (large) | 55.2M | 0.1361 | N/A | $\mathcal{O}(n+m)$ |
| *Transformers with strong graph-specific modifications* | | | | |
| Graphormer [63] | 48.3M | 0.0864 | N/A | $\mathcal{O}(n^2)$ |
| EGT [29] | 89.3M | 0.0869 | 0.0872 | $\mathcal{O}(n^2)$ |
| GRPE [54] | 46.2M | 0.0890 | 0.0898 | $\mathcal{O}(n^2)$ |
| *Pure Transformers* | | | | |
| Transformer | 48.5M | 0.2340 | N/A | $\mathcal{O}((n+m)^2)$ |
| TokenGT (ORF) | 48.6M | 0.0962 | N/A | $\mathcal{O}((n+m)^2)$ |
| TokenGT (Lap) | 48.5M | 0.0910 | 0.0919 | $\mathcal{O}((n+m)^2)$ |
| TokenGT (Lap) + Performer | 48.5M | 0.0935 | N/A | $\mathcal{O}(n+m)$ |

Figure 3: Attention distance by head and network depth. Each dot shows mean attention distance in hops across graphs of a head at a layer. The visualization is inspired by Dosovitskiy et al. (2020) [18]. More images can be found in Appendix A.4.2.

Non-orthogonal random embeddings lead to inaccurate approximation (Random, ◯), highlighting the importance of orthogonality of node identifiers. The approximation is also inaccurate when we sample ORF $\mathbf{P}_t$ independently for each token $t$ (ORF (first-order), ◯) instead of using concatenated node identifiers $[\mathbf{P}_u, \mathbf{P}_v]$ for token $(u,v)$. This supports our argument in Section 2 that the incidence information implicitly provided via node identifiers plays a key role in approximation.

In Figure 2, we provide a visualization of self-attention maps learned under various node and type identifier choices. Additional results can be found in Appendix A.4.1.

## 5.2 Large-Scale Graph Learning

An exclusive characteristic of TokenGT is its minimal graph-specific inductive bias, which requires it to learn internal computation structure largely from data. As such models are commonly known to work well with large-scale data [68, 18], we explore the capability of TokenGT on the PCQM4Mv2 quantum chemistry regression dataset [27], one of the current largest with 3.7M molecular graphs.

For TokenGT, we use both node and type identifiers, and use main Transformer encoder configuration based on Graphormer [78] with 12 layers, 768 hidden dimension, and 32 attention heads. We try both ORF and Laplacian eigenvector as node identifiers, and denote corresponding models as **TokenGT (ORF)** and **TokenGT (Lap)** respectively. As an ablation, we also experiment with the same Transformer without node and type identifiers, which we denote as **Transformer**. Finally, we apply the kernel attention [11] that approximates the attention computation to linear cost (**TokenGT (Lap) + Performer**). We use AdamW optimizer with $(\beta_1, \beta_2) = (0.99, 0.999)$ and weight decay 0.1, and 60k learning rate warmup steps followed by linear decay over 1M iteration with batch size 1024. For fine-tuning, we use 1k warmup, 0.1M training steps, and cosine learning rate decay. We train the models on 8 RTX 3090 GPUs for 3 days. Further details are in Appendix A.3.3.

We provide the results in Table 2. A standard Transformer on the node and edge tokens cannot recognize graph structure and shows low performance (0.2340 valid MAE). Yet, the picture changes as soon as we augment the tokens with node and type identifiers. Notably, TokenGT (ORF) achieves 0.0962 MAE, which is already better than all GNN baselines. This is a somewhat surprising result, as both ORF and the Transformer are not aware of graph structures. This implies Transformer is strong enough to learn to interpret and reason over the incidence structure of tokens provided only implicitly by the node and type identifiers. By further switching to Laplacian eigenvectors that encode position on graphs [20], we observe a performance boost to 0.0910 MAE, competitive to Transformers with sophisticated graph-specific modifications (*e.g.*, shortest path-based spatial encoding [78]). While such methods inject graph structure into attention matrix via bias term and therefore strictly require $\mathcal{O}(n^2)$ cost, TokenGT enables adopting kernelization for pure self-attention [11], resulting in TokenGT (Lap) + Performer with the best performance among $\mathcal{O}(n + m)$ models (0.0935 MAE). Further discussion on the empirical performance of TokenGT can be found in Appendix A.5.

While our theory in Section 3 *guarantees* that TokenGT can reduce to an equivariant layer by learning fixed equivariant basis at each attention head, in practice, it can freely utilize multihead self-attention to learn less restricted and more useful computation structure from data. To analyze such a structure, we compute the attention distance across heads and network depth by averaging pairwise token distances on a graph weighted by their attention scores (Figure 3). This distance is analogous to the number of hops in message-passing. In both TokenGT (ORF) and TokenGT (Lap), in the lowest layers, some heads attend globally over the graph while others consistently have small receptive fields (acting like a local message-passing operator). In deeper layers, the attention distances increase, and most heads attend globally. Interestingly, this behavior is highly consistent with Vision Transformers on image patches [18], suggesting that hybrid architectures based on convolution to aid ViT [16, 80] might also work well for graphs. While TokenGT (ORF) shows relatively consistent attention distance over heads, TokenGT (Lap) shows higher variance, implying that it learns more diverse attention patterns. Judging from the higher performance of TokenGT (Lap), this suggests that the graph structure information of the Laplacian eigenvector facilitates learning useful and diverse attention structures, which calls for future exploration of better node identifiers based on graph PEs [38, 42].

## 6 Conclusion

We showed that Transformers directly applied to graphs can work well in both theory and practice. In the theoretical aspect, we proved that with appropriate token-wise embeddings, a Transformer on node and edge tokens is at least as expressive as $k$-IGN and $k$-WL test, making it more expressive than all message-passing GNNs. For such token-wise embeddings, we showed that a combination of simple orthonormal node identifiers and trainable type identifiers suffices, which we also verified with a synthetic experiment. In an experiment with PCQM4Mv2 large-scale dataset, we show that Tokenized Graph Transformer (TokenGT) performs significantly better than all GNNs and is competitive with Transformer variants with strong graph-specific architectural components [78, 29, 54].

While the results suggest a promising research direction, there are challenges to be addressed in future work. First, treating each node and edge as tokens requires $\mathcal{O}((n + m)^2)$ asymptotic cost due to the quadratic nature of self-attention. While we address this to some degree with kernelization and achieve $\mathcal{O}(n + m)$ cost, other types of efficient Transformers (*e.g.*, sparse) that can deliver better performance are left to be tested. Another issue is slightly lower performance compared to the state-of-the-art. Adopting Transformer engineering techniques from vision and language domains, such as data scaling [7, 18], deepening [70, 74], hybrid architectures [16, 80], and self-supervision [17, 7, 24], are promising. In the societal aspect, to prevent the potential risky behavior in, *e.g.*, decision making from graph-structured inputs, interpretability research regarding self-attention on graphs is desired.

We finish with interesting research directions that stem from our work. As our approach advocates viewing a graph as $(n + m)$ tokens [37], it opens up new paradigms of graph learning, including autoregressive decoding, in-context learning, prompting, and multimodal learning. Another interesting direction is to extend our theory and use self-attention to approximate equivariant basis for general discrete group actions, which might be a viable approach for *learning equivariance from data*.

**Acknowledgement**    This work was supported in part by Institute of Information & communications Technology Planning & Evaluation (IITP) (No. 2022-0-00926, 2022-0-00959, 2021-0-02068, and 2019-0-00075) and the National Research Foundation of Korea (NRF) (No. 2021R1C1C1012540) grants funded by the Korea government (MSIT).

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
