# A Appendix

## A.1 Proofs

### A.1.1 Extended Preliminary (Cont. from Section 3.1)

Before proceeding to the proofs, we first provide additional preliminary material that supplements Section 3.1. We begin by formally defining multihead self-attention and Transformer. Our definition is equivalent to Vaswani et al. (2017) [68], except we omit layer normalization for simplicity as in [81, 23, 34]. Specifically, a multihead self-attention layer MSA : $\mathbb{R}^{n \times d} \to \mathbb{R}^{n \times d}$ is defined as:

$$\boldsymbol{\alpha}^h = \text{softmax}\left(\mathbf{X}w_h^Q (\mathbf{X}w_h^K)^\top / \sqrt{d_H}\right), \tag{7}$$

$$\text{MSA}(\mathbf{X})_i = \sum_{h=1}^{H} \sum_{j=1}^{n} \boldsymbol{\alpha}_{ij}^h \mathbf{X}_j w_h^V w_h^O, \tag{8}$$

where $H$ is number of heads, $d_H$ is head size, and $w_h^Q, w_h^K \in \mathbb{R}^{d \times d_H}$, $w_h^V \in \mathbb{R}^{d \times d_v}$ $w_h^O \in \mathbb{R}^{d_v \times d}$. In our proofs, we use biases for query and key projections as in [81] but omit them here for brevity. With multihead self-attention, a Transformer layer $\mathcal{T} : \mathbb{R}^{n \times d} \to \mathbb{R}^{n \times d}$ is defined as:

$$\mathbf{H} = \mathbf{X} + \text{MSA}(\mathbf{X}), \tag{9}$$

$$\mathcal{T}(\mathbf{X}) = \mathbf{H} + \text{MLP}(\mathbf{H}), \tag{10}$$

where MSA : $\mathbb{R}^{n \times d} \to \mathbb{R}^{n \times d}$ is a multihead self-attention layer with $H$ heads of size $d_H$ and MLP : $\mathbb{R}^{n \times d} \to \mathbb{R}^{n \times d}$ is a tokenwise MLP with hidden dimension $d_F$.

We now provide the complete definition of invariant graph networks (IGNs) [47, 46] and maximally expressive equivariant linear layers [47] summarized in Section 3.1. We first recall Definition 1 and 2:

**Definition 1.** *An order-$k$ Invariant Graph Network ($k$-IGN) is a function $F_k : \mathbb{R}^{n^k \times d_0} \to \mathbb{R}$ written as the following:*

$$F_k = \text{MLP} \circ L_{k \to 0} \circ L_{k \to k}^{(T)} \circ \sigma \circ ... \circ \sigma \circ L_{k \to k}^{(1)}, \tag{1}$$

*where each $L_{k \to k}^{(t)}$ is equivariant linear layer [47] from $\mathbb{R}^{n^k \times d_{t-1}}$ to $\mathbb{R}^{n^k \times d_t}$, $\sigma$ is activation function, and $L_{k \to 0}$ is a invariant linear layer from $\mathbb{R}^{n^k \times d_T}$ to $\mathbb{R}$.*

**Definition 2.** *An equivariant linear layer is a function $L_{k \to l} : \mathbb{R}^{n^k \times d} \to \mathbb{R}^{n^l \times d'}$ written as follows for order-$k$ input $\mathbf{X} \in \mathbb{R}^{n^k \times d}$:*

$$L_{k \to l}(\mathbf{X})_{\mathbf{i}} = \sum_{\mu} \sum_{\mathbf{j}} \mathbf{B}_{\mathbf{i},\mathbf{j}}^{\mu} \mathbf{X}_{\mathbf{j}} w_\mu + \sum_{\lambda} \mathbf{C}_{\mathbf{i}}^{\lambda} b_\lambda, \tag{2}$$

*where $\mathbf{i} \in [n]^l, \mathbf{j} \in [n]^k$ are multi-indices, $w_\mu \in \mathbb{R}^{d \times d'}$, $b_\lambda \in \mathbb{R}^{d'}$ are weight and bias parameters, and $\mathbf{B}^\mu \in \mathbb{R}^{n^{l+k}}$ and $\mathbf{C}^\lambda \in \mathbb{R}^{n^l}$ are binary basis tensors corresponding to order-$(l+k)$ and order-$l$ equivalence classes $\mu$ and $\lambda$, respectively. Invariant linear layer is a special case of $L_{k \to l}$ with $l = 0$.*

We now define *equivalence classes* and *basis tensors* mentioned briefly in Definition 2. The equivalence classes are defined upon a specific *equivalence relation* $\sim$ on the index space of higher-order tensors as follows:

**Definition 3.** *An order-$l$ equivalence class $\gamma \in [n]^l/_\sim$ is an equivalence class of $[n]^l$ under the equivalence relation $\sim$, where the equivalence relation $\sim$ on multi-index space $[n]^l$ relates $\mathbf{i} \sim \mathbf{j}$ if and only if $(i_1, ..., i_l) = (\pi(j_1), ..., \pi(j_l))$ for some node permutation $\pi \in S_n$.*

We note that a multi-index $\mathbf{i}$ has the same permutation-invariant *equality pattern* to any $\mathbf{j}$ that satisfies $\mathbf{i} \sim \mathbf{j}$, *i.e.*, $\mathbf{i}_a = \mathbf{i}_b \Leftrightarrow \mathbf{j}_a = \mathbf{j}_b$ for all $a, b \in [k]$. Consequently, each equivalence class $\gamma$ in Definition 3 is a distinct set of all order-$l$ multi-indices having a specific equality pattern.

Now, for each equivalence class, we define the corresponding *basis tensor* as follows:

**Definition 4.** *An order-$l$ basis tensor $\mathbf{B}^\gamma \in \mathbb{R}^{n^l}$ corresponding to an order-$l$ equivalence class $\gamma$ is a binary tensor defined as follows:*

$$\mathbf{B}_{\mathbf{i}}^\gamma = \begin{cases} 1 & \mathbf{i} \in \gamma \\ 0 & otherwise \end{cases} \tag{11}$$

For a given $l$, it is known that there exist bell$(l)$ order-$l$ equivalence classes $\{\gamma_1, ..., \gamma_{\text{bell}(l)}\} = [n]^l/\sim$ regardless of $n$ [47]. This gives bell$(l)$ order-$l$ basis tensors $\mathbf{B}^{\gamma_1}, ..., \mathbf{B}^{\gamma_{\text{bell}(l)}}$ accordingly. Thus, an equivariant linear layer $L_{k\to l}$ in Definition 2 has bell$(l+k)$ weights and bell$(l)$ biases.

Let us consider the first-order equivariant layer $L_{1\to1}$ as an example. We have bell$(2) = 2$ second-order equivalence classes $\gamma_1$ and $\gamma_2$ for the weight, with $\gamma_1$ the set of all $(i_1, i_2)$ with $i_1 = i_2$ and $\gamma_2$ the set of all $(i_1, i_2)$ with $i_1 \neq i_2$. From Definition 4, their corresponding basis tensors are $\mathbf{B}^{\gamma_1} = \mathbf{I}$ and $\mathbf{B}^{\gamma_2} = \mathbf{1}\mathbf{1}^\top - \mathbf{I}$. Given a set of features $\mathbf{X} \in \mathbb{R}^{n \times d}$,

$$L_{1\to1}(\mathbf{X}) = \mathbf{I}\mathbf{X}w_1 + (\mathbf{1}\mathbf{1}^\top - \mathbf{I})\mathbf{X}w_2 + \mathbf{1}b^\top, \tag{12}$$

with two weights $w_1, w_2 \in \mathbb{R}^{d \times d'}$, and a single bias $b \in \mathbb{R}^{d'}$. For graphs ($k = l = 2$), we have bell$(4) = 15$ weights and bell$(2) = 2$ biases.

### A.1.2 Proof of Lemma 1 (Section 3.3)

To prove Lemma 1, we need to show that each basis tensor $\mathbf{B}^\mu$ (Eq. (11)) in weights of equivariant linear layers (Eq. (2)) can be approximated by the self-attention coefficient $\boldsymbol{\alpha}^h$ (Eq. (7)) to arbitrary precision up to normalization if its input is augmented by node and type identifiers (Section 3.3).

From Definition 4, each entry of basis tensor $\mathbf{B}^\mu_{\mathbf{i},\mathbf{j}}$ encodes whether $(\mathbf{i}, \mathbf{j}) \in \mu$ or not. Here, our key idea is to break down the inclusion test $(\mathbf{i}, \mathbf{j}) \in \mu$ into equivalent but simpler Boolean tests that can be implemented in self-attention (Eq. (8)) as dot product of $\mathbf{i}$-th query and $\mathbf{j}$-th key followed by softmax.

To achieve this, we show some supplementary Lemmas. We start with Lemma 2, which comes from Lemma 1 of Kim et al. (2021) [34] (we repeat their proof here for completeness).

**Lemma 2.** *For any order-$(l+k)$ equivalence class $\mu$, the set of all $\mathbf{i} \in [n]^l$ such that $(\mathbf{i}, \mathbf{j}) \in \mu$ for some $\mathbf{j} \in [n]^k$ forms an order-$l$ equivalence class. Likewise, the set of all $\mathbf{j}$ such that $(\mathbf{i}, \mathbf{j}) \in \mu$ for some $\mathbf{i}$ forms an order-$k$ equivalence class.*

*Proof.* We only prove for $\mathbf{i}$ as proof for $\mathbf{j}$ is analogous. For some $(\mathbf{i}^1, \mathbf{j}^1) \in \mu$, let us denote the equivalence class of $\mathbf{i}^1$ as $\gamma^l$ (*i.e.*, $\mathbf{i}^1 \in \gamma^l$). It is sufficient that we prove $\mathbf{i} \in \gamma^l \Leftrightarrow \exists \mathbf{j} : (\mathbf{i}, \mathbf{j}) \in \mu$.

($\Rightarrow$) For all $\mathbf{i} \in \gamma^l$, as $\mathbf{i}^1 \sim \mathbf{i}$, there exists some $\pi \in S_n$ such that $\mathbf{i} = \pi(\mathbf{i}^1)$ by definition. As $\pi$ acts on multi-indices entry-wise, we have $\pi(\mathbf{i}^1, \mathbf{j}^1) = (\mathbf{i}, \pi(\mathbf{j}^1))$. As $\pi(\mathbf{i}^1, \mathbf{j}^1) \sim (\mathbf{i}^1, \mathbf{j}^1)$ holds by definition, we have $(\mathbf{i}, \pi(\mathbf{j}^1)) \sim (\mathbf{i}^1, \mathbf{j}^1)$, and thus $(\mathbf{i}, \pi(\mathbf{j}^1)) \in \mu$. Therefore, for all $\mathbf{i} \in \gamma^l$, by setting $\mathbf{j} = \pi(\mathbf{j}^1)$ we can always obtain $(\mathbf{i}, \mathbf{j}) \in \mu$.

($\Leftarrow$) For all $(\mathbf{i}, \mathbf{j}) \in \mu$, as $(\mathbf{i}, \mathbf{j}) \sim (\mathbf{i}^1, \mathbf{j}^1)$, there exists some $\pi \in S_n$ such that $(\mathbf{i}, \mathbf{j}) = \pi(\mathbf{i}^1, \mathbf{j}^1)$. This gives $\mathbf{i} = \pi(\mathbf{i}^1)$ and $\mathbf{j} = \pi(\mathbf{j}^1)$, leading to $\mathbf{i} \sim \mathbf{i}^1$ and therefore $\mathbf{i} \in \gamma^l$. $\qquad\square$

Lemma 2 states that the equivalence classes $\gamma^l$ of $\mathbf{i}$ and $\gamma^k$ of $\mathbf{j}$ are identical for all $(\mathbf{i}, \mathbf{j}) \in \mu$. Based on this, we appropriately break down the test $(\mathbf{i}, \mathbf{j}) \in \mu$ into a combination of several simpler tests, in particular including $\mathbf{i} \in \gamma^l$ and $\mathbf{j} \in \gamma^k$:

**Lemma 3.** *For a given order-$(l+k)$ equivalence class $\mu$, let $\gamma^l$ and $\gamma^k$ be equivalence classes of some $\mathbf{i}^1 \in [n]^l, \mathbf{j}^1 \in [n]^k$ respectively that satisfies $(\mathbf{i}^1, \mathbf{j}^1) \in \mu$. Then, for any $\mathbf{i} \in [n]^l$ and $\mathbf{j} \in [n]^k$, $(\mathbf{i}, \mathbf{j}) \in \mu$ holds if and only if the following conditions both hold:*

   *1. $\mathbf{i} \in \gamma^l$ and $\mathbf{j} \in \gamma^k$*

   *2. $\mathbf{i}_a = \mathbf{j}_b \Leftrightarrow \mathbf{i}^2_a = \mathbf{j}^2_b$ for all $a \in [l]$, $b \in [k]$, and $(\mathbf{i}^2, \mathbf{j}^2) \in \mu$*

*Proof.* ($\Rightarrow$) If $(\mathbf{i}, \mathbf{j}) \in \mu$, from Lemma 2 it follows that $\mathbf{i} \in \gamma^l$ and $\mathbf{j} \in \gamma^k$. Also, as all $(\mathbf{i}^2, \mathbf{j}^2) \in \mu$ including $(\mathbf{i}, \mathbf{j})$ have the same equality pattern, it follows that for all $a \in [l]$, $b \in [k]$, and $(\mathbf{i}^2, \mathbf{j}^2) \in \mu$, if $\mathbf{i}^2_a = \mathbf{j}^2_b$ then $\mathbf{i}_a = \mathbf{j}_b$ and if $\mathbf{i}^2_a \neq \mathbf{j}^2_b$ then $\mathbf{i}_a \neq \mathbf{j}_b$.

($\Leftarrow$) We show that the conditions specify that the equivalence class of $(\mathbf{i}, \mathbf{j})$ is $\mu$.

For this, it is convenient to represent an order-$l$ equivalence class $\gamma$ as an equivalent *undirected graph* $\mathcal{G} = (\mathcal{V}, \mathcal{E})$ defined on vertex set $\mathcal{V} = \{v_1, ..., v_l\}$ where the vertices $v_a$ and $v_b$ are connected, *i.e.*, $(v_a, v_b) \in \mathcal{E}$ if and only if the equivalence class $\gamma$ specifies $\mathbf{i}_a = \mathbf{i}_b \forall \mathbf{i} \in \gamma$. Then, for some multi-index $\mathbf{i}' \in [n]^l$, the inclusion $\mathbf{i}' \in \gamma$ holds if and only if the equivalence class of $\mathbf{i}'$ is represented as $\mathcal{G}$.

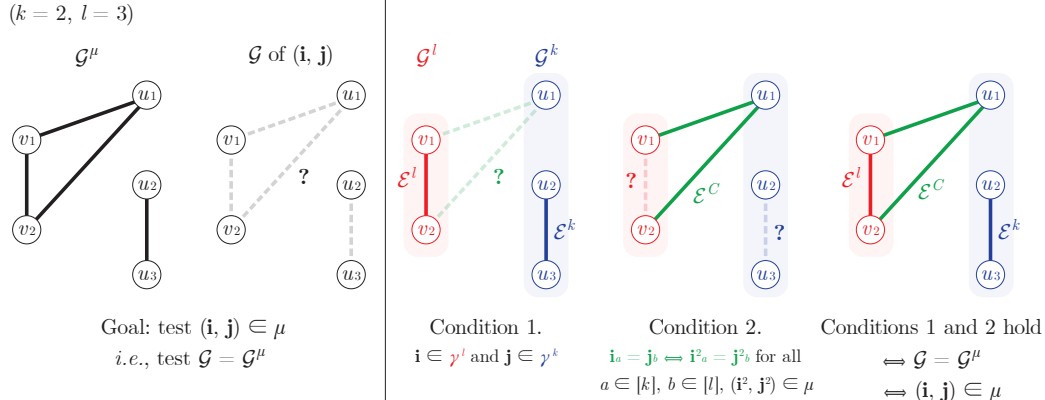

**Figure 4:** An exemplary illustration of testing $(\mathbf{i}, \mathbf{j}) \in \mu$ as a combination of simpler tests, based on equivalence classes $\mu$, $\gamma^l$, and $\gamma^k$ represented as graphs $\mathcal{G}^\mu$, $\mathcal{G}^l$, and $\mathcal{G}^k$, respectively.

Given this, let us represent the equivalence classes $\gamma^l$, $\gamma^k$, and $\mu$ as graphs $\mathcal{G}^l$, $\mathcal{G}^k$, and $\mathcal{G}^\mu$ respectively:

$$\mathcal{G}^l = (\mathcal{V}^l, \mathcal{E}^l) \text{ where } \mathcal{V}^l = \{v_1, ..., v_l\}, \tag{13}$$

$$\mathcal{G}^k = (\mathcal{V}^k, \mathcal{E}^k) \text{ where } \mathcal{V}^k = \{u_1, ..., u_k\}, \tag{14}$$

$$\mathcal{G}^\mu = (\mathcal{V}^\mu, \mathcal{E}^\mu) \text{ where } \mathcal{V}^\mu = \mathcal{V}^l \cup \mathcal{V}^k = \{v_1, ..., v_l, u_1, ..., u_k\}. \tag{15}$$

From the precondition that $\gamma^l$ and $\gamma^k$ are equivalence classes of $\mathbf{i}^1 \in [n]^l, \mathbf{j}^1 \in [n]^k$ that satisfies $(\mathbf{i}^1, \mathbf{j}^1) \in \mu$, we can see that $(v_a, v_b) \in \mathcal{E}^l \Leftrightarrow (v_a, v_b) \in \mathcal{E}^\mu$ and $(u_a, u_b) \in \mathcal{E}^k \Leftrightarrow (u_a, u_b) \in \mathcal{E}^\mu$. That is, if we consider $\mathcal{V}^l$ and $\mathcal{V}^k$ as a *graph cut* of $\mathcal{G}^\mu$ and write the cut-set (edges between $\mathcal{V}^l$ and $\mathcal{V}^k$) as $\mathcal{E}^C = \{(v_a, u_b) | (v_a, u_b) \in \mathcal{E}^\mu\}$, we obtain a partition $\{\mathcal{E}^l, \mathcal{E}^k, \mathcal{E}^C\}$ of the edge set $\mathcal{E}^\mu$.

We now move to the conditions.

Let us assume the first condition that $\mathbf{i} \in \gamma^l$ and $\mathbf{j} \in \gamma^k$, with the equivalence classes represented as $\mathcal{G}^l$ and $\mathcal{G}^k$, respectively. Now, let us consider the equivalence class of $(\mathbf{i}, \mathbf{j})$ represented by (unknown) graph $\mathcal{G} = (\mathcal{V}, \mathcal{E})$. Considering $\mathcal{V}^k$ and $\mathcal{V}^l$ as a graph cut of $\mathcal{G}$, we can see that $\mathcal{E}$ is partitioned as $\{\mathcal{E}^l, \mathcal{E}^k, \mathcal{E}^D\}$ where $\mathcal{E}^D$ is the cut-set (edges between $\mathcal{V}^l$ and $\mathcal{V}^k$).

Let us also assume the second condition $\mathbf{i}_a = \mathbf{j}_b \Leftrightarrow \mathbf{i}_a^2 = \mathbf{j}_b^2$ for all $a \in [l]$, $b \in [k]$, and $(\mathbf{i}^2, \mathbf{j}^2) \in \mu$. This directly implies that $e \in \mathcal{E}^C \Leftrightarrow e \in \mathcal{E}^D$, meaning that $\mathcal{E}^C = \mathcal{E}^D$. As a result, we see that $\mathcal{G}$ and $\mathcal{G}^\mu$ are identical graphs, and therefore the equivalence class of $(\mathbf{i}, \mathbf{j})$ is $\mu$ and $(\mathbf{i}, \mathbf{j}) \in \mu$ holds.

In Figure 4, we provide an exemplary illustration of testing $(\mathbf{i}, \mathbf{j}) \in \mu$ following the above discussion.
□

With Lemma 3, we have a decomposition of $(\mathbf{i}, \mathbf{j}) \in \mu$ into independent conditions on $\mathbf{i}$ and $\mathbf{j}$ combined with pairwise conditions between $\mathbf{i}$ and $\mathbf{j}$. In the following Definition 5 and Property 1, we encode these tests into a single *scoring function* that can be later implemented by self-attention.

**Definition 5.** *A scoring function $\delta(\mathbf{i}, \mathbf{j}; \mu, \epsilon)$ is a map that, given an order-$(l + k)$ equivalence class $\mu$ and $\epsilon > 0$, takes multi-indices $\mathbf{i} \in [n]^l, \mathbf{j} \in [n]^k$ and gives the following:*

$$\delta(\mathbf{i}, \mathbf{j}; \mu, \epsilon) = \mathbb{1}_{\mathbf{i} \in \gamma^l} + (1 - \epsilon) \mathbb{1}_{\mathbf{i} \notin \gamma^l} + \mathbb{1}_{\mathbf{j} \in \gamma^k} + (1 - \epsilon) \mathbb{1}_{\mathbf{j} \notin \gamma^k} + \sum_{a \in [l]} \sum_{b \in [k]} \text{sgn}(a, b) \mathbb{1}_{\mathbf{i}_a = \mathbf{j}_b}, \tag{16}$$

*where $\mathbb{1}$ is indicator, $\gamma^l$ and $\gamma^k$ are equivalence classes of $\mathbf{i}^1 \in [n]^l, \mathbf{j}^1 \in [n]^k$ such that $(\mathbf{i}^1, \mathbf{j}^1) \in \mu$, and the sign function $\text{sgn}(\cdot, \cdot)$ is defined as follows:*

$$\text{sgn}(a, b) = \begin{cases} +1 & \mathbf{i}_a^2 = \mathbf{j}_b^2 \forall (\mathbf{i}^2, \mathbf{j}^2) \in \mu \\ -1 & \mathbf{i}_a^2 \neq \mathbf{j}_b^2 \forall (\mathbf{i}^2, \mathbf{j}^2) \in \mu \end{cases}. \tag{17}$$

An important property of the scoring function $\delta(\mathbf{i}, \mathbf{j}; \mu)$ is that it gives the maximum possible value if and only if the input satisfies $(\mathbf{i}, \mathbf{j}) \in \mu$, as shown in the below Property 1.

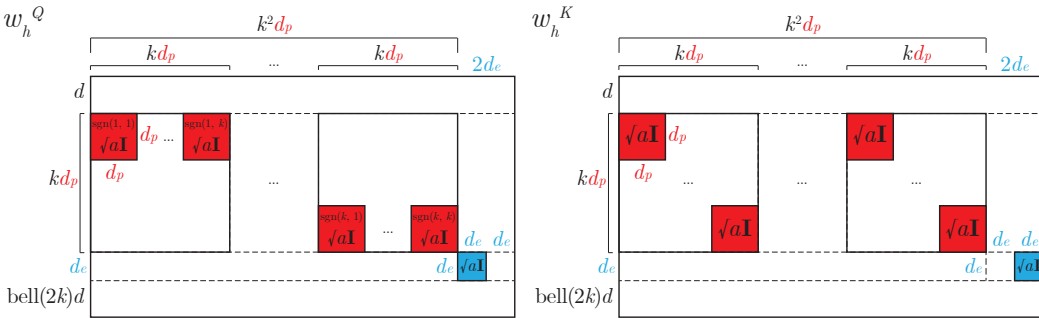

Figure 5: Query and key projection matrices $w_h^Q, w_h^K$ (Eq. (18), Eq. (19)). Uncolored cells are zeros.

**Property 1.** *For given order-$(l+k)$ equivalence class $\mu$ and positive real number $\epsilon > 0$, for any $\mathbf{i} \in [n]^l$ and $\mathbf{j} \in [n]^k$, $(\mathbf{i}, \mathbf{j}) \in \mu$ holds if and only if the scoring function $\delta(\mathbf{i}, \mathbf{j}; \mu, \epsilon)$ (Eq. (16)) outputs the maximum possible value.*

*Proof.* As shown in Lemma 3, $(\mathbf{i}, \mathbf{j}) \in \mu$ holds if and only if the following two conditions are met.

1. $\mathbf{i} \in \gamma^l$ and $\mathbf{j} \in \gamma^k$

2. $\mathbf{i}_a = \mathbf{j}_b \Leftrightarrow \mathbf{i}_a^2 = \mathbf{j}_b^2$ for all $a \in [l]$, $b \in [k]$, and $(\mathbf{i}^2, \mathbf{j}^2) \in \mu$

When both conditions are satisfied, in Eq. (16), we always have $\mathbb{1}_{\mathbf{i} \in \gamma^l} + (1 - \epsilon)\mathbb{1}_{\mathbf{i} \notin \gamma^l} = 1$ and $\mathbb{1}_{\mathbf{j} \in \gamma^k} + (1 - \epsilon)\mathbb{1}_{\mathbf{j} \notin \gamma^k} = 1$. We also have $\mathbb{1}_{\mathbf{i}_a = \mathbf{j}_b} = 1$ for $\mathrm{sgn}(a, b) = 1$ and $\mathbb{1}_{\mathbf{i}_a = \mathbf{j}_b} = 0$ for $\mathrm{sgn}(a, b) = -1$ for all $a \in [l], b \in [k]$. As a result, Eq. (16) gives a constant output for all $(\mathbf{i}, \mathbf{j}) \in \mu$.

On the other hand, if given $(\mathbf{i}, \mathbf{j})$ violates any of the conditions (thus $(\mathbf{i}, \mathbf{j}) \notin \mu$), we either have $\mathbb{1}_{\mathbf{i} \in \gamma^l} + (1 - \epsilon)\mathbb{1}_{\mathbf{i} \notin \gamma^l} = (1 - \epsilon)$, or $\mathbb{1}_{\mathbf{j} \in \gamma^k} + (1 - \epsilon)\mathbb{1}_{\mathbf{j} \notin \gamma^k} = (1 - \epsilon)$, or $\mathbb{1}_{\mathbf{i}_a = \mathbf{j}_b} = 0$ for $\mathrm{sgn}(a, b) = 1$ or $\mathbb{1}_{\mathbf{i}_a = \mathbf{j}_b} = 1$ for $\mathrm{sgn}(a, b) = -1$ for some $a \in [l], b \in [k]$. Any of these violations decrements the output of Eq. (16) by a positive (1 or $\epsilon$), resulting in a non-maximum output.

Thus, the scoring function $\delta(\mathbf{i}, \mathbf{j}; \mu, \epsilon)$ gives the maximum possible output if and only if $(\mathbf{i}, \mathbf{j}) \in \mu$. $\quad \square$

Now, we prove Lemma 1.

**Lemma 1.** *For all $\mathbf{X} \in \mathbb{R}^{n^k \times d}$ and their augmentation $\mathbf{X}^{in}$, self-attention coefficients $\boldsymbol{\alpha}^h$ (Eq. (3)) computed with $\mathbf{X}^{in} w^{in}$ can approximate any basis tensor $\mathbf{B}^\mu \in \mathbb{R}^{n^{2k}}$ of order-$k$ equivariant linear layer $L_{k \to k}$ (Definition 2) to arbitrary precision up to normalization.*

*Proof.* Let us first recall the node and type identifiers (Section 3.3) for order-$k$ tensors $\mathbf{X} \in \mathbb{R}^{n^k \times d}$. Node identifier $\mathbf{P} \in \mathbb{R}^{n \times d_p}$ is an orthonormal matrix with $n$ rows, and type identifier is a trainable matrix $\mathbf{E} \in \mathbb{R}^{\mathrm{bell}(k) \times d_e}$ with $\mathrm{bell}(k)$ rows $\mathbf{E}^{\gamma_1}, ..., \mathbf{E}^{\gamma_{\mathrm{bell}(k)}}$, each designated for an order-$k$ equivalence class $\gamma$. For each multi-index $\mathbf{i} = (i_1, ..., i_k) \in [n]^k$, we augment the corresponding input tensor entry as $[\mathbf{X_i}, \mathbf{P}_{i_1}, ..., \mathbf{P}_{i_k}, \mathbf{E}^{\gamma^{\mathbf{i}}}]$ where $\mathbf{i} \in \gamma^{\mathbf{i}}$, obtaining the augmented order-$k$ tensor $\mathbf{X}^{in} \in \mathbb{R}^{n^k \times (d + k d_p + d_e)}$. We use a trainable projection $w^{in} \in \mathbb{R}^{(d + k d_p + d_e) \times d_{\mathcal{T}}}$ to map them to a hidden dimension $d_{\mathcal{T}}$.

We now use self-attention on $\mathbf{X}^{in} w^{in}$ to perform an accurate approximation of the equivariant basis. Specifically, we use each self-attention matrix $\boldsymbol{\alpha}^h$ (Eq. (7)) to approximate each basis tensor $\mathbf{B}^{\mu_h}$ of $L_{k \to k}$ (Eq. (2)) to arbitrary precision up to normalization.

Let us take $d_{\mathcal{T}} = (d + k d_p + d_e) + \mathrm{bell}(2k)d$, putting $\mathrm{bell}(2k)d$ extra channels on top of channels of the augmented input $\mathbf{X}^{in}$. We now let $w^{in} = [\mathbf{I}, \mathbf{0}]$, where $\mathbf{I} \in \mathbb{R}^{(d + k d_p + d_e) \times (d + k d_p + d_e)}$ is an identity matrix and $\mathbf{0} \in \mathbb{R}^{(d + k d_p + d_e) \times (d_{\mathcal{T}} - (d + k d_p + d_e))}$ is a matrix filled with zeros. With this, $\mathbf{X}' = \mathbf{X}^{in} w^{in}$ simply contains $\mathbf{X}^{in}$ in the first $(d + k d_p + d_e)$ channels and zeros in the rest.

Now we pass $\mathbf{X}'$ to the self-attention layer in Eq. (7), where each self-attention matrix is given as $\boldsymbol{\alpha}^h = \mathrm{softmax}((\mathbf{X}' w_h^Q + b_h^Q)(\mathbf{X}' w_h^K + b_h^K)^\top / \sqrt{d_H})$. The key idea is to set query and key projection

parameters $w_h^Q, w_h^K \in \mathbb{R}^{d_T \times d_H}$ and $b_h^Q, b_h^K \in \mathbb{R}^{d_H}$ appropriately so that the self-attention matrix $\boldsymbol{\alpha}^h$ approximates a given basis tensor $\mathbf{B}^\mu$ corresponding to an order-$2k$ equivalence class $\mu$. Let $\gamma^Q$ and $\gamma^K$ be equivalence classes of some $\mathbf{i}^1, \mathbf{j}^1 \in [n]^k$ respectively that satisfy $(\mathbf{i}^1, \mathbf{j}^1) \in \mu$ (see Lemma 3). We set head dimension $d_H = k^2 d_p + 2d_e$ and set $w_h^Q, w_h^K, b_h^Q, b_h^K$ as follows:

$$
(w_h^Q)_{ij} = \begin{cases} \mathrm{sgn}(s,r)\sqrt{a}\mathbf{I}_{i-I,j-J} & \begin{cases} I < i \leq I + d_p & \text{for} \quad I = d + (s-1)d_p, \\ J < j \leq J + d_p & \text{for} \quad J = (s-1)kd_p + (r-1)d_p, \\ \text{for all } s, r \in [k] \end{cases} \\ \sqrt{a}\mathbf{I}_{i-I,j-J} & \begin{cases} I < i \leq I + d_e & \text{for} \quad I = d + kd_p, \\ J < j \leq J + d_e & \text{for} \quad J = k^2 d_p, \end{cases} \\ 0 & \text{otherwise} \end{cases}
\tag{18}
$$

$$
(w_h^K)_{ij} = \begin{cases} \sqrt{a}\mathbf{I}_{i-I,j-J} & \begin{cases} I < i \leq I + d_p & \text{for} \quad I = d + (r-1)d_p, \\ J < j \leq J + d_p & \text{for} \quad J = (s-1)kd_p + (r-1)d_p, \\ \text{for all } s, r \in [k] \end{cases} \\ \sqrt{a}\mathbf{I}_{i-I,j-J} & \begin{cases} I < i \leq I + d_e & \text{for} \quad I = d + kd_p, \\ J < j \leq J + d_e & \text{for} \quad J = k^2 d_p + d_e, \end{cases} \\ 0 & \text{otherwise} \end{cases}
\tag{19}
$$

$$
(b_h^Q)_j = \begin{cases} \sqrt{a}\mathbf{E}_{j-J}^{\gamma^K} & J < j \leq J + d_e \quad \text{for} \quad J = k^2 d_p \\ 0 & \text{otherwise} \end{cases}
\tag{20}
$$

$$
(b_h^K)_j = \begin{cases} \sqrt{a}\mathbf{E}_{j-J}^{\gamma^Q} & J < j \leq J + d_e \quad \text{for} \quad J = k^2 d_p + d_e \\ 0 & \text{otherwise} \end{cases}
\tag{21}
$$

where $a > 0$ is a positive real, $\mathbf{I}$ is an identity matrix, and $\mathrm{sgn}(\cdot, \cdot)$ is the sign function defined in Eq. (17) (Definition 5). In Figure 5 we provide an illustration of the query and key weights $w_h^Q, w_h^K$.

With the parameters, $\mathbf{i}$-th query and $\mathbf{j}$-th key entries are computed as follows:

$$
\mathbf{X}_\mathbf{i}' w_h^Q + b_h^Q = \sqrt{a}[[\mathrm{sgn}(1,1)\mathbf{P}_{i_1}, ..., \mathrm{sgn}(1,k)\mathbf{P}_{i_1}], ..., [\mathrm{sgn}(k,1)\mathbf{P}_{i_k}, ..., \mathrm{sgn}(k,k)\mathbf{P}_{i_k}], \mathbf{E}^{\gamma^\mathbf{i}}, \mathbf{E}^{\gamma^K}],
\tag{22}
$$

$$
\mathbf{X}_\mathbf{j}' w_h^K + b_h^K = \sqrt{a}[\overbrace{[\mathbf{P}_{j_1}, ..., \mathbf{P}_{j_k}], ..., [\mathbf{P}_{j_1}, ..., \mathbf{P}_{j_k}]}^{k \text{ repeats}}, \mathbf{E}^{\gamma^Q}, \mathbf{E}^{\gamma^\mathbf{j}}].
\tag{23}
$$

Then, scaled pairwise dot product of query and key is given as follows:

$$
\frac{(\mathbf{X}_\mathbf{i}' w_h^Q + b_h^Q)^\top (\mathbf{X}_\mathbf{j}' w_h^K + b_h^K)}{\sqrt{d_H}} = \frac{a}{\sqrt{d_H}} \left( (\mathbf{E}^{\gamma^\mathbf{i}})^\top \mathbf{E}^{\gamma^Q} + (\mathbf{E}^{\gamma^\mathbf{j}})^\top \mathbf{E}^{\gamma^K} + \sum_{a \in [k]} \sum_{b \in [k]} \mathrm{sgn}(a,b) \mathbf{P}_{i_a}^\top \mathbf{P}_{j_b} \right).
\tag{24}
$$

We refer to the scaled dot product in Eq. (24) as the *unnormalized* attention coefficient $\tilde{\boldsymbol{\alpha}}_{\mathbf{i},\mathbf{j}}^h$.

We now let the type identifiers $\mathbf{E}^{\gamma_1}, ..., \mathbf{E}^{\gamma_{\mathrm{bell}(k)}}$ be radially equispaced unit vectors on any two-dimensional subspace (Figure 6). This guarantees that any pair of type identifiers $\mathbf{E}^{\gamma_1}, \mathbf{E}^{\gamma_2}$ with $\gamma_1 \neq \gamma_2$ have dot product $(\mathbf{E}^{\gamma_1})^\top \mathbf{E}^{\gamma_2} \leq \cos(2\pi/\mathrm{bell}(k))$. By setting $\epsilon = 1 - \cos(2\pi/\mathrm{bell}(k)) > 0$, this can be equivalently written as $(\mathbf{E}^{\gamma_1})^\top \mathbf{E}^{\gamma_2} \leq 1 - \epsilon$. We additionally note that $(\mathbf{E}^{\gamma^\mathbf{i}})^\top \mathbf{E}^{\gamma^Q} = 1$ if and only if $\mathbf{i} \in \gamma^Q$ because $\gamma^\mathbf{i} = \gamma^Q \Leftrightarrow \mathbf{i} \in \gamma^Q$.

Combining the above, Eq. (24), and Eq. (16), we have the following:

$$
\tilde{\boldsymbol{\alpha}}_{\mathbf{i},\mathbf{j}}^h = \frac{a}{\sqrt{d_H}} \delta(\mathbf{i}, \mathbf{j}; \mu, \epsilon) \quad \text{if } (\mathbf{i}, \mathbf{j}) \in \mu,
\tag{25}
$$

$$
\tilde{\boldsymbol{\alpha}}_{\mathbf{i},\mathbf{j}}^h \leq \frac{a}{\sqrt{d_H}} \delta(\mathbf{i}, \mathbf{j}; \mu, \epsilon) \quad \text{otherwise},
\tag{26}
$$

where $\epsilon = 1 - \cos(2\pi/\mathrm{bell}(k))$ and $\delta(\mathbf{i}, \mathbf{j}; \mu, \epsilon)$ is the scoring function in Eq. (16) (Definition 5).

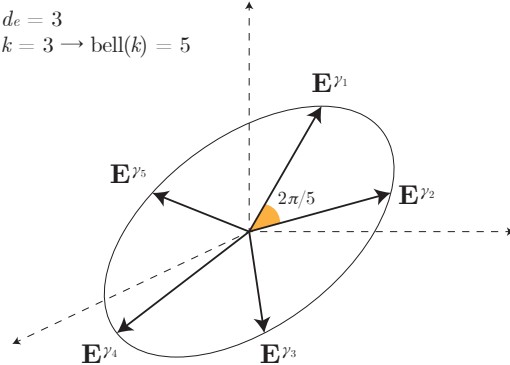

$$d_e = 3$$
$$k = 3 \rightarrow \text{bell}(k) = 5$$

$\mathbf{E}^{\gamma_1}$  $\mathbf{E}^{\gamma_5}$  $2\pi/5$  $\mathbf{E}^{\gamma_2}$  $\mathbf{E}^{\gamma_4}$  $\mathbf{E}^{\gamma_3}$

Figure 6: $k = 3$ case example of $\text{bell}(k) = 5$ type identifiers embedded in $d_e = 3$ dimensional space.

For a given query index $\mathbf{i}$, let us assume there exists at least one key index $\mathbf{j}$ such that $(\mathbf{i}, \mathbf{j}) \in \mu$ [3]. From Property 1 and Eq. (25), all keys $\mathbf{j}$ that give $(\mathbf{i}, \mathbf{j}) \in \mu$ hold the same maximum value $\tilde{\alpha}_{\mathbf{i},\mathbf{j}}^h = \frac{a}{\sqrt{d_H}}\delta(\mathbf{i}, \mathbf{j}; \mu, \epsilon)$, and any $(\mathbf{i}, \mathbf{j}) \notin \mu$ gives a value smaller at least by $\min(1, \epsilon) > 0$. Then, in softmax normalization, we send $a \rightarrow \infty$ by scaling up the query and key projection parameters. This pushes softmax arbitrarily close to the hardmax operator, leaving only the maximal entries leading to the following:

$$\alpha_{\mathbf{i},\mathbf{j}}^h = \frac{\exp(\tilde{\alpha}_{\mathbf{i},\mathbf{j}}^h)}{\sum_{\mathbf{j}} \exp(\tilde{\alpha}_{\mathbf{i},\mathbf{j}}^h)} \rightarrow \frac{\mathbb{1}_{(\mathbf{i},\mathbf{j}\in\mu)}}{\sum_{\mathbf{j}} \mathbb{1}_{(\mathbf{i},\mathbf{j}\in\mu)}} = \frac{\mathbf{B}_{\mathbf{i},\mathbf{j}}^\mu}{\sum_{\mathbf{j}} \mathbf{B}_{\mathbf{i},\mathbf{j}}^\mu} \text{ as } a \rightarrow \infty. \tag{27}$$

Thus, as shown in Eq. (27), the attention coefficient $\alpha^h$ can arbitrarily accurately approximate the normalized basis tensor $\mathbf{B}^\mu$ for given equivalence class $\mu$. $\qquad\square$

### A.1.3 Proof of Theorem 1 (Section 3.3)

**Theorem 1.** *For all* $\mathbf{X} \in \mathbb{R}^{n^k \times d}$ *and their augmentation* $\mathbf{X}^{in}$*, a Transformer layer with* $\text{bell}(2k)$ *self-attention heads that operates on* $\mathbf{X}^{in}w^{in}$ *can approximate an order-$k$ equivariant linear layer* $L_{k \rightarrow k}(\mathbf{X})$ *(Definition 2) to arbitrary precision.*

*Proof.* We continue from the proof of Lemma 1 and assume that each attention matrix $\alpha^1, ..., \alpha^{\text{bell}(2k)}$ in Eq. (7) head-wise approximates each normalized basis tensor $\mathbf{B}^{\mu_1}, ..., \mathbf{B}^{\mu_{\text{bell}(2k)}}$ respectively, *i.e.*, $\alpha_{\mathbf{i},\mathbf{j}}^h = \mathbf{B}_{\mathbf{i},\mathbf{j}}^{\mu_h} / \sum_{\mathbf{j}} \mathbf{B}_{\mathbf{i},\mathbf{j}}^{\mu_h}$.[4]

Then, in Eq. (8) we use $d_v = d$ and set $w_h^V \in \mathbb{R}^{d_T \times d}$ to $w_h^V = [\mathbf{I}; \mathbf{0}]$, where $\mathbf{I} \in \mathbb{R}^{d \times d}$ is an identity matrix and $\mathbf{0} \in \mathbb{R}^{(d_T - d) \times d}$ is a matrix filled with zeros. With this, the value projection of each $\mathbf{i}$-th entry simply gives the original input features, $\mathbf{X}_{\mathbf{i}}'w_h^V = \mathbf{X}_{\mathbf{i}}$.

Then, we set output projections $w_h^O \in \mathbb{R}^{d \times d_T}$ as follows:

$$(w_h^O)_{ij} = \begin{cases} (w_{\mu_h})_{i,j-J} & J < j \leq J + d \text{ for } J = (d + kd_p + d_e) + (h-1)d \\ 0 & \text{otherwise} \end{cases}, \tag{28}$$

where $w_{\mu_1}, ..., w_{\mu_{\text{bell}(2k)}} \in \mathbb{R}^{d \times d}$ are weight matrices of the given equivariant linear layer $L_{k \rightarrow k}$ in Eq. (2) (Definition 2), each corresponding to equivalence classes $\mu_1, ..., \mu_{\text{bell}(2k)}$.

Then, output projection applied after value projection of each $\mathbf{i}$-th input entry gives the following:

$$\mathbf{X}_{\mathbf{i}}'w_h^V w_h^O = \mathbf{X}_{\mathbf{i}}w_h^O = [\mathbf{0}, \mathbf{0}_L, \mathbf{X}_{\mathbf{i}}w_{\mu_h}, \mathbf{0}_R], \tag{29}$$

where $\mathbf{0} \in \mathbb{R}^{(d + kd_p + d_e)}, \mathbf{0}_L \in \mathbb{R}^{(h-1)d}, \mathbf{0}_R \in \mathbb{R}^{d_T - (d + kd_p + d_e) - hd}$ are zero vectors.

---

[3] If such key index $\mathbf{j}$ does not exist, corresponding basis tensor entries are $\mathbf{B}_{\mathbf{i},\mathbf{j}}^\mu = 0 \forall \mathbf{j}$, and approximation target cannot be defined as normalizing denominator $\sum_{\mathbf{j}} \mathbf{B}_{\mathbf{i},\mathbf{j}}^\mu$ is 0. Thus we do not approximate for such $\mathbf{i}$, let attention row $\alpha_{\mathbf{i},\cdot}$ have any finite values, and later silence their attention output by multiplying zero at MLP.

[4] we handle the case $\sum_{\mathbf{j}} \mathbf{B}_{\mathbf{i},\mathbf{j}}^{\mu_h} = 0$ later separately as mentioned in footnote 3.

Based on the results, we compute the MSA with skip connection $\mathbf{H} = \mathbf{X}' + \text{MSA}(\mathbf{X}')$ (Eq. (9)):

$$\mathbf{H_i} = \mathbf{X_i'} + \text{MSA}(\mathbf{X}')_\mathbf{i} \tag{30}$$

$$= \left[\mathbf{X_i}, \mathbf{P}_{i_1}, ..., \mathbf{P}_{i_k}, \mathbf{E}^{\gamma^\mathbf{i}}, \mathbf{0}_1\right]$$

$$+ \left[\mathbf{0}_2, \sum_\mathbf{j} \frac{\mathbf{B}_{\mathbf{i,j}}^{\mu_1}}{\sum_\mathbf{j} \mathbf{B}_{\mathbf{i,j}}^{\mu_1}} \mathbf{X_j} w_{\mu_1}, ..., \sum_\mathbf{j} \frac{\mathbf{B}_{\mathbf{i,j}}^{\mu_{\text{bell}(2k)}}}{\sum_\mathbf{j} \mathbf{B}_{\mathbf{i,j}}^{\mu_{\text{bell}(2k)}}} \mathbf{X_j} w_{\mu_{\text{bell}(2k)}}\right] \tag{31}$$

$$= \left[\mathbf{X_i}, \mathbf{P}_{i_1}, ..., \mathbf{P}_{i_k}, \mathbf{E}^{\gamma^\mathbf{i}}, \sum_\mathbf{j} \frac{\mathbf{B}_{\mathbf{i,j}}^{\mu_1}}{\sum_\mathbf{j} \mathbf{B}_{\mathbf{i,j}}^{\mu_1}} \mathbf{X_j} w_{\mu_1}, ..., \sum_\mathbf{j} \frac{\mathbf{B}_{\mathbf{i,j}}^{\mu_{\text{bell}(2k)}}}{\sum_\mathbf{j} \mathbf{B}_{\mathbf{i,j}}^{\mu_{\text{bell}(2k)}}} \mathbf{X_j} w_{\mu_{\text{bell}(2k)}}\right], \tag{32}$$

where $\mathbf{0}_1 \in \mathbb{R}^{d_\mathcal{T} - (d + kd_p + d_e)}$, $\mathbf{0}_2 \in \mathbb{R}^{(d + kd_p + d_e)}$ are zero vectors.

We use feedforward MLP (Eq. (10)) to denormalize and combine the result. Specifically, we make the elementwise MLP approximate following $f : \mathbb{R}^{d_\mathcal{T}} \to \mathbb{R}^{d_\mathcal{T}}$ based on universal approximation [23, 26]:

$$f(\mathbf{H_i})_j = \begin{cases} -\mathbf{H}_{\mathbf{i},j} + \sum_{h \in [\text{bell}(2k)]} g(\mathbf{H_i})_h \mathbf{H}_{\mathbf{i},j+J} + b(\mathbf{H_i})_j & j \leq d \\ 0 & d < j \leq (d + kd_p + d_e) \\ -\mathbf{H}_{\mathbf{i},j} & j > (d + kd_p + d_e) \end{cases}, \tag{33}$$

$$g(\mathbf{H_i})_h = \sum_\mathbf{j} \mathbf{B}_{\mathbf{i,j}}^{\mu_h}, \tag{34}$$

$$b(\mathbf{H_i})_j = (b_{\gamma^\mathbf{i}})_j = (\sum_\gamma \mathbf{C}_\mathbf{i}^\gamma b_\gamma)_j, \tag{35}$$

where $J = (d + kd_p + d_e) + (h - 1)d$, and $b_{\gamma_1}, ..., b_{\gamma_{\text{bell}(k)}}$ are biases of the given equivariant linear layer $L_{k \to k}$ with corresponding basis tensors $\mathbf{C}^{\gamma_1}, ..., \mathbf{C}^{\gamma_{\text{bell}(k)}}$ (Eq. (2)).

Within the function $f$, the auxiliary function $g : \mathbb{R}^{d_\mathcal{T}} \to \mathbb{R}^{\text{bell}(2k)}$ computes head-wise attention denormalization factor[5] and $b : \mathbb{R}^{d_\mathcal{T}} \to \mathbb{R}^d$ computes bias. As $n$ and $k$ are fixed constants, the outputs $g(\mathbf{H_i})$ and $b(\mathbf{H_i})$ only depend on the equivalence class $\gamma^\mathbf{i}$ of $\mathbf{i}$. We note that the functions $g$ and $b$ can deduce the equivalence class from the input $\mathbf{H_i}$, by extracting the type identifier $\mathbf{E}^{\gamma^\mathbf{i}} = \mathbf{H_i}^\top [\mathbf{0}_3, \mathbf{I}, \mathbf{0}_4]$ with $\mathbf{I} \in \mathbb{R}^{d_e \times d_e}$ an identity matrix and $\mathbf{0}_3 \in \mathbb{R}^{d + kd_p}$, $\mathbf{0}_4 \in \mathbb{R}^{\text{bell}(2k)d}$ zero matrices.

Based on the results, we compute the feedforward MLP with skip connection $\mathcal{T}(\mathbf{X}') = \mathbf{H} + \text{MLP}(\mathbf{H})$ (Eq. (10)), which is the output of Transformer layer $\mathcal{T}$:

$$\mathcal{T}(\mathbf{X}')_\mathbf{i} = \mathbf{H_i} + \text{MLP}(\mathbf{H})_\mathbf{i} \tag{36}$$

$$= \mathbf{H_i} + f(\mathbf{H_i}) \tag{37}$$

$$= \left[\mathbf{X_i}, \mathbf{P}_{i_1}, ..., \mathbf{P}_{i_k}, \mathbf{E}^{\gamma^\mathbf{i}}, \mathbf{S}_\mathbf{i}^1, ..., \mathbf{S}_\mathbf{i}^{\text{bell}(2k)}\right]$$

$$+ \left[-\mathbf{X_i} + \sum_{h \in [\text{bell}(2k)]} \sum_\mathbf{j} \mathbf{B}_{\mathbf{i,j}}^{\mu_h} \mathbf{X_j} w_{\mu_h} + \sum_\gamma \mathbf{C}_\mathbf{i}^\gamma b_\gamma, \mathbf{0}_5, -\mathbf{S}_\mathbf{i}^1, ..., -\mathbf{S}_\mathbf{i}^{\text{bell}(2k)}\right], \tag{38}$$

$$= \left[\sum_{h \in [\text{bell}(2k)]} \sum_\mathbf{j} \mathbf{B}_{\mathbf{i,j}}^{\mu_h} \mathbf{X_j} w_{\mu_h} + \sum_\gamma \mathbf{C}_\mathbf{i}^\gamma b_\gamma, \mathbf{P}_{i_1}, ..., \mathbf{P}_{i_k}, \mathbf{E}^{\gamma^\mathbf{i}}, \mathbf{0}_6\right], \tag{39}$$

where we write $\mathbf{S}_\mathbf{i}^h = \sum_\mathbf{j} \frac{\mathbf{B}_{\mathbf{i,j}}^{\mu_h}}{\sum_\mathbf{j} \mathbf{B}_{\mathbf{i,j}}^{\mu_h}} \mathbf{X_j} w_{\mu_h}$ and $\mathbf{0}_5 \in \mathbb{R}^{kd_p + d_e}$, $\mathbf{0}_6 \in \mathbb{R}^{(d_\mathcal{T} - (d + kd_p + d_e))}$ are zeros.

In Eq. (39), note that the Transformer layer $\mathcal{T}(\mathbf{X}')_\mathbf{i}$ only updates the first $d$ channels of $\mathbf{X_i'}$ from $\mathbf{X_i}$ to $\sum_\mu \sum_\mathbf{j} \mathbf{B}_{\mathbf{i,j}}^\mu \mathbf{X_j} w_\mu + \sum_\gamma \mathbf{C}_\mathbf{i}^\gamma b_\gamma$. Therefore, with a simple projection $w^{out} = [\mathbf{I}; \mathbf{0}] \in \mathbb{R}^{d_\mathcal{T} \times d}$ where $\mathbf{I} \in \mathbb{R}^{d \times d}$ is an identity matrix and $\mathbf{0} \in \mathbb{R}^{(d_\mathcal{T} - d) \times d}$ is a matrix filled with zeros, we can select the first $d$ channels of the output and finally obtain $\mathcal{T}(\mathbf{X}')w^{out} = L_{k \to k}(\mathbf{X})$.

In conclusion, a Transformer layer with $\text{bell}(2k)$ self-attention heads that operates on augmented $\mathbf{X}'$ can approximate any given $L_{k \to k}(\mathbf{X})$ to arbitrary precision. $\qquad \square$

---

[5]Note that the $g(\mathbf{H_i})_h$ gives 0 for all $\mathbf{i}$ that $\sum_\mathbf{j} \mathbf{B}_{\mathbf{i,j}}^{\mu_h} = 0$, which automatically handles the corner case as discussed at footnote 3 and footnote 4.

### A.1.4 Proof of Theorem 2 (Section 3.3)

**Theorem 2.** *For all $\mathbf{X} \in \mathbb{R}^{n^k \times d}$ and their augmentation $\mathbf{X}^{in}$, a Transformer composed of $T$ layers that operates on $\mathbf{X}^{in} w^{in}$ followed by sum-pooling and MLP can approximate an $k$-IGN $F_k(\mathbf{X})$ (Definition 1) to arbitrary precision.*

*Proof.* We continue from the proof of Theorem 1, and assume that each Transformer layer $\mathcal{T}$ can approximate a given $L_{k \to k}$ by only updating the first $d$ channels.

Then, based on Theorem 1 we assume the following for each $t < T$:

$$\mathcal{T}^{(t)}(\mathbf{X}')_{\mathbf{i}} = \left[ \sigma(L_{k \to k}^{(t)}(\mathbf{X}))_{\mathbf{i}}, \mathbf{P}_{i_1}, ..., \mathbf{P}_{i_k}, \mathbf{E}^{\gamma^{\mathbf{i}}}, \mathbf{0}_6 \right] \tag{40}$$

where $\mathbf{X}'_{\mathbf{i}} = [\mathbf{X}_{\mathbf{i}}, \mathbf{P}_{i_1}, ..., \mathbf{P}_{i_k}, \mathbf{E}^{\gamma^{\mathbf{i}}}, \mathbf{0}_6]$. While Theorem 1 gives $L_{k \to k}^{(t)}(\mathbf{X})$ in the first $d$ channels, we add elementwise activation $\sigma(\cdot)$ by absorbing it into the elementwise MLP in Eq. (33). Then, leveraging the property that each Transformer layer $\mathcal{T}^{(t)}$ only updates the first $d$ channels, we stack $T - 1$ Transformer layers $\mathcal{T}^{(1)}, ..., \mathcal{T}^{(T-1)}$ and obtain the following:

$$\mathcal{T}^{(T-1)} \circ ... \circ \mathcal{T}^{(1)}(\mathbf{X}')_{\mathbf{i}} = \left[ \sigma \circ L_{k \to k}^{(T-1)} \circ \sigma \circ ... \circ \sigma \circ L_{k \to k}^{(1)}(\mathbf{X})_{\mathbf{i}}, \mathbf{P}_{i_1}, ..., \mathbf{P}_{i_k}, \mathbf{E}^{\gamma^{\mathbf{i}}}, \mathbf{0}_6 \right]. \tag{41}$$

For the last layer $\mathcal{T}^{(T)}$, we follow the procedure in the proof of Theorem 1 to approximate $L_{k \to k}^{(T)}$, but slightly tweak Eq. (33) so that elementwise MLP copies each output entry $L_{k \to k}(\mathbf{X})_{\mathbf{i}}^{(T)}$ in appropriate reserved channels. Specifically, we let the elementwise MLP approximate following $f'$:

$$f'(\mathbf{H}_{\mathbf{i}})_j = \begin{cases} -\mathbf{H}_{\mathbf{i},j} & j \le D \\ -\mathbf{H}_{\mathbf{i},j} + \mathbf{C}_{\mathbf{i}}^{\gamma_a} \mathbf{F}_{\mathbf{i},j-(D+(a-1)d)} & D + (a-1)d < j \le D + ad \text{ for all } a \in [\text{bell}(k)] \\ -\mathbf{H}_{\mathbf{i},j} & D + \text{bell}(k)d < j \end{cases}, \tag{42}$$

where $D = (d + kd_p + d_e)$ and we abbreviate $\mathbf{F}_{\mathbf{i},j} = \sum_{h \in [\text{bell}(2k)]} g(\mathbf{H}_{\mathbf{i}})_h \mathbf{H}_{\mathbf{i},j+J} + b(\mathbf{H}_{\mathbf{i}})_j$ with $J, g, b$ defined as same as in Eq. (33). Recall that $\mathbf{C}_{\mathbf{i}}^{\gamma_a} = 1$ if and only if $\mathbf{i} \in \gamma_a$. Therefore, with Eq. (42), we are simply duplicating each output entry $\mathbf{F}_{\mathbf{i}} = L_{k \to k}^{(T)}(\mathbf{X})_{\mathbf{i}}$ to spare channel indices reserved for the equivalence class of $\mathbf{i}$ ($\gamma_a$ that $\mathbf{i} \in \gamma_a$).

With the choice of $\mathcal{T}^{(T)}$, the layer output $\mathcal{T}^{(T)}(\mathbf{X}') = \mathbf{H} + \text{MLP}(\mathbf{H})$ (Eq. (10)) is computed as:

$$\mathcal{T}^{(T)}(\mathbf{X}')_{\mathbf{i}} = \mathbf{H}_{\mathbf{i}} + \text{MLP}(\mathbf{H})_{\mathbf{i}} \tag{43}$$

$$= \mathbf{H}_{\mathbf{i}} + f'(\mathbf{H}_{\mathbf{i}}) \tag{44}$$

$$= \left[ \mathbf{0}_7, \mathbf{C}_{\mathbf{i}}^{\gamma_1} L_{k \to k}^{(T)}(\mathbf{X})_{\mathbf{i}}, ..., \mathbf{C}_{\mathbf{i}}^{\gamma_{\text{bell}(k)}} L_{k \to k}^{(T)}(\mathbf{X})_{\mathbf{i}}, \mathbf{0}_8 \right], \tag{45}$$

where $\mathbf{0}_7 \in \mathbb{R}^{(d+kd_p+d_e)}, \mathbf{0}_8 \in \mathbb{R}^{d_{\mathcal{T}}-(d+kd_p+d_e)-\text{bell}(k)d}$ are zero vectors.

Then, by applying $\mathcal{T}^{(T)}$ (Eq. (45)) on top of $\mathcal{T}^{(T-1)} \circ ... \circ \mathcal{T}^{(1)}$ (Eq. (41)), we obtain the following:

$$\mathcal{T}^{(T)} \circ ... \circ \mathcal{T}^{(1)}(\mathbf{X}')_{\mathbf{i}} = \left[ \mathbf{0}_7, \mathbf{C}_{\mathbf{i}}^{\gamma_1} \mathbf{Y}_{\mathbf{i}}, ..., \mathbf{C}_{\mathbf{i}}^{\gamma_{\text{bell}(k)}} \mathbf{Y}_{\mathbf{i}}, \mathbf{0}_8 \right]. \tag{46}$$

where we abbreviate $\mathbf{Y} = L_{k \to k}^{(T)} \circ \sigma \circ ... \circ \sigma \circ L_{k \to k}^{(1)}(\mathbf{X})$.

The remaining step is to utilize MLP $\circ$ sumpool to approximate $\text{MLP}_k \circ L_{k \to 0}$ that tops $F_k$. By sum-pooling over all indices $\mathbf{i}$, we obtain the following:

$$\text{sumpool} \circ \mathcal{T}^{(T)} \circ ... \circ \mathcal{T}^{(1)}(\mathbf{X}') = \left[ \mathbf{0}_7, \sum_{\mathbf{i}} \mathbf{C}_{\mathbf{i}}^{\gamma_1} \mathbf{Y}_{\mathbf{i}}, ..., \sum_{\mathbf{i}} \mathbf{C}_{\mathbf{i}}^{\gamma_{\text{bell}(k)}} \mathbf{Y}_{\mathbf{i}}, \mathbf{0}_8 \right]. \tag{47}$$

Now, we let the final MLP approximate the following function $f'' : \mathbb{R}^{d_{\mathcal{T}}} \to \mathbb{R}^d$:

$$f''(\mathbf{X}) = \text{MLP}_k \left( \sum_{a \in [\text{bell}(k)]} \mathbf{X}^a w_{\mu_a} + b_f \right) \text{ where } \mathbf{X}_j^a = \mathbf{X}_{D+(a-1)d+j} \text{ for } j \in [d], \tag{48}$$

where $w_{\mu_1}, ..., w_{\mu_{\text{bell}(k)}} \in \mathbb{R}^{d \times d}$ and $b_f \in \mathbb{R}^d$ are the weights and bias of the given invariant linear layer $L_{k \to 0}$, and each $\mathbf{X}^a \in \mathbb{R}^d$ is a chunk that coincides with reserved channels in Eq. (42). By plugging in the sum-pooled representation in Eq. (47), we finally obtain the following:

$$\text{MLP} \circ \text{sumpool} \circ \mathcal{T}^{(T)} \circ ... \circ \mathcal{T}^{(1)}(\mathbf{X}') = f'' \circ \text{sumpool} \circ \mathcal{T}^{(T)} \circ ... \circ \mathcal{T}^{(1)}(\mathbf{X}') \tag{49}$$

$$= \text{MLP}_k \left( \sum_{a \in [\text{bell}(k)]} \sum_{\mathbf{i}} \mathbf{C}_{\mathbf{i}}^{\gamma_a} \mathbf{Y}_{\mathbf{i}} w_{\mu_a} + b_f \right) \tag{50}$$

$$= \text{MLP}_k \circ L_{k \to 0}(\mathbf{Y}) \tag{51}$$

$$= \text{MLP}_k \circ L_{k \to 0} \circ L_{k \to k}^{(T)} \circ \sigma \circ ... \circ \sigma \circ L_{k \to k}^{(1)}(\mathbf{X}) \tag{52}$$

$$= F_k(\mathbf{X}), \tag{53}$$

where the last equality comes from Definition 1.

Taken together, we arrive at the conclusion that $\text{MLP} \circ \text{sumpool} \circ \mathcal{T}^{(T)} \circ ... \circ \mathcal{T}^{(1)}(\mathbf{X}')$ can approximate $F_k(\mathbf{X})$ to arbitrary precision. $\qquad\square$

## A.2 Additional Discussion on Linear Attention for Graph Transformers (Section 4)

We provide an additional discussion on related work, specifically on why Graphormer [78], based on fully-connected self-attention on nodes, is not compatible with many linear attention methods that reduce the memory complexity from $\mathcal{O}(n^2)$ to $\mathcal{O}(n)$. A range of prior graph Transformers including EGT [29], GRPE [54], and SAN [38] can be analyzed analogously. Let us first remind self-attention with query, key, value $\mathbf{Q}, \mathbf{K}, \mathbf{V} \in \mathbb{R}^{n \times d}$ and self-attention matrix $\boldsymbol{\alpha} \in \mathbb{R}^{n \times n}$:

$$\text{Att}(\mathbf{Q}, \mathbf{K}, \mathbf{V})_i = \sum_j \boldsymbol{\alpha}_{ij} \mathbf{V}_j \text{ where } \boldsymbol{\alpha}_{ij} = \frac{\exp(\mathbf{Q}_i^\top \mathbf{K}_j / \sqrt{d})}{\sum_k \exp(\mathbf{Q}_i^\top \mathbf{K}_k / \sqrt{d})}. \tag{54}$$

For graphs, as self-attention on nodes alone cannot recognize the edge connectivity, Graphormer incorporates the structural information of an input graph $\mathcal{G}$ into the self-attention matrix $\boldsymbol{\alpha}^{\mathcal{G}} \in \mathbb{R}^{n \times n}$ via attention bias matrix $\mathbf{b}^{\mathcal{G}} \in \mathbb{R}^{n \times n}$ (referred to as the edge and spatial encoding) as the following:

$$\boldsymbol{\alpha}_{ij}^{\mathcal{G}} = \frac{\exp(\mathbf{Q}_i^\top \mathbf{K}_j / \sqrt{d} + \mathbf{b}_{ij}^{\mathcal{G}})}{\sum_k \exp(\mathbf{Q}_i^\top \mathbf{K}_k / \sqrt{d} + \mathbf{b}_{ij}^{\mathcal{G}})}. \tag{55}$$

Unfortunately, this modification immediately precludes the adaptation of many efficient attention techniques developed for pure self-attention. As representative examples, we take Performer [11], Linear Transformer [32], Efficient Transformer [62], and Random Feature Attention [55]. The methods are based on kernelization of the $\text{Att}(\cdot)$ operator as the following:

$$\text{Att}_\phi(\mathbf{Q}, \mathbf{K}, \mathbf{V})_i = \sum_j \frac{\phi(\mathbf{Q}_i)^\top \phi(\mathbf{K}_j)}{\sum_k \phi(\mathbf{Q}_i)^\top \phi(\mathbf{K}_k)} \mathbf{V}_j = \frac{\phi(\mathbf{Q}_i)^\top \left( \sum_j \phi(\mathbf{K}_j) \mathbf{V}_j^\top \right)}{\phi(\mathbf{Q}_i)^\top \left( \sum_k \phi(\mathbf{K}_k) \right)}. \tag{56}$$

As the above factorization of $\exp(\cdot)$ into a pairwise dot product eliminates the need to explicitly compute the attention matrix, it reduces both time and memory cost of self-attention to $\mathcal{O}(n)$. Yet, in Eq. (55), since the bias $\mathbf{b}_{ij}^{\mathcal{G}}$ is added to the dot product *before* $\exp(\cdot)$, it is required that the full attention matrix $\boldsymbol{\alpha}^{\mathcal{G}} \in \mathbb{R}^{n \times n}$ is always explicitly computed. Thus, Graphormer and related variations are unable to utilize the method and are bound to $\mathcal{O}(n^2)$.

While above discussion regards kernelization, a wide range of other efficient Transformers, including Set Transformer [39], LUNA [45], Linformer [71], Nyströmformer [76], Perceiver [31], and Perceiver-IO [30] are not applicable to Graphormer due to similar reasons.

## A.3 Experimental Details (Section 5)

We provide detailed information on the datasets and models used in our experiments in Section 5. Dataset statistics can be found in Table 3.

Table 3: Statistics of the datasets.

(a) Statistics of Barabási-Albert random graph dataset.

| Dataset | Barabási-Albert |
|---|---|
| Size | 1280 |
| Average # node | 14.9 |
| Average # edge | 47.8 |

(b) Statistics of PCQM4Mv2 dataset.

| Dataset | PCQM4Mv2 |
|---|---|
| Size | 3.7M |
| Average # node | 14.1 |
| Average # edge | 14.6 |

### A.3.1 Implementation Details of Node and Type Identifiers

In most of our experiments on graph data ($k = 2$), we fix the Transformer encoder configuration and experiment with choices of node identifiers $\mathbf{P} \in \mathbb{R}^{n \times d_p}$ and type identifiers $\mathbf{E} \in \mathbb{R}^{2 \times d_e}$ (Section 2).

For type identifiers $\mathbf{E}$, we set $d_e$ equal to the hidden dimension $d$ of the main encoder $d_e = d$ and initialize and train them jointly with the model.

For orthonormal node identifiers $\mathbf{P}$, we use normalized orthogonal random features (ORFs) or Laplacian eigenvectors obtained as follows:

- For orthogonal random features (ORFs), we use rows of random orthogonal matrix $\mathbf{Q} \in \mathbb{R}^{n \times n}$ obtained with QR decomposition of random Gaussian matrix $\mathbf{G} \in \mathbb{R}^{n \times n}$ [79, 12].

- For Laplacian eigenvectors, we perform eigendecomposition of graph Laplacian matrix, *i.e.*, rows of $\mathbf{U}$ from $\boldsymbol{\Delta} = \mathbf{I} - \mathbf{D}^{-1/2} \mathbf{A} \mathbf{D}^{-1/2} = \mathbf{U}^\top \boldsymbol{\Lambda} \mathbf{U}$, where $\mathbf{A} \in \mathbb{R}^{n \times n}$ is adjacency matrix, $\mathbf{D}$ is degree matrix, and $\boldsymbol{\Lambda}$, $\mathbf{U}$ correspond to eigenvalues and eigenvectors respectively [20].

The model expects $d_p$-dimensional node identifiers $\mathbf{P} \in \mathbb{R}^{n \times d_p}$, while ORF and Laplacian eigenvectors are $n$-dimensional. To resolve this, if $n < d_p$, we zero-pad the channels. If $n > d_p$, for ORF we randomly sample $d_p$ channels and discard the rest, and for Laplacian eigenvectors we use $d_p$ eigenvectors with the smallest eigenvalues following common practice [20, 42, 38].

As the Laplacian eigenvectors are defined up to the factor $\pm 1$ after normalized to unit length [20], we randomly flip their signs during training. For PCQM4Mv2 (Section 5.2), we apply random dropout on eigenvectors during training, similar to 2D channel dropout in ConvNets [67]. In our experiments with PCQM4Mv2, we find that both sign flip and eigenvector dropout work as effective regularizers and improves performance on validation data.

### A.3.2 Second-Order Equivariant Basis Approximation (Section 5.1)

**Dataset** For the equivariant basis approximation experiment, we use a synthetic dataset containing Barabási-Albert (BA) random graphs [2]. With $\mathcal{U}$ denoting discrete uniform distribution, each graph is generated by first sampling the number of nodes $n \sim \mathcal{U}(10, 20)$ and the number for preferential attachment $k \sim \mathcal{U}(2, 3)$, then iteratively adding $n$ nodes by linking each new node to $k$ random previous nodes. We do not utilize node or edge attributes and only use edge connectivity. We generate 1152 graphs for training and 128 for testing. Further dataset statistics is provided in Table 3a.

**Architecture** Each model tested in Table 1 is a single multihead self-attention layer (Eq. (9)) with hidden dimension $d = 1024$, heads $H = \text{bell}(2 + 2) = 15$, and head dimension $d_H = 128$. As for the node identifier dimension, we use $d_p = 24$ for ORF and $d_p = 20$ for Laplacian eigenvectors.

**Experimental Setup** We experiment with sparse or dense input graph representations. For sparse input, we embed each graph with $n$ nodes and $m$ edges into $\mathbf{X}^{in} \in \mathbb{R}^{(n+m) \times (2d_p + d_e)}$. For dense input, we use all $n^2$ pairwise edges and obtain $\mathbf{X}^{in} \in \mathbb{R}^{n^2 \times (2d_p + d_e)}$, so that sparse edge connectivity is only used for obtaining Laplacian node identifiers.

For both sparse and dense inputs, we follow the standard procedure in Section 2 to use node and type identifiers to obtain $\mathbf{X}^{in} \in \mathbb{R}^{N \times (2d_p + d_e)}$ where $N = (n + m)$ or $n^2$, and project it to dimension $d$ with trainable projection $w^{in}$. We also utilize a special token [null] with trainable embedding $\mathbf{X}_{\texttt{[null]}} \in \mathbb{R}^d$ (we shortly explain its use) to obtain the final input $[\mathbf{X}_{\texttt{[null]}}; \mathbf{X}^{in} w^{in}] \in \mathbb{R}^{(1+N) \times d}$.

For an input $[\mathbf{X}_{\texttt{[null]}}; \mathbf{X}^{in}w^{in}]$, the goal is to supervise each of the $H = 15$ self-attention heads with attention matrices $\boldsymbol{\alpha}^1, ..., \boldsymbol{\alpha}^{15}$ to explicitly approximate row-normalized version of each equivariant basis $\mathbf{B}^{\mu_1}, ..., \mathbf{B}^{\mu_{15}} \in \mathbb{R}^{N \times N}$ (Eq. (2)) on the $N = (n + m)$ or $n^2$ input tokens (except $\texttt{[null]}$). An issue in supervision is that for rows of $\mathbf{B}^\mu$ that only contains zeros, normalization is not defined. To sidestep this, for such rows we simply supervise to attend to the special $\texttt{[null]}$ token only. For all other rows of $\mathbf{B}^\mu$ that contains nonzero entry, we supervise the model to ignore $\texttt{[null]}$ token.

**Training and Evaluation** We train and evaluate all models with L2 loss between attention matrix $\boldsymbol{\alpha}^h$ and normalized basis tensor $\mathbf{B}^{\mu_h}$ (involving $\texttt{[null]}$ token) averaged over heads $h = 1, ..., 15$. We train all models with AdamW optimizer [44] on 4 RTX 3090 GPUs each with 24GB. For sparse inputs we use batch size 512, and for dense inputs we use batch size 256 due to increased memory cost. We train all models for 3k steps (which takes about ∼1.5 hours) and apply linear learning rate warmup for 1k steps up to 1e-4 followed by linear decay to 0. For all models, we use dropout rate of 0.1 on the input $[\mathbf{X}_{\texttt{[null]}}; \mathbf{X}^{in}w^{in}]$ to prevent overfitting.

### A.3.3 Large-Scale Graph Learning (Section 5.2)

**Dataset** For large-scale learning, we use the PCQM4Mv2 quantum chemistry regression dataset from the OGB-LSC benchmark [27] that contains 3.7M molecular graphs. Along with graph structure, we utilize both node and edge features *e.g.*, atom and bond types following our standard procedure in Section 2. Dataset statistics is provided in Table 3b.

**Architecture** All our models in Table 2 (under *Pure Transformers*) have the same encoder configuration following Graphormer [78], with 12 layers, hidden dimension $d = 768$, heads $H = 32$, and head dimension $d_H = 24$. We adopt PreLN [75] that places layer normalization before MSA layer (Eq. (9)), MLP layer (Eq. (10)), and the final output projection after the last encoder layer. We implement MLP (Eq. (10)) as a stack of two linear layers with GeLU nonlinearity [25] in between. As for node identifier dimension, we use $d_p = 64$ for ORF and $d_p = 16$ for Laplacian eigenvectors.

As an additional GNN baseline, we run Graph Attention Network (GATv2) [69, 5] under several configurations. For GAT and GAT-VN in Table 2, we use 5-layer GATv2 with hidden dimension 600 and a single attention head, having 6.7M parameters in total. For GAT-VN (large), we use a 10-layer GATv2 with hidden dimension 1200 and a single attention head, having 55.2M parameters in total. For GAT-VN and GAT-VN (large), we use virtual node that helps modeling global interaction [27].

**Training and Evaluation** We mainly report and compare the Mean Absolute Error (MAE) on the validation data, and report MAE on the hidden test data if possible. We train all models with L1 loss using AdamW optimizer [44] with gradient clipping at global norm 5.0. We use batch size 1024 and train the models on 8 RTX 3090 GPUs with 24GB for ∼3 days. We train our models for 1M iterations, and apply linear lr warmup for 60k iterations up to 2e-4 followed by linear decay to 0. For our models in Table 2 except TokenGT (Lap) + Performer, we use the following regularizers:

- Attention and MLP dropout rate 0.1
- Weight decay 0.1
- Stochastic depth [28, 64] with linearly increasing layer drop rate, reaching 0.1 at last layer
- Eigenvector dropout rate 0.2 for TokenGT (Lap) (see Appendix A.3.1)

For TokenGT (Lap) + Performer in Table 2, we load a trained model checkpoint of TokenGT (Lap), change its self-attention to FAVOR+ kernelized attention of Performer [11] that can provably accurately approximate softmax attention, and fine-tune it with AdamW optimizer for 0.1M training steps with 1k warmup iterations and cosine learning rate decay. With batch size 1024 on 8 RTX 3090 GPUs, fine-tuning takes ∼ 12 hours. We do not use stochastic depth and eigenvector dropout for fine-tuning. For GAT baselines in Table 2, we use batch size 256 and train the models for 100 epochs with initial learning rate 0.001 decayed with a factor of 0.25 every 30 epochs.

### A.4 Additional Experimental Results (Section 5)

We report additional experimental results and discussions that could not be included in the main text due to space restriction.

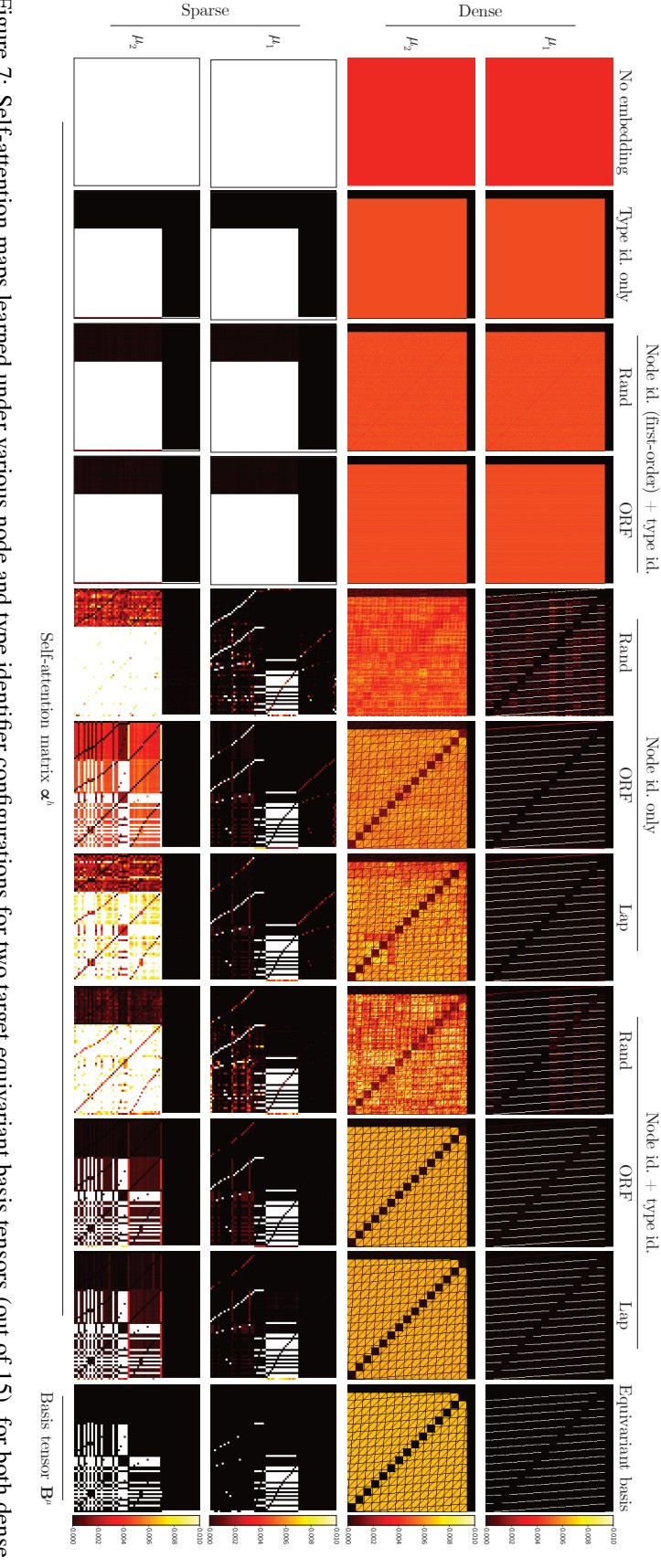

Figure 7: Self-attention maps learned under various node and type identifier configurations for two target equivariant basis tensors (out of 15), for both dense and sparse inputs. For better visualization, we clamp the entries by 0.01. Self-attention learns acute pattern coherent to equivariant basis when orthonormal node identifiers and type identifiers are provided both as input.

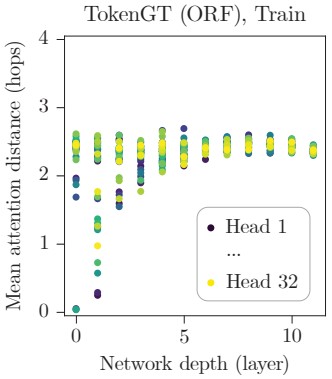
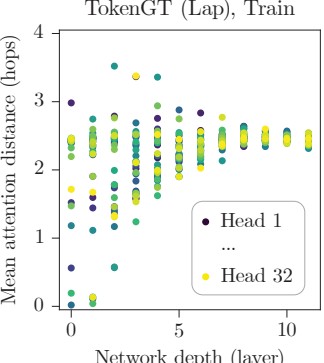

Figure 8: Attention distance by head and network depth, measured for entire PCQM4Mv2 training set. Each dot shows mean attention distance in hops across graphs of a head at a layer.

Table 4: Statistics of the transductive node classification datasets.

| Dataset | CS | Physics | Photo | Computers | Chameleon | Crocodile |
|---|---|---|---|---|---|---|
| # nodes | 18,333 | 34,493 | 7,650 | 13,752 | 2,277 | 11,631 |
| # edges | 81,894 | 247,962 | 119,081 | 245,861 | 36,101 | 180,020 |
| # classes | 15 | 5 | 8 | 10 | 6 | 6 |

Table 5: Transductive node classification. OOM denotes out-of-memory error on a 24GB RTX 3080 GPU. We report aggregated test accuracy at best validation accuracy over 7 randomized runs.

| | CS | Physics | Photo | Computers | Chameleon | Crocodile |
|---|---|---|---|---|---|---|
| GCN | $0.895 \pm 0.004$ | $0.932 \pm 0.004$ | $0.926 \pm 0.008$ | $0.873 \pm 0.004$ | $0.593 \pm 0.01$ | $0.660 \pm 0.01$ |
| GAT | $0.893 \pm 0.005$ | $0.937 \pm 0.01$ | $0.947 \pm 0.006$ | $\mathbf{0.914 \pm 0.002}$ | $0.632 \pm 0.011$ | $0.692 \pm 0.017$ |
| GIN | $0.895 \pm 0.005$ | $0.886 \pm 0.046$ | $0.886 \pm 0.017$ | $0.362 \pm 0.051$ | $0.479 \pm 0.027$ | $0.515 \pm 0.041$ |
| Graphormer | $0.791 \pm 0.015$ | OOM | $0.894 \pm 0.004$ | $0.814 \pm 0.013$ | $0.457 \pm 0.011$ | $0.489 \pm 0.014$ |
| TokenGT (Near-ORF) + Performer | $0.882 \pm 0.007$ | $0.931 \pm 0.009$ | $0.872 \pm 0.011$ | $0.82 \pm 0.019$ | $0.568 \pm 0.019$ | $0.583 \pm 0.024$ |
| TokenGT (Lap) + Performer | $0.902 \pm 0.004$ | $0.941 \pm 0.007$ | $0.919 \pm 0.009$ | $0.86 \pm 0.012$ | $0.637 \pm 0.032$ | $0.638 \pm 0.025$ |
| TokenGT (Lap) + Performer + SEB | $\mathbf{0.903 \pm 0.004}$ | $\mathbf{0.950 \pm 0.003}$ | $\mathbf{0.949 \pm 0.007}$ | $0.912 \pm 0.006$ | $\mathbf{0.653 \pm 0.029}$ | $\mathbf{0.718 \pm 0.012}$ |

### A.4.1 Second-Order Equivariant Basis Approximation (Section 5.1)

In addition to the Figure 2 in the main text that shows learned self-attention maps for dense input, in Figure 7, we provide an extended visualization of self-attention maps for both dense and sparse inputs. Consistent to Lemma 1 and Table 1, self-attention achieves accurate approximation of equivariant basis only when both the orthonormal node identifiers (ORF or Lap) and type identifiers are given.

### A.4.2 Large-Scale Graph Learning (Section 5.2)

In addition to the Figure 3 in the main text that shows attention distance measured for the PCQM4Mv2 validation data, in Figure 8, we provide an extended figure of attention distance measured for the entire training set that contains ∼3M graphs. Overall we find similar trends as analyzed in Section 5.2.

### A.4.3 Transductive Node Classification on Large Graphs (Section 5)

While our main experiment in Section 5.2 focuses on graph-level predictions, TokenGT can in principle be applied to a more broad class of node-level or edge-level graph understanding tasks by putting prediction head on appropriate output tokens. To demonstrate this, we conduct additional experiments on a variety of transductive node classification datasets. In contrast to PCQM4Mv2, they involve large graphs with up to tens of thousands of nodes, posing a challenge to $\mathcal{O}(n^2)$ complexity methods such as graph Transformers that rely on dense attention bias.

**Dataset** We use transductive node classification datasets, where each data is represented as a node in a large-scale graph, including co-authorship (CS, Physics) [61], co-purchase (Photo, Computers) [61], and Wikipedia page networks (Chameleon, Crocodile) [59]. We randomly split the dataset into train, validation, and test sets by randomly reserving 30 random nodes per class for validation and test respectively, and use the rest of the nodes for training. Dataset statistics is provided in Table 4.

**Approach**   We utilize simple variants of TokenGT with Performer kernel attention of $\mathcal{O}(n+m)$ complexity. Due to the large number of nodes $n$, an immediate challenge for TokenGT is dealing with the orthonormality assumption on the node identifiers (Lemma 1) as the maximal number of orthonormal node identifiers is bounded by dimension $d_p$. In this case, it is reasonable to introduce *near-orthonormal* vectors as node identifiers, as it is theoretically guaranteed that we can draw an exponential number $\mathcal{O}(e^{\Omega(d_p)})$ of $d_p$-dimensional near-orthonormal vectors [22]. For *TokenGT (Near-ORF)*, we use $d_p = 64$-dimensional random node identifiers where each entry is sampled from $\{-1/d_p, +1/d_p\}$ with coin toss [22]. For *TokenGT (Lap)*, we use a subset of the Laplacian eigenvectors as node identifiers, specifically $d_p/2$ eigenvectors with lowest eigenvalues and $d_p/2$ eigenvectors with highest eigenvalues, and choose $d_p$ among 64-100 based on validation performance.

While *Near-ORF* and *Lap* can theoretically serve as an efficient low-rank approximation for orthonormal node identifiers, their approximation can affect the quality of modeled equivariant basis (Section 3). In particular, equivariant basis ($\mu$) represented as **sparse** basis tensor ($\mathbf{B}^\mu$; Definition 4) are expected to be affected more, as they require most entries to be zero. To remedy this, we take a simple approach of residually adding one of such sparse equivariant operators $\mathbf{X}_{ii} \mapsto \mathbf{X}_{ii} + \sum_{j \neq i} \mathbf{X}_{ij}$ explicitly after each Transformer layer. We denote this variant as *TokenGT (Lap) + Performer + SEB*, where SEB abbreviates sparse equivariant basis. This fix is minimal, easy to implement, and highly efficient as it only requires a single `torch.coalesce()` call, and also empirically effective.

**Architecture**   All our models in Table 5 utilize a linear prediction head on the node tokens obtained at the final Transformer layer to perform node-level classification. We perform an exploratory hyperparameter search over the number of layers from 2-4, heads $H$ from 1-4, hidden dimension $d$ from 128-1024, and dropout rate from $\{0.1, 0.5\}$, based on validation performance.

We employ strong message-passing GNN and graph Transformer baselines, including GCN [36], GAT [69], GIN [77] which has 2-WL expressiveness similar to ours, and Graphormer [78] based on fully-connected node self-attention. For message-passing GNNs, we use a 4-layer architecture and search hidden dimension $d$ from $\{64, 1024\}$ based on validation performance. For Graphormer, we perform an exploratory search on the number of layers from 1-4, heads $H$ from 1-4, and hidden dimension $d$ from 128-1024 based on validation performance. We apply 0.5 dropout for all baselines.

**Training and Evaluation**   We report and compare classification accuracy on the test nodes at best validation accuracy aggregated over 7 randomized runs. We train all models with node-level categorical cross-entropy loss using Adam optimizer [35] on a single RTX 3090 GPU with 24GB. We train all models with a learning rate of 1e-3 for 300 epochs.

**Results**   The results are in Table 5. Graphormer [78] suffers out-of-memory in the Physics dataset mainly due to the spatial encoding that requires $\mathcal{O}(n^2)$ memory. By constraining the model capacity appropriately, we were able to run Graphormer on other datasets. However, we observe a low performance, presumably due to the memory cost that prevents depth and head scaling. As the spatial encoding is incorporated into the model via attention bias, the model strictly requires $\mathcal{O}(n^2)$ memory and cannot be easily made more efficient. On the other hand, TokenGT variants are able to utilize Performer attention with $\mathcal{O}(m+n)$ cost, which allows using larger models to achieve the best performance in all but one dataset (Computers, where the performance is on par with the best model).

## A.5   Additional Discussion on Performance on PCQM4Mv2 (Section 5.2)

As in the Table 2 in the main text, TokenGT currently shows a slightly lower performance compared to the Graphormer and its successors in the PCQM4Mv2 benchmark. We conjecture this is partly because we intentionally keep its components simple to faithfully adhere to the equivariance theory. We discuss some engineering approaches that may enhance the performance of TokenGT *at the cost of differentiating from the theory*. We consider engineering TokenGT to match or outperform sophisticated graph Transformers as a promising and important next research direction.

**Node Identifiers**   Our best performing TokenGT (Lap) currently uses Laplacian eigenvectors [20] as the node identifiers, which has been criticized for issues such as loss of structural information [38] and sign ambiguity [42]. Thus, one could try to relax the theoretical requirement for orthonormality of node identifiers and incorporate more powerful node positional encodings [38, 42] as node identifiers, which could potentially yield better performance in practice.

**(Hyper)Edge Tokens** TokenGT currently treats an undirected input edge $(u, v)$ as if both directions $(u, v)$ and $(v, u)$ are present, leading to a pair of edge tokens $[\mathbf{X}_{(u,v)}, \mathbf{P}_u, \mathbf{P}_v]$ and $[\mathbf{X}_{(v,u)}, \mathbf{P}_v, \mathbf{P}_u]$. Similarly, an undirected order-$k$ input hyperedge $(v_1, ..., v_k)$ of an higher-order hypergraph is parsed to all possible orderings of node identifiers. While this is a common characteristic of tensor-based permutation equivariant neural networks [47, 48, 33, 60, 46, 34], they can lead to memory overhead and redundancy since multiple tokens represent an identical undirected edge. To avoid this, one can use a single token for each undirected (hyper)edge and pool the node identifiers as $\sum_{i=1}^{k} \rho(\mathbf{P}_{v_i})$. Combined with powerful node identifiers, this approach could potentially enhance the model performance.