# OpenReview forum: "Pure Transformers are Powerful Graph Learners"
_NeurIPS.cc/2022/Conference — NeurIPS 2022 Accept_

### Official Review · Reviewer_pDDE · 2022-07-11

**Rating:** 6
**Confidence:** 3
**Soundness:** 3 good
**Presentation:** 3 good
**Contribution:** 3 good

**Summary:**

The authors show that we can use the vanilla Transformers for languages and Vit without any graph-specific modifications can lead to promising results in graph learning. The interesting view is that the proposed model Soft Graph Transformer (SGT) uses edge features as tokens compared to other graph learners who only use node features as input tokens. And identifiers are designed to encode the graph's local information. Therefore, they can use the techniques for Transformers without any other modifications. Furthermore, they also theoretically proved their equivalence to the k-IGN models and the k-WL algorithms. However, I think their empirical evaluations are not enough and the advantages of their proposed SGT are not clear compared with other Graph Transformers.

**Questions:**

Questions:

1. I'd like to know whether other baselines like GraphFormers use the Laplacian positional embedding or not. Since the Laplacian positional embeddings are useful tricks and have been used in GraphFormers, SAN and other works.

2. I'd like to know why SGT performs worse than other GraphFormers. Do you have any possible causes?

**Ethics Review Area:**

["I don’t know"]

**Limitations:**

I think they should clearly tell the readers that they are not the first who uses the Laplacian's eigenvectors in Graph Transformers since this trick is a simple but effective one.

**Strengths And Weaknesses:**

Strengths:
1. Proposed a new view on the tokens instead of only using node features.
2. They also give a theoretical analysis of SGT's representative ability as powerful as k-WL and more expressive than GCN. Their analysis also gives us a direct reason why the node identifiers are orthogonal.

Weaknesses:
1. It is still not clear to me why Transformers for graph need to be the same as other transformers. I think it is natural that Graph Transformers need to have something special for graph tasks. According to "No free lunch" theorem, there can not be an algorithm that works the best for all tasks. Therefore, considering task-specific models are necessary. For example, Swin-Transformers takes the inductive bias for vision tasks into Transformers for vision tasks.

2. The empirical results also show no advantages compared with other Transformers for Graphs with larger resources cost. As for SGT+Performer, I'm not sure whether such a comparison is fair since other Transformers for Graphs may also use the linearized acceleration by modifying their models like Linformer's attention?


Typo:
line 286 Figure 1 -> Figure 2

---

> ### Author Response · Authors · 2022-08-02
> **Official Response to Reviewer pDDE (5/5)**
>
> > Typo: line 286 Figure 1 -> Figure 2
>
> A5. We appreciate the comment and will fix the typo.
>
> > I'd like to know whether other baselines like GraphFormers use the Laplacian positional embedding or not. Since the Laplacian positional embeddings are useful tricks and have been used in GraphFormers, SAN and other works. (...) I think they should clearly tell the readers that they are not the first who uses the Laplacian's eigenvectors in Graph Transformers since this trick is a simple but effective one.
>
> A6. We appreciate the comment. Other baselines in Table 2, including Graphormer that uses shortest path-based spatial encoding, do not utilize Laplacian eigenvectors. An exception is EGT [1] that uses singular value decomposition on the adjacency matrix to obtain node features, which can be interpreted similarly to Laplacian eigenvectors. Still, we note that their framework is based on the edge encodings of Graphormer [2], which is largely different from our approach that does not require attention bias. We will revise the main text so that the relation to prior work [3, 4, 5] that introduce Laplacian eigenvectors as graph positional embedding is more clear.
>
> [1] Hussain et al., Global Self-Attention as a Replacement for Graph Convolution (2022)
>
> [2] Ying et al., Do Transformers Really Perform Bad for Graph Representation? (2022)
>
> [3] Dwivedi et al., Benchmarking Graph Neural Networks (2020)
>
> [4] Kreuzer et al., Rethinking Graph Transformers with Spectral Attention (2021)
>
> [5] Lim et al., Sign and Basis Invariant Networks for Spectral Graph Representation Learning (2022)

---

> ### Author Response · Authors · 2022-08-02
> **Official Response to Reviewer pDDE (4/5)**
>
> > As for SGT+Performer, I'm not sure whether such a comparison is fair since other Transformers for Graphs may also use the linearized acceleration by modifying their models like Linformer's attention?
>
> A4. To address the concern, let us explain why Graphormer [1] that utilizes fully-connected self-attention operator on nodes cannot utilize many efficient attention methods to reduce the memory complexity from $\mathcal{O}(n^2)$ to $\mathcal{O}(n)$. Prior graph transformers including EGT [2], GRPE [3], and SAN [4] can be analyzed analogously. Let us first remind self-attention with query, key, and value $\mathbf{Q}, \mathbf{K}, \mathbf{V}\in\mathbb{R}^{n\times d}$ and the attention matrix $\boldsymbol{\alpha}\in\mathbb{R}^{n\times n}$:
>
> $\text{Att}(\mathbf{Q}, \mathbf{K}, \mathbf{V})\_i = \sum\_j\boldsymbol{\alpha}\_{ij}\mathbf{V}\_j \text{ where }\boldsymbol{\alpha}\_{ij} = \frac{\exp\left(\mathbf{Q}\_i^\top \mathbf{K}\_j/\sqrt{d}\right)}{\sum\_{k}{\exp\left(\mathbf{Q}\_i^\top \mathbf{K}\_k/\sqrt{d}\right)}}.$
>
> For graphs, as the self-attention on nodes alone cannot incorporate the edge connectivity structure, Graphormer incorporates the graph information into the self-attention matrix $\alpha\in\mathbb{R}^{n\times n}$ utilizing the attention bias matrix $\textbf{b}\in\mathbb{R}^{n\times n}$ (referred to as the edge/spatial encoding) as the following:
>
> $\boldsymbol{\alpha}\_{ij} = \frac{\exp\left(\mathbf{Q}\_i^\top \mathbf{K}\_j/\sqrt{d} + \mathbf{b}\_{ij}\right)}{\sum\_{k}{\exp\left(\mathbf{Q}\_i^\top \mathbf{K}\_k/\sqrt{d} + \mathbf{b}\_{ik}\right)}}.$
>
> Here, the attention bias matrix $\textbf{b}\in\mathbb{R}^{n\times n}$ is the essential component to encorporate graph structure into computation. Unfortunately, this modification immediately precludes the adaptation of many efficient attention techniques developed for pure self-attention. As representative examples, let us take Performer [5] (which we mainly used in our work), linear Transformer [6], efficient Transformer [7], and Random Feature Attention [8]. The methods are based on kernelization of the $\text{Att}()$ operator as following [5]:
>
> $\text{Att}\_\phi(\mathbf{Q}, \mathbf{K}, \mathbf{V})\_i = \sum\_j\frac{\exp\left(\mathbf{Q}\_i^\top \mathbf{K}\_j/\sqrt{d}\right)}{\sum\_{k}{\exp\left(\mathbf{Q}\_i^\top \mathbf{K}\_k/\sqrt{d}\right)}}\mathbf{V}\_j$
> $= \sum\_j\frac{\phi(\mathbf{Q}\_i)^\top \phi(\mathbf{K}\_j)}{\sum\_{k}{\phi(\mathbf{Q}\_i)^\top \phi(\mathbf{K}\_k)}}\mathbf{V}\_j = \frac{\phi(\mathbf{Q}\_i)^\top\left(\sum\_j\phi(\mathbf{K}\_j)\mathbf{V}\_j^\top\right)}{{\phi(\mathbf{Q}\_i)^\top\left(\sum\_{k}\phi(\mathbf{K}\_k)\right)}}.$
>
> The above factorization of the exponential into a pairwise dot product, in turn, eliminates the need to explicitly compute the attention matrix for computing $\text{Att}\_\phi()$ and consequently reduces both time and memory cost to $\mathcal{O}(n)$. Unfortunately, Graphormer and related variations are fundamentally unable to utilize the method. The bias term $\mathbf{b}\_{ij}$ is added to the dot product before the exponential, requiring that the full pairwise self-attention matrix $\boldsymbol{\alpha}\in\mathbb{R}^{n\times n}$ is always explicitly computed to obtain the output and leading to $\mathcal{O}(n^2)$.
>
> While our above explanation mainly regards kernelization methods, a number of other efficient Transformers, including Set Transformer [9], LUNA [10], Linformer [11], Nyströmformer[12], Perceiver [13], and Perceiver-IO [14] are not applicable to Graphormer due to similar reasons. We appreciate the comment and will add relevant discussions to the main text.
>
> [1] Ying et al., Do Transformers Really Perform Bad for Graph Representation? (2022)
>
> [2] Hussain et al., Global Self-Attention as a Replacement for Graph Convolution (2022)
>
> [3] Park et al., GRPE: Relative Positional Encoding for Graph Transformer (2022)
>
> [4] Kreuzer et al., Rethinking Graph Transformers with Spectral Attention (2022)
>
> [5] Choromanski et al., Rethinking Attention with Performers (2020)
>
> [6] Katharopoulos et al., Transformers are RNNs: Fast Autoregressive Transformers with Linear Attention (2020)
>
> [7] Shen et al., Efficient Attention: Attention with Linear Complexities (2018)
>
> [8] Peng et al., Random Feature Attention (2021)
>
> [9] Lee et al., Set Transformer: A Framework for Attention-based Permutation-Invariant Neural Networks (2018)
>
> [10] Ma et al., Luna: Linear Unified Nested Attention (2021)
>
> [11] Wang et al., Linformer: Self-Attention with Linear Complexity (2020)
>
> [12] Xiong et al., Nyströmformer: A Nyström-Based Algorithm for Approximating Self-Attention (2021)
>
> [13] Jaegle et al., Perceiver: General Perception with Iterative Attention (2021)
>
> [14] Jaegle et al., Perceiver IO: A General Architecture for Structured Inputs & Outputs (2021)

---

> ### Author Response · Authors · 2022-08-02
> **Official Response to Reviewer pDDE (3/5)**
>
> > It is still not clear to me why Transformers for graph need to be the same as other transformers. I think it is natural that Graph Transformers need to have something special for graph tasks. According to "No free lunch" theorem, there can not be an algorithm that works the best for all tasks. Therefore, considering task-specific models are necessary. For example, Swin-Transformers takes the inductive bias for vision tasks into Transformers for vision tasks.
>
> A2. While we agree with the reviewer's opinion, we think graph Transformers with a minimal graph-specific inductive bias is an exciting direction to explore by itself. This is mainly due to its potential to open a new chapter for incorporating graph-structured data such as scene graphs to generalist multi-modal agents similar to Perceiver [1] and Perceiver IO [2] . The design of such models are focused on reducing the data-specific inductive bias and instead scaling the data and model; to incorporate graphs to such pipeline, a design that adheres better to modality-agnostic architecture tech stack is needed. Our work provides a rigorous theoretical base on such design, by showing that pure Transformers can work well on graphs.
>
> [1] Jaegle et al., Perceiver: General Perception with Iterative Attention (2021)
>
> [2] Jaegle et al., Perceiver IO: A General Architecture for Structured Inputs & Outputs (2021)
>
> > The empirical results also show no advantages compared with other Transformers for Graphs with larger resources cost (...) I'd like to know why SGT performs worse than other GraphFormers. Do you have any possible causes?
>
> A3. While our model is currently underperformed by Graphormer and its successors, we think the low performance is, in part, because we intentionally kept its components simple to faithfully adhere to the equivariance theory. Indeed, direct comparison of our method to the graph transformers is unfair since these methods rely on various graph-specific features and inductive biases (e.g., centrality encoding, minimum shortest path distance, etc.) while our method intentionally eliminated them from the model and is based on pure attention mechanism of Transformers. We think it has a lot of room for performance improvement if we focus on engineering. For example, the model currently uses Laplacian eigenvectors [1] as node identifiers, which has been criticized for issues such as loss of structural information [2] and sign ambiguity [3]. We could, e.g., try to relax the theoretical requirement for orthonormality of node identifiers and incorporate such more powerful node positional encodings as node identifiers, which could potentially yield better performance in practice. We think engineering our model to match or outperform more sophisticated graph Transformers is a promising and important next research direction.
>
> [1] Dwivedi et al., Benchmarking Graph Neural Networks (2020)
>
> [2] Kreuzer et al., Rethinking Graph Transformers with Spectral Attention (2021)
>
> [3] Lim et al., Sign and Basis Invariant Networks for Spectral Graph Representation Learning (2022)

---

> ### Author Response · Authors · 2022-08-02
> **Official Response to Reviewer pDDE (2/5)**
>
> (continued from 1/5)
>
> Let us finish by explaining how we applied SGT for this task. Considering large $n$, an immediate challenge for our model is dealing with the orthonormality assumption on the node identifiers (Lemma 1), as the maximal number of orthonormal node identifers is upper bounded by its dimension $d_p$. In this case, it is reasonable to introduce *near-orthonormal vectors* as node identifiers, as it is theoretically guaranteed that we can draw an exponential number $\mathcal{O}^{\Omega(d_P)}$ of $d_p$-dimensional near-orthonormal vectors [7]. For *SGT (Near-ORF)*, we used $d_p = 64$-dimensional random node identifiers where each entry is sampled from $\{-1/d_p, +1/d_p\}$ under equal probability [7]. For *SGT (Lap)*, we used a subset of the Laplacian eigenvectors as node identifiers, more specifically $d_p/2$ eigenvectors with lowest eigenvalues and $d_p/2$ eigenvectors with highest eigenvalues, and choose $d_p$ in the range of $64$ to $100$ based on validation performance.
>
> While *Near-ORF* and *Lap* can theoretically serve as an efficient low-rank approximation for orthonormal node identifiers, their approximation can affect the quality of modeled equivariant basis. In particular, equivariant basis ($\mu$) represented as ***sparse*** basis tensor ($\mathbf{B}^\mu$; Definition 2) are expected to be affected more, as they require most entries to be zero. To remedy this, we take a simple approach of residually adding one of such sparse equivariant operators $\mathbf{X}\_{ii} \mapsto \mathbf{X}\_{ii} + \sum\_{j\neq i}\mathbf{X}\_{ij}$ explicitly after each Transformer layer. We denote this variant as *SGT (Lap) + Performer + Sp. Equiv. Basis*. This fix is minimal, easy to implement, and highly efficient as it only requires a single $\texttt{torch.coalesce()}$ call, and also empirically effective, as shown in Table 1.
>
> [1] Hu et al., OGB-LSC: A Large-Scale Challenge for Machine Learning on Graphs (2021)
>
> [2] Brown et al., Language Models are Few-Shot Learners (2020)
>
> [3] Dosovitsky et al., An Image is Worth 16x16 Words: Transformers for Image Recognition at Scale (2020)
>
> [4] Shchur et al., Pitfalls of Graph Neural Network Evaluation (2019)
>
> [5] Rozemberczki et al., Multi-scale Attributed Node Embedding (2019)
>
> [6] Xu et al., How Powerful are Graph Neural Networks? (2019)
>
> [7] Gorban et al., Approximation with random bases: Pro et Contra (2016)
>
> [8] Ying et al., Do Transformers Really Perform Badly for Graph Representation? (2021)

---

> ### Author Response · Authors · 2022-08-02
> **Official Response to Reviewer pDDE (1/5)**
>
> We thank the reviewer for the positive comments. Below we respond to the questions.
>
> > However, I think their empirical evaluations are not enough and the advantages of their proposed SGT are not clear compared with other Graph Transformers.
>
> A1. While we acknowledge that our empirical evaluation is conducted only on PCQM4Mv2, we would like to note that PCQM4Mv2 is one of the largest-scale graph dataset up to date containing 3.8 million graphs [1]. This makes it one of the few suitable benchmark to test our model as Transformers are generally designed to work with extremely large-scale data [2, 3].
>
> To further demonstrate the effectiveness of our method in a more broad class of graph understanding tasks, we conducted additional experiments on transductive node classification datasets including co-authorship (CS, Physics) [4], co-purchase (Photo, Computers) [4], and Wikipedia page networks (Chameleon, Crocodile) [5], which generally involve large-scale graphs. The statistics of the datasets are outlined below:
>
> | Dataset | CS | Physics | Photo | Computers | Chameleon | Crocodile |
> | --- | --- | --- | --- | --- | --- | --- |
> | # nodes ($n$) | 18,333 | 34,493 | 7,650 | 13,752 | 2,277 | 11,631 |
> | # edges ($m$) | 81,894 | 247,962 | 119,081 | 245,861 | 36,101 | 180,020 |
> | # classes | 15 | 5 | 8 | 10 | 6 | 6 |
>
> We randomly split the dataset into train, validation, and test sets by randomly reserving 30 random nodes per class for validation and test respectively, and using the rest for training. We experiment with simple variants of SGT equipped with Performer kernel attention (*details can be found at the end of the response*), and compare them against strong GNN baselines including the following:
> * GCN (message-passing that works well on large graphs [4])
> * GAT (message-passing based on attention)
> * GIN (message-passing with 2-WL expressiveness guarantee similar to ours [6])
> * Graphormer (graph transformer equipped with shortest path-based edge encoding and spatial encoding [9])
>
> Table 1. Results of transductive node classification experiment. OOM denotes out-of-memory error on a 24GB RTX 3080 GPU. We report aggregated test accuracy at best validation accuracy over 7 randomized runs.
> |  | CS | Physics | Photo | Computers | Chameleon | Crocodile |
> | --- | --- | --- | --- | --- | --- | --- |
> GCN | 0.895 +- 0.004 | 0.932 +- 0.004 | 0.926 +- 0.008 | 0.873 +- 0.004 | 0.593 +- 0.01 | 0.660 +- 0.01 |
> GAT | 0.893 +- 0.005 | 0.937 +- 0.01 | 0.947 +- 0.006 | **0.914 +- 0.002** | 0.632 +- 0.011 | 0.692 +- 0.017 |
> GIN | 0.895 +- 0.005 | 0.886 +- 0.046 | 0.886 +- 0.017 | 0.362 +- 0.051 | 0.479 +- 0.027 | 0.515 +- 0.041 |
> | Graphormer | 0.791 +- 0.015 | *OOM* | 0.894 +- 0.004 | 0.814 +- 0.013 | 0.457 +- 0.011 | 0.489 +- 0.014 |
> SGT (Near-ORF) + Performer | 0.882 +- 0.007 | 0.931 +- 0.009 | 0.872 +- 0.011 | 0.82 +- 0.019 | 0.568 +- 0.019 | 0.583 +- 0.024 |
> SGT (Lap) + Performer | 0.902 +- 0.004 | 0.941 +- 0.007 | 0.919 +- 0.009 | 0.86 +- 0.012 | 0.637 +- 0.032 | 0.638 +- 0.025 |
> SGT (Lap) + Performer + Sp. Equiv. Basis | **0.903 +- 0.004** | **0.950 +- 0.003** | **0.949 +- 0.007** | 0.912 +- 0.006 | **0.653 +- 0.029** | **0.718 +- 0.012** |
>
> The results are outlined in Table 1. Graphormer [8] results in out-of-memory in the Physics dataset mainly due to the spatial encoding that requires $\mathcal{O}(n^2)$ memory. By constraining the model capacity appropriately, we were able to run Graphormer on other datasets. However, we observe a low performance, presumably due to the memory complexity that prevents depth and head scaling. As the spatial encoding is incorporated into the model via attention bias, the model strictly requires $\mathcal{O}(n^2)$ memory and cannot be easily made more efficient. On the other hand, SGT variants are able to utilize Performer kernel attention with $\mathcal{O}(n+m)$ cost, which allows using larger models to achieve the best performance in all but one datasets (Computers, where the performance is on par with the best model).
>
> (continued to 2/5)

---

> > ### Author Response · Authors · 2022-08-02
> > **An update in the table**
> >
> > Dear reviewer, we managed to train Graphormer on Photo, Computers, and Crocodile (previously OOM) through a bit of optimization and hyperparameter search. We updated the scores in Table 1 and updated the relevant discussions.

---

> ### Comment · Reviewer_pDDE · 2022-08-09
> **Thanks for your detailed response.**
>
> Thanks for your detailed response, my concerns have been addressed. Thus I remain my tendency to accept this paper.

---

### Official Review · Reviewer_YAnt · 2022-07-12

**Rating:** 6
**Confidence:** 5
**Soundness:** 3 good
**Presentation:** 4 excellent
**Contribution:** 3 good

**Summary:**

The paper proposes a new way to encode nodes and edges on the graph so that one can directly transform the input graph into a sequence and directly use the sequence transformers. With carefully designed position, type encodings, the paper proves that it is theoretically as expressive as a second-order invariant graph network. The authors validate several of their claims through a synthetic experiment and an experiment on real-world dataset PCQM4M. It achieves comparable performance with prior models and has the potential to scale to large graphs since the proposed method can just do a drop-in replacement of the transformer with all efficient transformer architectures.

**Questions:**

- How do you set the dimension of node identifiers? Is the dimension the number of nodes in the graph? How do you deal with a batch of graphs with different sizes?
- Is there any difference between modeling directed and undirected edges? If one edge (u,v) is undirected, I think the modeling of (u,v) and (v,u) should be identical. However, it seems that the edge embedding will be $[\mathbf{X}_{(u,v)}, \mathbf{P}_u, \mathbf{P}_v]$ for (u,v) and $[\mathbf{X}_{(u,v)}, mathbf{P}_v, \mathbf{P}_u]$ for (v,u).
- Also wonder how does the above issue about directedness and undirectedness generalize to higher-order hypergraphs?
- It’s interesting that using Laplacian eigenvectors as node identifiers and not using type id leads to comparable performance with using type identifiers on sparse inputs (the 7th row vs the final three rows in Table 1). Any insights?
- Have you submitted your final results to get the numbers on the test set of PCQM4M? The authors do not seem to include a discussion on how they tune the hyperparameters on PCQM4M especially since you only have the validation set. The validation performance alone is not convincing enough, let alone the proposed method achieves higher MAE than prior transformer-based methods such as Graphormer.
- The model achieves higher MAE than all the three graph transformer baselines in Table 2. I did not find sufficient discussion on this result. There is basically a one-liner, which still emphasizes that the proposed method can be optimized computationally while not discussing the reasons for not matching the same empirical performance. State-of-the-art results are not necessary for publication but detailed discussion and analysis on why not is necessary.
- Can the proposed method be applied to other node-level or edge-level tasks like link prediction?
- The authors mentioned in the checklist that they included the code and data. However, their supplementary is only a PDF, I do not see the code to reproduce the concrete numbers listed in the experiment section.


**Limitations:**

The only limitation in my mind is that the proposed method may not necessarily generalize to large graphs as well as message passing networks. Although the paper mentioned that they can use all the efficient transformer architectures, yet there are no experiments, no evidence showing this. The full transformer architecture with quadratic complexity still runs. Besides, it would be nice to add more discussion on how the proposed method may be applied to other node and edge level tasks (as raised in the above section).

**Strengths And Weaknesses:**

The paper aims to serialize graphs so that we can leverage the benefits of all those expressive transformer architectures which have been investigated for the past few years. Although I’m not entirely sure that the prior graph transformers cannot leverage different attention/transformer architecture. I think the challenge of applying transformers to graphs is always how to differentiate nodes (i.e., designing different position encodings) while preserving the permutation invariance and equivariance. I like the second part of the paper where the model with the proposed position encodings is proved to be as expressive as a 2-IGN, and I also enjoy the first experiment which validates the theoretical claim.

One weakness/question in mind is on scalability and how to adapt to graphs with various sizes. The node identifier matrix $\mathbf{P} \in \mathbb{R}^{n \times d_p}$, where $d_p$ should be at least $n$. How can this scale to large graphs? Also the position encoding dimensions (node + type identifiers) may be much larger than the original node/edge features ($\mathbf{X}_v$).
Laplacian eigenvectors as node identifiers are still different from sinusoidal positional embeddings in NLP since the sinusoidal position embeddings can be of arbitrary dimension, but for the Laplacian eigenvectors, the dimension is fixed (which is the number of nodes on the graph).

Besides, from a practitioner point of view, the experimental results are not sufficient. The authors only evaluate their model on one real world dataset PCQM4M. There are multiple graph classification/regression dataset on Open Graph Benchmark and Benchmarking GNNs. Even on the PCQM4M dataset, it seems that all variants of the proposed methods are worse than prior graph transformers. Even with the claim that the proposed method can leverage efficient transformer implementations, it does not really address my concern. 1. you never show that the model actually works in a setting where we really need an efficient implementation, i.e., other quadratic implementations do not run or oom. 2. why can’t the prior works use the same efficient transformer implementations?

---

> ### Author Response · Authors · 2022-08-02
> **Official Response to Reviewer YAnt (5/5)**
>
> > Even on the PCQM4M dataset, it seems that all variants of the proposed methods are worse than prior graph transformers. (...) The model achieves higher MAE than all the three graph transformer baselines in Table 2. I did not find sufficient discussion on this result. There is basically a one-liner, which still emphasizes that the proposed method can be optimized computationally while not discussing the reasons for not matching the same empirical performance. State-of-the-art results are not necessary for publication but detailed discussion and analysis on why not is necessary.
>
> A8. We appreciate the comment. While we could not include the following in-depth discussion in the initial draft due to page restrictions, we will add them to the revised version. While our model is currently underperformed by Graphormer and its successors, we think the low performance is, in part, because we intentionally kept its components simple to faithfully adhere to the equivariance theory. We think it has a lot of room for performance improvement if we focus on engineering. For example, the model currently uses Laplacian eigenvectors [1] as node identifiers, which has been criticized for issues such as loss of structural information [2] and sign ambiguity [3]. We could, e.g., try to relax the theoretical requirement for orthonormality of node identifiers and incorporate more powerful node positional encodings [2, 3] as node identifiers, which could potentially yield better performance in practice. We consider engineering our model to match or outperform more sophisticated graph Transformers as a promising and important next research direction.
>
> [1] Dwivedi et al., Benchmarking Graph Neural Networks (2020)
>
> [2] Kreuzer et al., Rethinking Graph Transformers with Spectral Attention (2021)
>
> [3] Lim et al., Sign and Basis Invariant Networks for Spectral Graph Representation Learning (2022)
>
> > Can the proposed method be applied to other node-level or edge-level tasks like link prediction? (...) Besides, it would be nice to add more discussion on how the proposed method may be applied to other node and edge level tasks.
>
> We appreciate the comment and will add the following discussion to the main text. As our method produces a representation of each node and edge token, in principle, any node-level or edge-level prediction can be made by putting a prediction head on the appropriate output token. For link prediction, one could obtain the node tokens and use pairwise logistic regression head following standard practice [1], or more interestingly, could "query" the model with the concatenated pairs of node identifiers (which is just an edge token) and feed the output token to a logistic regresion head. While we demonstrate the use of our model for node-level classification task in Table 1, we consider extending our framework to more diverse tasks including link prediction as an important next direction.
>
> [1] Sankar et al., Dynamic Graph Representation Learning via Self-Attention Networks (2018)
>
> > The authors mentioned in the checklist that they included the code and data. However, their supplementary is only a PDF, I do not see the code to reproduce the concrete numbers listed in the experiment section.
>
> We apologize for the confusion, and will shortly provide the anonymized code for reproducing the results in the experiment section.

---

> > ### Author Response · Authors · 2022-08-08
> > **Code for the experiments**
> >
> > Dear reviewer YAnt,
> >
> > In the following link, we provide the code for reproducing our experiments in the paper (synthetic second-order basis approximation and PCQM4Mv2 large-scale graph regression). The data is either generated on the fly (synthetic) or automatically downloaded and processed (PCQM4Mv2).
> >
> > https://anonfiles.com/RdJ6Na2fye/code-20220808T134603Z-001_zip

---

> ### Author Response · Authors · 2022-08-02
> **Official Response to Reviewer YAnt (4/5)**
>
> > Is there any difference between modeling directed and undirected edges? If one edge (u,v) is undirected, I think the modeling of (u,v) and (v,u) should be identical. However, it seems that the edge embedding will be $[\mathbf{X}\_{(u,v)}, \mathbf{P}\_u, \mathbf{P}\_v][\mathbf{X}\_{(u,v)},\mathbf{P}\_v, \mathbf{P}\_u]$ for $(u, v)$. Also wonder how does the above issue about directedness and undirectedness generalize to higher-order hypergraphs?
>
> A5. As the reviewer noted, we treat an undirected input edge $(u, v)$ as if both diretions $(u,v)$ and $(v,u)$ are present. This leads to a pair of edge tokens $[\mathbf{X}\_{(u,v)}, \mathbf{P}\_u, \mathbf{P}\_v][\mathbf{X}\_{(v, u)},\mathbf{P}\_v, \mathbf{P}\_u]$. This is a common characteristic of tensor-based permutation equivariant neural networks, namely $k$-IGN [1, 2, 3, 4] and its successors such as PPGN [5]. Similar to the second-order case, an undirected order-$k$ input hyperedge $(v\_1, ..., v\_k)$ of an higher-order hypergraph is parsed to all possible orderings of node identifiers. While this can be easily avoided in practice by using a single token for each undirected edge and pooling the node identifiers as $\sum\_{i=1}^k\rho(\mathbf{P}\_{v\_i})$, we refrained from doing it to adhere more faithfully to the theory of $k$-IGN, and considering that having an additional edge token introduces a tolerable overhead in the second-order (graphs).
>
> [1] Maron et al., Invariant and Equivariant Graph Networks (2019)
>
> [2] Maron et al., On the Universality of Invariant Networks (2019)
>
> [3] Keriven et al., Universal Invariant and Equivariant Graph Neural Networks (2019)
>
> [4] Serviansky et al., Set2Graph: Learning Graphs From Sets (2020)
>
> [5] Maron et al., Provably Powerful Graph Networks (2019)
>
> > It’s interesting that using Laplacian eigenvectors as node identifiers and not using type id leads to comparable performance with using type identifiers on sparse inputs (the 7th row vs the final three rows in Table 1). Any insights?
>
> A6. We think that it might be possible that the network learned the sparse structures of the graphs encoded the Laplacian eigenvector, and partially utilized it to produce approximate patterns even in the absence of type identifiers. We still note that the difference coming from the presence of type identifiers is significant (please see Table 4).
>
> > Have you submitted your final results to get the numbers on the test set of PCQM4M? The authors do not seem to include a discussion on how they tune the hyperparameters on PCQM4M especially since you only have the validation set. The validation performance alone is not convincing enough, let alone the proposed method achieves higher MAE than prior transformer-based methods such as Graphormer.
>
> A7. We appreciate the comment. We will shortly submit our best-performing model (SGT (Lap) in Table 2) during the rebuttal (it requires a bit of time as the leaderboard submission requires a technical report), and will report the test score once we have it. Most of the hyperparameters of our model, including the depth, width, batch size, and learning schedule, were kept identical to the $\texttt{Graphormer-base}$ model [1] without extensive tuning for a controlled comparison. Thus, we (cautiosly) anticipate that the performance gap between the validation and the test set would not be high enough to affect our arguments in Section 4.
>
> [1] Ying et al., Do Transformers Really Perform Badly for Graph Representation? (2021)

---

> > ### Author Response · Authors · 2022-08-09
> > **Test scores of SGT (Lap) on PCQM4Mv2**
> >
> > Dear reviewer YAnt,
> >
> > We have just obtained the test scores of SGT (Lap) (best one in Table 2) on PCQM4Mv2. The result is given in the following table (please see **test-dev MAE**). Overall, we find a tendency consistent with our claims in the paper based on validation MAE scores (Table 2 in the main text).
> >
> > | method | test-dev MAE | validate MAE |
> > |---|---|---|
> > | ***Message-passing GNNs*** |
> > | GCN | 0.1398 | 0.1379 |
> > | GIN | 0.1218 | 0.1195 |
> > | GCN-VN | 0.1152 | 0.1153 |
> > | GIN-VN | 0.1084 | 0.1083 |
> > | ***Transformers with strong graph-specific modifications*** |
> > | Graphormer | N/A | 0.0864 |
> > | EGT | 0.0872 | 0.0869 |
> > | GRPE | 0.0898 | 0.0890 |
> > | ***Pure Transformer*** |
> > | SGT (Lap) | 0.0919 | 0.0910 |

---

> ### Author Response · Authors · 2022-08-02
> **Official Response to Reviewer YAnt (3/5)**
>
> > Even with the claim that the proposed method can leverage efficient transformer implementations, it does not really address my concern. 1. you never show that the model actually works in a setting where we really need an efficient implementation, i.e., other quadratic implementations do not run or oom. (...) The only limitation in my mind is that the proposed method may not necessarily generalize to large graphs as well as message passing networks. Although the paper mentioned that they can use all the efficient transformer architectures, yet there are no experiments, no evidence showing this. The full transformer architecture with quadratic complexity still runs.
>
> A4. To further demonstrate the effectiveness of our method in a more broad class of graph understanding tasks, we conducted additional experiments on transductive node classification datasets, including co-authorship (CS, Physics) [4], co-purchase (Photo, Computers) [4], and Wikipedia page networks (Chameleon, Crocodile) [5], which generally involve large-scale graphs. The statistics of the datasets are outlined below:
>
> | Dataset | CS | Physics | Photo | Computers | Chameleon | Crocodile |
> | --- | --- | --- | --- | --- | --- | --- |
> | # nodes ($n$) | 18,333 | 34,493 | 7,650 | 13,752 | 2,277 | 11,631 |
> | # edges ($m$) | 81,894 | 247,962 | 119,081 | 245,861 | 36,101 | 180,020 |
> | # classes | 15 | 5 | 8 | 10 | 6 | 6 |
>
> We randomly split the dataset into the train, validation, and test sets by randomly reserving 30 nodes per target class for validation and test, respectively, and using the rest for training. Our models are SGT equipped with Performer kernel attention. For the node identifiers, we either use $64$-dimensional near-orthonormal random vectors (denoted *SGT (Near-ORF)*) or a subset of Laplacian eigenvectors with half lowest and half highest eigenvalues (denoted *SGT (Near-ORF)*) with the number of eigenvectors chosen in the range of $64$ to $100$ based on validation performance. For *SGT (Near-ORF)*, we additionally introduce a very simple sparse equivariant basis $\mathbf{X}\_{ii} \mapsto \mathbf{X}\_{ii} + \sum\_{j\neq i}\mathbf{X}\_{ij}$ after each Transformer layer as suggested in A2 (denoted *SGT (Near-ORF) + Sp. Equiv. Basis*). We compare our method against strong GNN baselines, including the following:
> * GCN (message-passing that works well on large graphs [1])
> * GAT (message-passing based on attention)
> * GIN (message-passing with 2-WL expressiveness similar to ours [2])
> * Graphormer (graph transformer equipped with shortest path-based edge encoding and spatial encoding [3])
>
> Table 1. Results of transductive node classification experiment. OOM denotes out-of-memory error on a 24GB RTX 3080 GPU. We report aggregated test accuracy at best validation accuracy over 7 randomized runs.
> |  | CS | Physics | Photo | Computers | Chameleon | Crocodile |
> | --- | --- | --- | --- | --- | --- | --- |
> GCN | 0.895 +- 0.004 | 0.932 +- 0.004 | 0.926 +- 0.008 | 0.873 +- 0.004 | 0.593 +- 0.01 | 0.660 +- 0.01 |
> GAT | 0.893 +- 0.005 | 0.937 +- 0.01 | 0.947 +- 0.006 | **0.914 +- 0.002** | 0.632 +- 0.011 | 0.692 +- 0.017 |
> GIN | 0.895 +- 0.005 | 0.886 +- 0.046 | 0.886 +- 0.017 | 0.362 +- 0.051 | 0.479 +- 0.027 | 0.515 +- 0.041 |
> | Graphormer | 0.791 +- 0.015 | *OOM* | 0.894 +- 0.004 | 0.814 +- 0.013 | 0.457 +- 0.011 | 0.489 +- 0.014 |
> SGT (Near-ORF) + Performer | 0.882 +- 0.007 | 0.931 +- 0.009 | 0.872 +- 0.011 | 0.82 +- 0.019 | 0.568 +- 0.019 | 0.583 +- 0.024 |
> SGT (Lap) + Performer | 0.902 +- 0.004 | 0.941 +- 0.007 | 0.919 +- 0.009 | 0.86 +- 0.012 | 0.637 +- 0.032 | 0.638 +- 0.025 |
> SGT (Lap) + Performer + Sp. Equiv. Basis | **0.903 +- 0.004** | **0.950 +- 0.003** | **0.949 +- 0.007** | 0.912 +- 0.006 | **0.653 +- 0.029** | **0.718 +- 0.012** |
>
> The results are outlined in Table 1. Graphormer [4] results in out-of-memory in the Physics dataset mainly due to the spatial encoding that requires $\mathcal{O}(n^2)$ memory. By constraining the model capacity appropriately, we were able to run Graphormer on other datasets.  However, we observe a low performance, presumably due to the memory complexity that prevents depth and head scaling. As the spatial encoding is incorporated into the model via attention bias, the model strictly requires $\mathcal{O}(n^2)$ memory and cannot be easily made more efficient. On the other hand, SGT variants are able to utilize Performer kernel attention with $\mathcal{O}(n+m)$ cost, which allows using larger models to achieve the best performance in all but one dataset (Computers, where the performance is on par with the best model).
>
> [1] Shchur et al., Pitfalls of Graph Neural Network Evaluation (2019)
>
> [2] Rozemberczki et al., Multi-scale Attributed Node Embedding (2019)
>
> [3] Xu et al., How Powerful are Graph Neural Networks? (2019)
>
> [4] Ying et al., Do Transformers Really Perform Badly for Graph Representation? (2021)

---

> > ### Author Response · Authors · 2022-08-02
> > **An update in the table**
> >
> > Dear reviewer, we managed to train Graphormer on Photo, Computers, and Crocodile (previously OOM) through a bit of optimization and hyperparameter search. We updated the scores in Table 1 and updated the relevant discussions.

---

> ### Author Response · Authors · 2022-08-02
> **Official Response to Reviewer YAnt (2/5)**
>
> > One weakness/question in mind is on scalability and how to adapt to graphs with various sizes. The node identifier matrix $\mathbf{P}\in\mathbb{R}^{n\times d_p}$, where $d_p$ should be at least $n$. How can this scale to large graphs? Also the position encoding dimensions (node + type identifiers) may be much larger than the original node/edge features ($\mathbf{X}_v$). Laplacian eigenvectors as node identifiers are still different from sinusoidal positional embeddings in NLP since the sinusoidal position embeddings can be of arbitrary dimension, but for the Laplacian eigenvectors, the dimension is fixed (which is the number of nodes on the graph).  (...) How do you set the dimension of node identifiers? Is the dimension the number of nodes in the graph? How do you deal with a batch of graphs with different sizes?
>
> A2. For small $n$, we can set $d_p$ as an integer larger than the maximum number of nodes in the training set. Then, for an input graph with $n$ nodes, we zero-pad the $n\times n$ node identifier matrix to $n\times d_p$. This allows batching the node identifiers of $B$ graphs with $N$ maximum nodes into a single tensor of size $B\times N\times d_p$.
>
> For moderately large $n$ (often in an inductive learning setting), the number of nodes can be a problem as we can maximally draw $d_p$ orthonormal node identifiers for node identifier dimension $d_p$. If we need $n > d_p$, it is reasonable to introduce *near-orthonormal vectors* as node identifiers, as it is theoretically guaranteed that we can draw exponentially many ($\mathcal{O}^{\Omega(d_p)}$) $d_p$-dimensional near-orthonormal vectors [1]. Such near-orthonormal vectors include random vectors where each entry is a binary random variable with support $\{-d_p^{-1/2}, +d_p^{1/2}\}$ [1], and more practically, a subset of Laplacian eigenvectors containing $d_p<n$ vectors as often suggested in GNN literature [2]. While such vectors serve for an efficient low-rank approximation for orthonormal node identifiers, we observe that they do not necessarily harm the performance in practice. For example, in PCQM4Mv2 (Section 4.2), our preliminary observations suggest that using $d_p=16$ eigenvectors with the smallest eigenvalues for SGT (Lap) leads to the best performance even though the number of nodes can exceed $20$.
>
> However, for very large $n$ (often in a transductive learning setting), as noted by the reviewer, the approximation error of near-orthonormal node identifiers can bring a challenge to our model. More specifically, their approximation error can affect the quality of the learned equivariant basis (Lemma 1 and Definition 2), affecting the performance. In particular, we speculate that the equivariant basis ($\mu$) represented as ***sparse*** basis tensor ($\mathbf{B}^\mu$ with $\mathcal{O}(n)$ nonzero entries) can be affected; they require most of the self-attention coefficients to be zero, which particularly requires low-error orthonormality and can be challenging to model for $n \gg d_p$. Therefore, in our added experiments involving very large graphs, we additionally introduce a very simple remedy of manually adding one such sparse equivariant basis $\mathbf{X}\_{ii} \mapsto \mathbf{X}\_{ii} + \sum\_{j\neq i}\mathbf{X}\_{ij}$ after each Transformer layer. As we show in a later response, this fix is minimal, easy to implement, cost-efficient, and empirically fixes the performance issue of our model when applied to very large graphs.
>
> [1] Gorban et al., Approximation with random bases: Pro et Contra (2016)
>
> [2] Dwivedi et al., Benchmarking Graph Neural Networks (2020)
>
> > From a practitioner point of view, the experimental results are not sufficient. The authors only evaluate their model on one real world dataset PCQM4M. There are multiple graph classification/regression dataset on Open Graph Benchmark and Benchmarking GNNs.
>
> A3. While we acknowledge that our empirical evaluation is conducted only on PCQM4Mv2, we would like to note that PCQM4Mv2 is one of the largest-scale graph datasets up to date containing 3.8 million graphs [1]. This makes it one of the few suitable benchmarks to test our model, as Transformers are generally designed to work with extremely large-scale data [2, 3].
>
> To further demonstrate the effectiveness of our method in a more broad class of graph understanding tasks, we conducted additional experiments on transductive node classification datasets, which typically involve much larger graphs than PCQM4M. For this experiment, please refer to A4.
>
> [1] Hu et al., OGB-LSC: A Large-Scale Challenge for Machine Learning on Graphs (2021)
>
> [2] Brown et al., Language Models are Few-Shot Learners (2020)
>
> [3] Dosovitsky et al., An Image is Worth 16x16 Words: Transformers for Image Recognition at Scale (2020)

---

> ### Author Response · Authors · 2022-08-02
> **Official Response to Reviewer YAnt (1/5)**
>
> We thank the reviewer for the constructive comments. Below we respond to the questions.
>
> > I’m not entirely sure that the prior graph transformers cannot leverage different attention/transformer architecture. (...) 2. why can’t the prior works use the same efficient transformer implementations?
>
> A1. To address the concern, let us explain why Graphormer [1] that utilizes the fully-connected self-attention operator on nodes cannot utilize many efficient attention methods to reduce the memory complexity from $\mathcal{O}(n^2)$ to $\mathcal{O}(n)$. Prior graph transformers, including EGT [2], GRPE [3], and SAN [4], can be analyzed analogously. Let us first remind self-attention with the query, key, and value $\mathbf{Q}, \mathbf{K}, \mathbf{V}\in\mathbb{R}^{n\times d}$ and the attention matrix $\boldsymbol{\alpha}\in\mathbb{R}^{n\times n}$:
>
> $\text{Att}(\mathbf{Q}, \mathbf{K}, \mathbf{V})\_i = \sum_j\boldsymbol{\alpha}\_{ij}\mathbf{V}\_j \text{ where }\boldsymbol{\alpha}\_{ij} = \frac{\exp\left(\mathbf{Q}\_i^\top \mathbf{K}\_j/\sqrt{d}\right)}{\sum\_{k}{\exp\left(\mathbf{Q}\_i^\top \mathbf{K}\_k/\sqrt{d}\right)}}.$
>
> For graphs, as the self-attention on nodes alone cannot incorporate the edge connectivity structure, Graphormer incorporates the graph information into the self-attention matrix $\boldsymbol{\alpha}\in\mathbb{R}^{n\times n}$ utilizing the attention bias matrix $\textbf{b}\in\mathbb{R}^{n\times n}$ (referred to as the edge/spatial encoding) as the following:
>
> $\boldsymbol{\alpha}\_{ij} = \frac{\exp\left(\mathbf{Q}\_i^\top \mathbf{K}\_j/\sqrt{d} + \mathbf{b}\_{ij}\right)}{\sum\_{k}{\exp\left(\mathbf{Q}\_i^\top \mathbf{K}\_k/\sqrt{d} + \mathbf{b}\_{ik}\right)}}.$
>
> Here, the attention bias matrix $\textbf{b}\in\mathbb{R}^{n\times n}$ is the essential component to encorporate graph structure into computation. Unfortunately, this modification immediately precludes the adaptation of many efficient attention techniques developed for pure self-attention. As representative examples, let us take Performer [5] (which we mainly used in our work), linear Transformer [6], efficient Transformer [7], and Random Feature Attention [8]. The methods are based on kernelization of the $\text{Att}()$ operator as following [5]:
>
> $\text{Att}\_\phi(\mathbf{Q}, \mathbf{K}, \mathbf{V})\_i = \sum\_j\frac{\exp\left(\mathbf{Q}\_i^\top \mathbf{K}\_j/\sqrt{d}\right)}{\sum\_{k}{\exp\left(\mathbf{Q}\_i^\top \mathbf{K}\_k/\sqrt{d}\right)}}\mathbf{V}\_j$
> $= \sum\_j\frac{\phi(\mathbf{Q}\_i)^\top \phi(\mathbf{K}\_j)}{\sum\_{k}{\phi(\mathbf{Q}\_i)^\top \phi(\mathbf{K}\_k)}}\mathbf{V}\_j = \frac{\phi(\mathbf{Q}\_i)^\top\left(\sum\_j\phi(\mathbf{K}\_j)\mathbf{V}\_j^\top\right)}{{\phi(\mathbf{Q}\_i)^\top\left(\sum\_{k}\phi(\mathbf{K}\_k)\right)}}.$
>
> The above factorization of the exponential into a pairwise dot product, in turn, eliminates the need to explicitly compute the attention matrix for computing $\text{Att}\_\phi()$ and consequently reduces both time and memory cost to $\mathcal{O}(n)$. Unfortunately, Graphormer and related variations are fundamentally unable to utilize the method. The bias term $\mathbf{b}\_{ij}$ is added to the dot product before the exponential, requiring that the full pairwise self-attention matrix $\boldsymbol{\alpha}\in\mathbb{R}^{n\times n}$ is always explicitly computed to obtain the output and leading to $\mathcal{O}(n^2)$.
>
> While our above explanation mainly regards kernelization methods, a number of other efficient Transformers, including Set Transformer [9], LUNA [10], Linformer [11], Nyströmformer[12], Perceiver [13], and Perceiver-IO [14] are not applicable to Graphormer due to similar reasons. We appreciate the comment and will add relevant discussions to the main text.
>
> [1] Ying et al., Do Transformers Really Perform Bad for Graph Representation? (2022)
>
> [2] Hussain et al., Global Self-Attention as a Replacement for Graph Convolution (2022)
>
> [3] Park et al., GRPE: Relative Positional Encoding for Graph Transformer (2022)
>
> [4] Kreuzer et al., Rethinking Graph Transformers with Spectral Attention (2022)
>
> [5] Choromanski et al., Rethinking Attention with Performers (2020)
>
> [6] Katharopoulos et al., Transformers are RNNs: Fast Autoregressive Transformers with Linear Attention (2020)
>
> [7] Shen et al., Efficient Attention: Attention with Linear Complexities (2018)
>
> [8] Peng et al., Random Feature Attention (2021)
>
> [9] Lee et al., Set Transformer: A Framework for Attention-based Permutation-Invariant Neural Networks (2018)
>
> [10] Ma et al., Luna: Linear Unified Nested Attention (2021)
>
> [11] Wang et al., Linformer: Self-Attention with Linear Complexity (2020)
>
> [12] Xiong et al., Nyströmformer: A Nyström-Based Algorithm for Approximating Self-Attention (2021)
>
> [13] Jaegle et al., Perceiver: General Perception with Iterative Attention (2021)
>
> [14] Jaegle et al., Perceiver IO: A General Architecture for Structured Inputs & Outputs (2021)

---

> ### Comment · Reviewer_YAnt · 2022-08-09
> **Thank you for the detailed response**
>
> Thank you for the detailed response. Most of my concerns are addressed. Hence I raised my score to 6 (I am actually a 5.5 now). I strongly suggest the authors add all the discussions to the final version, especially on how the model needs modification and approximations when $n > d_p$. Following that, it would be helpful if the authors can evaluate the model on three datasets, each corresponding to one setting: (1) small $n$, (2) moderately large $n$ and (3) very large $n$. I would assume PCQM4M corresponds to the first setting.
>
> Besides, I am also not entirely satisfied with the discussion on why the model performs worse than other Graph Transformers. The authors mentioned several conjectures including the loss of structural information of Laplacian eigenvectors and sign ambiguity. What exactly do you refer to as "structural information"? Can you give examples and ground this conjecture on PCQM4M dataset? What type of structural information will be lossy for Laplacian eigenvectors?
>
> The authors propose one potential solution with the requirement of relaxing the theoretical assumptions, especially on orthonormality. I think more experiments on that would be helpful to gain more insights. I'm curious to see whether such expressiveness is useful in real world application, which I highly doubt. And actually the result of the paper is exactly saying such expressiveness is not necessary. Real world graphs often have rich node/edge/graph features, rendering the expressiveness less useful. I wonder why the authors do not conduct more expriments on graphs without any features, just demonstrating a setting where the model can actually shine.
>
> Generally, I think it's a solid paper on method and theoretical analysis. However, I maintain my opinion that significantly more experiments are needed for the paper to have more impact.

---

### Official Review · Reviewer_To3r · 2022-07-12

**Rating:** 7
**Confidence:** 3
**Soundness:** 3 good
**Presentation:** 3 good
**Contribution:** 3 good

**Summary:**

This paper proposes a variant of graph transformers, the Soft Graph Transformer (SGT). In particular, all nodes and edges are simply treated as independent tokens, and then they are augmented with token embedding. The authors prove this approach is theoretically at least as expressive as an invariant graph network, which is already more expressive than all message-passing graph neural networks. SGT performs significantly better than all GNNs and is competitive with Transformer variants with strong graph-specific architectural components.

**Questions:**

I have no questions.

**Ethics Review Area:**

["I don’t know"]

**Limitations:**

The authors have addressed the limitations.

**Strengths And Weaknesses:**

Strengths:
The motivation is clear. This paper proposes an approach of applying a standard Transformer directly for graphs. Although the algorithm is simple, it is shown that this simple approach yields a powerful graph learner in terms of both theory and experiment. The theoretical analysis is comprehensive and proves the effectiveness of the proposed method.

Weaknesses:
The experiments are not very convincing. Only one graph dataset, PCQM4Mv2, is considered in the paper. If one to two new datasets are added, that would be great. It will be better to conduct more experimental ablation studies, such as the effects of type embedding. The results seem not strong enough, and more experiments can make this paper stronger.

---

> ### Author Response · Authors · 2022-08-02
> **Official Response to Reviewer To3r (2/2)**
>
> (continued from 1/2)
>
> Let us finish by explaining how we applied SGT for this task. Considering large $n$, an immediate challenge for our model is dealing with the orthonormality assumption on the node identifiers (Lemma 1), as the maximal number of orthonormal node identifers is upper bounded by its dimension $d_p$. In this case, it is reasonable to introduce *near-orthonormal vectors* as node identifiers, as it is theoretically guaranteed that we can draw an exponential number $\mathcal{O}^{\Omega(d_P)}$ of $d_p$-dimensional near-orthonormal vectors [7]. For *SGT (Near-ORF)*, we used $d_p = 64$-dimensional random node identifiers where each entry is sampled from $\{-1/d_p, +1/d_p\}$ under equal probability [7]. For *SGT (Lap)*, we used a subset of the Laplacian eigenvectors as node identifiers, more specifically $d_p/2$ eigenvectors with lowest eigenvalues and $d_p/2$ eigenvectors with highest eigenvalues, and choose $d_p$ in the range of $64$ to $100$ based on validation performance.
>
> While *Near-ORF* and *Lap* can theoretically serve as an efficient low-rank approximation for orthonormal node identifiers, their approximation can affect the quality of modeled equivariant basis. In particular, equivariant basis ($\mu$) represented as ***sparse*** basis tensor ($\mathbf{B}^\mu$; Definition 2) are expected to be affected more, as they require most entries to be zero. To remedy this, we take a simple approach of residually adding one of such sparse equivariant basis operators $\mathbf{X}\_{ii} \mapsto \mathbf{X}\_{ii} + \sum\_{j\neq i}\mathbf{X}\_{ij}$ explicitly after each Transformer layer. We denote this variant as *SGT (Lap) + Performer + Sp. Equiv. Basis*. This fix is minimal, easy to implement, and highly efficient as it only requires a single $\texttt{torch.coalesce()}$ call, and also empirically effective, as shown in Table 1.
>
> [1] Hu et al., OGB-LSC: A Large-Scale Challenge for Machine Learning on Graphs (2021)
>
> [2] Brown et al., Language Models are Few-Shot Learners (2020)
>
> [3] Dosovitsky et al., An Image is Worth 16x16 Words: Transformers for Image Recognition at Scale (2020)
>
> [4] Shchur et al., Pitfalls of Graph Neural Network Evaluation (2019)
>
> [5] Rozemberczki et al., Multi-scale Attributed Node Embedding (2019)
>
> [6] Xu et al., How Powerful are Graph Neural Networks? (2019)
>
> [7] Gorban et al., Approximation with random bases: Pro et Contra (2016)
>
> [8] Ying et al., Do Transformers Really Perform Badly for Graph Representation? (2021)
>
>
> > It will be better to conduct more experimental ablation studies, such as the effects of type embedding.
>
> A2. We appreciate the comment. We would like to gently remind the reviewer that Table 1 (Section 4.1) provides a thorough analysis on the role of node and type identifiers in a close proximity to our theory in Section 3. Specifically, we have shown that both the node and type identifiers contribute to the approximation ability for equivariant basis, while node identifiers play a more critical role relative to type identifiers. Furthermore, please note that we conducted a comprehensive ablation study on the choice of node identifiers on the PCQM4Mv2 dataset (Table 2). Specifically, we have shown that, while node identifier based on orthonormal random embedding (ORF) works to some degree compared to absence of node identifier, the one based on the Laplacian eigenvector (Lap) provides additional information on the structure of a graph and empirically performs better. If the reviewer has additional concerns regarding the ablation study, please feel free to suggest us. We are happy to elaborate more in the rolling discussion period.

---

> > ### Author Response · Authors · 2022-08-02
> > **An update in the table**
> >
> > Dear reviewer, we managed to train Graphormer on Photo, Computers, and Crocodile (previously OOM) through a bit of optimization and hyperparameter search. We updated the scores in Table 1 and updated the relevant discussions.

---

> ### Author Response · Authors · 2022-08-02
> **Official Response to Reviewer To3r (1/2)**
>
> We thank the reviewer for the positive comments. Below we respond to the questions.
>
> > The experiments are not very convincing. Only one graph dataset, PCQM4Mv2, is considered in the paper. If one to two new datasets are added, that would be great. (...) The results seem not strong enough, and more experiments can make this paper stronger.
>
> A1. While we acknowledge that our empirical evaluation is conducted only on PCQM4Mv2, we would like to first appeal that PCQM4Mv2 is one of the largest-scale graph datasets up to date containing 3.8 million graphs [1]. This makes it one of the few suitable benchmarks to test our model, as Transformers are generally designed to work with extremely large-scale data [2, 3].
>
> To further demonstrate the effectiveness of our method in a more broad class of graph understanding tasks, we conducted additional experiments on transductive node classification datasets, including co-authorship (CS, Physics) [4], co-purchase (Photo, Computers) [4], and Wikipedia page networks (Chameleon, Crocodile) [5], which generally involve large-scale graphs. The statistics of the datasets are outlined below:
>
> | Dataset | CS | Physics | Photo | Computers | Chameleon | Crocodile |
> | --- | --- | --- | --- | --- | --- | --- |
> | # nodes ($n$) | 18,333 | 34,493 | 7,650 | 13,752 | 2,277 | 11,631 |
> | # edges ($m$) | 81,894 | 247,962 | 119,081 | 245,861 | 36,101 | 180,020 |
> | # classes | 15 | 5 | 8 | 10 | 6 | 6 |
>
> We randomly split the dataset into the train, validation, and test sets by reserving 30 random nodes per class for validation and test, respectively, and using the rest for training. We experiment with simple variants of SGT equipped with Performer kernel attention (*details can be found at the end of the response*), and compare them against strong GNN baselines, including the following:
> * GCN (message-passing that works well on large graphs [4])
> * GAT (message-passing based on attention)
> * GIN (message-passing with 2-WL expressiveness guarantee similar to ours [6])
> * Graphormer (graph transformer equipped with shortest path-based edge encoding and spatial encoding [8])
>
> Table 1. Results of transductive node classification experiment. OOM denotes out-of-memory error on a 24GB RTX 3080 GPU. We report aggregated test accuracy at best validation accuracy over 7 randomized runs.
> |  | CS | Physics | Photo | Computers | Chameleon | Crocodile |
> | --- | --- | --- | --- | --- | --- | --- |
> GCN | 0.895 +- 0.004 | 0.932 +- 0.004 | 0.926 +- 0.008 | 0.873 +- 0.004 | 0.593 +- 0.01 | 0.660 +- 0.01 |
> GAT | 0.893 +- 0.005 | 0.937 +- 0.01 | 0.947 +- 0.006 | **0.914 +- 0.002** | 0.632 +- 0.011 | 0.692 +- 0.017 |
> GIN | 0.895 +- 0.005 | 0.886 +- 0.046 | 0.886 +- 0.017 | 0.362 +- 0.051 | 0.479 +- 0.027 | 0.515 +- 0.041 |
> | Graphormer | 0.791 +- 0.015 | *OOM* | 0.894 +- 0.004 | 0.814 +- 0.013 | 0.457 +- 0.011 | 0.489 +- 0.014 |
> SGT (Near-ORF) + Performer | 0.882 +- 0.007 | 0.931 +- 0.009 | 0.872 +- 0.011 | 0.82 +- 0.019 | 0.568 +- 0.019 | 0.583 +- 0.024 |
> SGT (Lap) + Performer | 0.902 +- 0.004 | 0.941 +- 0.007 | 0.919 +- 0.009 | 0.86 +- 0.012 | 0.637 +- 0.032 | 0.638 +- 0.025 |
> SGT (Lap) + Performer + Sp. Equiv. Basis | **0.903 +- 0.004** | **0.950 +- 0.003** | **0.949 +- 0.007** | 0.912 +- 0.006 | **0.653 +- 0.029** | **0.718 +- 0.012** |
>
> The results are outlined in Table 1. Graphormer [8] results in out-of-memory in the Physics dataset mainly due to the spatial encoding that requires $\mathcal{O}(n^2)$ memory. By constraining the model capacity appropriately, we were able to run Graphormer on other datasets. However, we observe a low performance, presumably due to the memory complexity that prevents depth and head scaling. As the spatial encoding is incorporated into the model via attention bias, the model strictly requires $\mathcal{O}(n^2)$ memory and cannot be easily made more efficient. On the other hand, SGT variants are able to utilize Performer kernel attention with $\mathcal{O}(n+m)$ cost, which allows using larger models to achieve the best performance in all but one dataset (Computers, where the performance is on par with the best model).
>
> (continued to 2/2)

---

### Official Review · Reviewer_E51P · 2022-07-15

**Rating:** 5
**Confidence:** 3
**Soundness:** 3 good
**Presentation:** 4 excellent
**Contribution:** 3 good

**Summary:**

The authors of the paper claim that Transformers can be as expressive as graph-invariant networks by applying them directly to a graph and treating all nodes and edges in the graph as tokens. Using appropriate node and type identifiers, self-attention can approximate any permutation equivariant linear operator in the graph. Orthogonal random features and Laplacian feature vectors for node identifiers are proposed. The final proposed model, Soft Graph Transformer (SGT), achieves performance improvements over the GNN baseline on the PCQM4Mv2 dataset.

**Questions:**

1. Are node and type identifiers learned?

**Limitations:**

1. Evaluate on only one graph dataset.

**Strengths And Weaknesses:**

Strengths:
1. It is novel to treat nodes and edges as input tokens to Transformers and add node and type identifiers.
2. It is shown that SGT is as expressive as 2-IGN with equivariant linear layers.

Weakness:
1. Evaluate on only one graph dataset.

---

> ### Author Response · Authors · 2022-08-02
> **Official Response to Reviewer E51P (2/2)**
>
> (continued from 1/2)
>
> Let us finish by explaining how we applied SGT for this task. Considering large $n$, an immediate challenge for our model is dealing with the orthonormality assumption on the node identifiers (Lemma 1), as the maximal number of orthonormal node identifers is upper bounded by its dimension $d_p$. In this case, it is reasonable to introduce *near-orthonormal vectors* as node identifiers, as it is theoretically guaranteed that we can draw an exponential number $\mathcal{O}^{\Omega(d_P)}$ of $d_p$-dimensional near-orthonormal vectors [7]. For *SGT (Near-ORF)*, we used $d_p = 64$-dimensional random node identifiers where each entry is sampled from $\{-1/d_p, +1/d_p\}$ under equal probability [7]. For *SGT (Lap)*, we used a subset of the Laplacian eigenvectors as node identifiers, more specifically $d_p/2$ eigenvectors with lowest eigenvalues and $d_p/2$ eigenvectors with highest eigenvalues, and choose $d_p$ in the range of $64$ to $100$ based on validation performance.
>
> While *Near-ORF* and *Lap* can theoretically serve for an efficient low-rank approximation for orthonormal node identifiers, their approximation can affect the quality of modeled equivariant basis. In particular, equivariant basis ($\mu$) represented as ***sparse*** basis tensor ($\mathbf{B}^\mu$; Definition 2) are expected to be affected more, as they require most entries to be zero. To remedy this, we take a simple approach of residually adding one of such sparse equivariant operators $\mathbf{X}\_{ii}\mapsto \mathbf{X}\_{ii} + \sum\_{j\neq i} \mathbf{X}\_{ij}$ explicitly after each Transformer layer. We denote this variant as *SGT (Lap) + Performer + Sp. Equiv. Basis*. This fix is minimal, easy to implement, and highly efficient as it only requires a single $\texttt{torch.coalesce()}$ call, and also empirically effective as shown in Table 1.
>
> [1] Hu et al., OGB-LSC: A Large-Scale Challenge for Machine Learning on Graphs (2021)
>
> [2] Brown et al., Language Models are Few-Shot Learners (2020)
>
> [3] Dosovitsky et al., An Image is Worth 16x16 Words: Transformers for Image Recognition at Scale (2020)
>
> [4] Shchur et al., Pitfalls of Graph Neural Network Evaluation (2019)
>
> [5] Rozemberczki et al., Multi-scale Attributed Node Embedding (2019)
>
> [6] Xu et al., How Powerful are Graph Neural Networks? (2019)
>
> [7] Gorban et al., Approximation with random bases: Pro et Contra (2016)
>
> [8] Ying et al., Do Transformers Really Perform Badly for Graph Representation? (2021)

---

> > ### Author Response · Authors · 2022-08-02
> > **An update in the table**
> >
> > Dear reviewer, we managed to train Graphormer on Photo, Computers, and Crocodile (previously OOM) through a bit of optimization and hyperparameter search. We updated the scores in the table and updated the relevant discussions.

---

> > > ### Comment · Reviewer_E51P · 2022-08-09
> > > **I maintain my original rating**
> > >
> > > Thanks to the author for the detailed reply. Most of my concerns have been addressed. I keep my original rating.

---

> ### Author Response · Authors · 2022-08-02
> **Official Response to Reviewer E51P (1/2)**
>
> We thank the reviewer for the positive comments. Below we respond to the questions.
>
> > Are node and type identifiers learned?
>
> A1. For the type identifiers (Section 2), we initialize them as learnable vectors and jointly train them with the model. For the node identifiers (Section 2), our theory only requires them to be orthonormal (Section 3.3); therefore, we choose them as simple randomized vectors (ORF) or matrix factorization-based vectors (Lap) that are not learned but algorithmically obtained at each forward pass.
>
> > Evaluated on only one graph dataset.
>
> A2. While we acknowledge that our empirical evaluation is conducted only on PCQM4Mv2, we would like to note that PCQM4Mv2 is one of the largest-scale graph datasets up to date containing 3.8 million graphs [1]. This makes it one of the few suitable benchmarks to test our model, as Transformers are generally designed to work with extremely large-scale data [2, 3].
>
> To further demonstrate the effectiveness of our method in a more broad class of graph understanding tasks, we conducted additional experiments on transductive node classification datasets, including co-authorship (CS, Physics) [4], co-purchase (Photo, Computers) [4], and Wikipedia page networks (Chameleon, Crocodile) [5], which generally involve large-scale graphs. The statistics of the datasets are outlined below:
>
> | Dataset | CS | Physics | Photo | Computers | Chameleon | Crocodile |
> | --- | --- | --- | --- | --- | --- | --- |
> | # nodes ($n$) | 18,333 | 34,493 | 7,650 | 13,752 | 2,277 | 11,631 |
> | # edges ($m$) | 81,894 | 247,962 | 119,081 | 245,861 | 36,101 | 180,020 |
> | # classes | 15 | 5 | 8 | 10 | 6 | 6 |
>
> We randomly split the dataset into train, validation, and test sets by randomly reserving 30 random nodes per class for validation and test, respectively, and using the rest for training.
> |  | CS | Physics | Photo | Computers | Chameleon | Crocodile |
> | --- | --- | --- | --- | --- | --- | --- |
> GCN | 0.895 +- 0.004 | 0.932 +- 0.004 | 0.926 +- 0.008 | 0.873 +- 0.004 | 0.593 +- 0.01 | 0.660 +- 0.01 |
> GAT | 0.893 +- 0.005 | 0.937 +- 0.01 | 0.947 +- 0.006 | **0.914 +- 0.002** | 0.632 +- 0.011 | 0.692 +- 0.017 |
> GIN | 0.895 +- 0.005 | 0.886 +- 0.046 | 0.886 +- 0.017 | 0.362 +- 0.051 | 0.479 +- 0.027 | 0.515 +- 0.041 |
> | Graphormer | 0.791 +- 0.015 | OOM | 0.894 +- 0.004 | 0.814 +- 0.013 | 0.457 +- 0.011 | 0.489 +- 0.014 |
> SGT (Near-ORF) + Performer | 0.882 +- 0.007 | 0.931 +- 0.009 | 0.872 +- 0.011 | 0.82 +- 0.019 | 0.568 +- 0.019 | 0.583 +- 0.024 |
> SGT (Lap) + Performer | 0.902 +- 0.004 | 0.941 +- 0.007 | 0.919 +- 0.009 | 0.86 +- 0.012 | 0.637 +- 0.032 | 0.638 +- 0.025 |
> SGT (Lap) + Performer + Sp. Equiv. Basis | **0.903 +- 0.004** | **0.950 +- 0.003** | **0.949 +- 0.007** | 0.912 +- 0.006 | **0.653 +- 0.029** | **0.718 +- 0.012** |
>
> The results are outlined in the above Table. Graphormer [8] results in out-of-memory in the Physics dataset mainly due to the spatial encoding that requires $\mathcal{O}(n^2)$ memory. By constraining the model capacity appropriately, we were able to run Graphormer on other datasets. However, we observe a relatively low performance, presumably due to the memory complexity that prevents depth and head scaling. As the spatial encoding is incorporated into the model via attention bias, the model strictly requires $\mathcal{O}(n^2)$ memory and cannot be easily made more efficient. On the other hand, SGT variants are able to utilize Performer kernel attention with $\mathcal{O}(n+m)$ cost, which allows using larger models to achieve the best performance in all but one dataset (Computers, where the performance is on par with the best model).
>
> (continued to 2/2)

---

### Author Response · Authors · 2022-08-02
**Common Response (to all reviewers)**

We sincerely thank all the reviewers for the constructive comments. We are encouraged by the positive feedback on the novelty (E51P, YAnt) and clarity (To3r) of our work, in particular the theoretical contributions (E51P, To3r, YAnt) and their validation with the synthetic experiment (YAnt). Our responses to the specific questions, mainly regarding additional experiments, can be found in respective comments.

---

### Meta-Review · Area_Chair_tC2c · 2022-08-28

**Recommendation:** Accept
**Confidence:** Certain

**Metareview:**

The paper applies transformers directly to a graph by treating all nodes and edges in the graph as tokens. The author prove this approach is theoretically at least as expressive as an invariant graph network, which is already more expressive than all message-passing graph neural networks. The approach is simple and interesting. Reviewers had concerns on the empirical studies as only one dataset was used. In the response, the authors have partially addressed the concerns by offering more results. More discussion/analysis of the empirical results is expected. Considering that the paper is mainly on the theoretical side and the paper does provides interesting new insights, I'd recommend acceptance.

**Award:**

No

---

### Decision · Program_Chairs · 2022-09-14

Accept